# EFFICIENTLY LEARNING AT TEST-TIME: ACTIVE FINE-TUNING OF LLMS

**Jonas Hübotter,**[*] **Sascha Bongni, Ido Hakimi, Andreas Krause**
ETH Zürich, Switzerland

## ABSTRACT

Recent efforts in fine-tuning language models often rely on automatic data selection, commonly using Nearest Neighbors retrieval from large datasets. However, we theoretically show that this approach tends to select redundant data, limiting its effectiveness or even hurting performance. To address this, we introduce SIFT, a data selection algorithm designed to reduce uncertainty about the model's response given a prompt, which unifies ideas from retrieval and active learning. Whereas Nearest Neighbor retrieval typically fails in the presence of information duplication, SIFT accounts for information duplication and optimizes the overall information gain of the selected examples. We focus our evaluations on fine-tuning at test-time for prompt-specific language modeling on the Pile dataset, and show that SIFT consistently outperforms Nearest Neighbor retrieval, with minimal computational overhead. Moreover, we show that our uncertainty estimates can predict the performance gain of test-time fine-tuning, and use this to develop an adaptive algorithm that invests test-time compute proportional to realized performance gains. We provide the `activeft` (Active Fine-Tuning) library which can be used as a drop-in replacement for Nearest Neighbor retrieval.

## 1 INTRODUCTION

The standard paradigm of machine learning separates training and testing. Training aims to learn a model by *inductively* extracting general rules from data, and testing applies this model to new, unseen data. We investigate an alternative *transductive* paradigm where the model is fine-tuned at test-time specifically to the given task. Variations of this paradigm have been studied since the inception of machine learning as a field. Early examples are local learning (Cleveland, 1979; Cleveland & Devlin, 1988; Atkeson et al., 1997) and local fine-tuning (Bottou & Vapnik, 1992). More recently, with the advent of large pre-trained models which have good representations and are strong foundations for fine-tuning, the idea of *test-time fine-tuning* has re-gained attention (Krause et al., 2018; 2019; Sun et al., 2020). Hardt & Sun (2024) show that fine-tuning on data related to the prompt to a large language model (LLM) can significantly improve performance. Also, test-time fine-tuning is the central component of state-of-the-art approaches to the ARC challenge (Chollet, 2019; Cole & Osman, 2023; Akyürek et al., 2024), a non-saturated benchmark which is intended to test reasoning capabilities based on "core knowledge" rather than mere memorization.

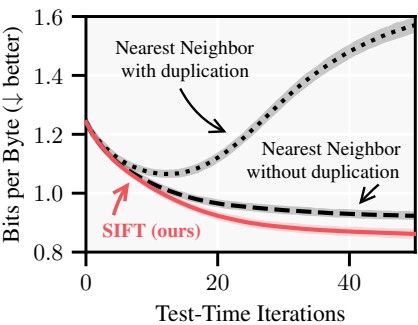

Figure 1: Selecting fine-tuning data using SIFT (red) robustly outperforms Nearest Neighbor retrieval (black) and avoids the failure-mode of Nearest Neighbor retrieval where the same data is selected repeatedly, which is a common result of information duplication.

***Active Fine-Tuning:* Effective data selection for fine-tuning LLMs** Test-time fine-tuning demands automatic data selection since manually selecting data for each test instance is infeasible. Moreover, the sample efficiency of test-time fine-tuning is a central bottleneck as the number of gradient steps is directly proportional to inference time. Previous works on data selection for fine-tuning LLMs have fundamentally relied on Nearest Neighbor retrieval within some embedding space (Hardt & Sun, 2024; Xia et al., 2024). We show theoretically and empirically that Nearest

---

[*]Correspondence to `jonas.huebotter@inf.ethz.ch`

Neighbor retrieval is insufficient for fine-tuning LLMs since it can lead to the selection of redundant data. Notably, recent works using influence functions for data selection such as Xia et al. (2024) have pointed out this limitation. In contrast, a large body of work on (inductive) active learning has studied non-redundant data selection (e.g., Sener & Savarese, 2017; Ash et al., 2020; Yehuda et al., 2021; Kirsch et al., 2018) that covers the data manifold well (cf. Figure 2). Retrieval and active learning can be seen as two extreme ends of a spectrum: retrieval selects relevant but potentially redundant data, while active learning selects diverse but potentially irrelevant data.

We bridge this gap by unifying ideas from retrieval and active learning in SIFT, an algorithm based on emerging literature on transductive active learning (Hübotter et al., 2024) that **Selects Informative data for Fine-Tuning** as illustrated in Figure 2. Our results show that SIFT leads to substantial improvements in performance and efficiency. Concretely, we show the following:

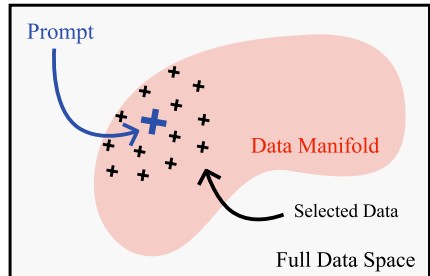

Figure 2: We consider a scenario where we have a pre-trained language model capturing a latent manifold (red) in the large sequence space (white). We aim to improve the models performance on a given prompt (blue) by *efficiently* fine-tuning the model on *few* relevant and diverse data points (black) at test-time.

1. **Nearest Neighbor retrieval is insufficient (§2)**: We prove that selecting the top-$N$ highest scoring points from a large dataset according to a fixed scoring function leads to the selection of redundant data.

2. **SIFT estimates uncertainty about responses (§3)**: We develop the notion of *uncertainty about the response to the prompt*, and derive an anytime high probability bound to the total variation distance between the model's distribution over responses and the ground truth which is governed by this uncertainty.

3. **SIFT provably reduces uncertainty (§4)**: We propose SIFT, an algorithm that selects data which reduces uncertainty about the response to the prompt. We prove statistical rates for the uncertainty reduction (§4.1) and show that SIFT is compute-efficient, with minimal overhead compared to Nearest Neighbor retrieval (§4.2).

4. **SIFT performs better and is more robust than Nearest Neighbor retrieval (§5)**: We find that fine-tuning an LLM on data selected by SIFT consistently and robustly improves performance, which is not the case with Nearest Neighbor retrieval. Moreover, our results suggest that at test-time, an LLM might be able to learn more effectively through fine-tuning than from its context.

5. **SIFT can invest test-time compute proportionally to performance gains (§6)**: We observe that our uncertainty estimates can accurately predict the performance gain of test-time fine-tuning. Motivated by this, we dynamically adapt compute to the expected performance gain.

## 2 TEST-TIME FINE-TUNING

We define test-time fine-tuning of LLMs (Hardt & Sun, 2024) as follows. We consider a domain $\mathcal{X}$ of token sequences and assume that we have access to a large dataset of examples $\mathcal{D} \subseteq \mathcal{X}$ which we call the *data space*. We further assume that we have access to a pre-trained autoregressive language model that maps token sequences $\mathcal{X}$ to probability distributions over the next token from a vocabulary of size $V$. Our work addresses the central question:

*Given a prompt $\boldsymbol{x}^{\star} \in \mathcal{X}$, how can we effectively select fine-tuning data from the large dataset $\mathcal{D}$ such that the fine-tuned model performs well on the prompt?*

We then fine-tune the model for a single gradient step on each selected sequence.

Locally adjusting a model at test-time has gained popularity with few-shot in-context learning (Brown et al., 2020; Wei et al., 2022b; Bubeck et al., 2023; OpenAI, 2024) and retrieval augmented generation (RAG, Lewis et al., 2019; Guu et al., 2020; Borgeaud et al., 2022). In contrast to this approach, test-time fine-tuning works by fine-tuning the parameters of a pre-trained model at test-time specifically to each prompt. Notably, test-time fine-tuning takes time linear in the number of tokens whereas in-context learning with a transformer has quadratic complexity (Vaswani et al., 2017). Next to this, Hardt & Sun (2024) and other works have found (test-time) fine-tuning to perform

substantially better than in-context learning (Hu et al., 2022; Mosbach et al., 2023). This work further improves the performance of test-time fine-tuning. Prior work has also studied how one can explicitly meta-learn the ability to perform test-time fine-tuning (Finn et al., 2017; Sun et al., 2024), though we find this capability to emerge even from models that are not explicitly trained in this way.

The central question studied in this work also arises when fine-tuning LLMs during post-training. For example, in targeted instruction tuning, the goal is to fine-tune a model to obtain desired capabilities, which are commonly embodied by a set of examples $\boldsymbol{x}^\star$ (Xia et al., 2024). The extension of our work to such a "batched" setting is straightforward.

## 2.1 NEAREST NEIGHBOR RETRIEVAL IS INSUFFICIENT

Prior work on data selection for fine-tuning has relied on Nearest Neighbor retrieval. The idea of making predictions on $\boldsymbol{x}^\star$ depending on its nearest neighbors has been around as long as machine learning itself (Fix & Hodges Jr., 1951; Cover & Hart, 1967). Bottou & Vapnik (1992) were the first to apply this idea to the fine-tuning of convolutional neural networks by selecting the nearest neighbors of a test image in pixel-space. More recently, due to advances in representation learning (Devlin et al., 2018; Reimers & Gurevych, 2019) and efficiency (e.g., Johnson et al., 2019), Nearest Neighbor retrieval has regained attention and been applied to test-time fine-tuning (Hardt & Sun, 2024).

> **Prompt:** What is the age of Michael Jordan and how many kids does he have?
>
> **Nearest Neighbor:**
> 1. The age of Michael Jordan is 61 years.
> 2. Michael Jordan was born on February 17, 1963.
>
> **SIFT (ours):**
> 1. The age of Michael Jordan is 61 years.
> 2. Michael Jordan has five children.

Figure 3: We retrieve two data points to answer the prompt. Nearest Neighbor selects redundant data, while SIFT yields maximal information (cf. §L).

Xia et al. (2024) use influence functions (Cook, 1977; Koh & Liang, 2017; Pruthi et al., 2019) to select data for fine-tuning LLMs. This line of work aims to select data that reduces a first-order Taylor approximation to the test loss after fine-tuning, an approach that corresponds to Nearest Neighbor retrieval in a certain embedding space. They highlight two main limitations of the use of influence functions and Nearest Neighbor retrieval for data selection:

- Nearest Neighbor retrieval leads to the selection of redundant data. Figure 3 illustrates this limitation with a qualitative example. We formalize this limitation in Proposition K.1, which we summarize here informally:

  **Informal Proposition 2.1.** *Selecting the top-$N$ nearest neighbors from the data space (according to cosine similarity or Euclidean distance) may not reduce the uncertainty about the response to the prompt beyond fine-tuning on the closest neighbor. Every additional passage may be redundant.*

- Nearest Neighbor retrieval selects data with high positive cosine similarity to the prompt. Yet, data with high *negative* cosine similarity can be equally informative as data with high positive cosine similarity (Xia et al., 2024, Appendix K.2), but is ignored by standard Nearest Neighbor retrieval.

In this work, we propose SIFT and show that it naturally addresses both limitations. SIFT unifies work on retrieval, which finds relevant but redundant data, and active learning (AL), which finds non-redundant but irrelevant data. In §B, we discuss how SIFT relates to prior work in retrieval and AL.

## 3 PRELIMINARIES: UNCERTAINTY ESTIMATION FOR FINE-TUNING

We suppose the assigned probability that $y \in [V]$ is the class label of an input $\boldsymbol{x} \in \mathcal{X}$ is given by $s_y(\boldsymbol{f}^\star(\boldsymbol{x}))$, where $s_y$ is the softmax $s_y(\boldsymbol{f}) \doteq \exp(f_y)/(\sum_{i=1}^{V} \exp(f_i))$. That is, $\boldsymbol{f}^\star(\boldsymbol{x})$ denotes the "ground truth" logits for a given input $\boldsymbol{x}$. In the context of language modeling, $V$ is the number of tokens in the vocabulary, and $y$ denotes the index of the next token. We defer all proofs to Appendix K.

We use a surrogate model to quantify the informativeness of data, which we define next.

**Assumption 3.1** (*Surrogate model:* Linear model class within a known latent space). We assume $\boldsymbol{f}^\star(\boldsymbol{x}) = \boldsymbol{W}^\star\boldsymbol{\phi}(\boldsymbol{x})$ with unknown $\boldsymbol{W}^\star \in \mathbb{R}^{V \times d}$ and where $\boldsymbol{\phi}(\cdot) \in \mathbb{R}^d$ denotes known embeddings.

The surrogate model uses the latent space induced by the pre-trained model to describe the data manifold. We emphasize that while SIFT relies on this surrogate model for data selection, it still fine-

tunes the full pre-trained model, including latent features. Surrogate dense embedding models of this kind have been used extensively for data selection via Nearest Neighbor retrieval (e.g., Lewis et al., 2019; Karpukhin et al., 2020; Borgeaud et al., 2022; Xia et al., 2024), and to understand the training dynamics and generalization of large neural networks (e.g., Jacot et al., 2017; Lee et al., 2018; Malladi et al., 2023; Templeton et al., 2024; Park et al., 2024). Furthermore, a surrogate model that assumes linearity in some fixed latent space may be a reasonable approximation for test-time fine-tuning since the latent space of the unfrozen model is not expected to change substantially by a few gradient steps.

In this work, we explore a scenario where we have a pre-trained model $\boldsymbol{f}^{\mathrm{pre}}(\boldsymbol{x}) = \boldsymbol{W}^{\mathrm{pre}}\boldsymbol{\phi}(\boldsymbol{x})$. We let $\boldsymbol{f}(\boldsymbol{x}; \boldsymbol{W}) \doteq \boldsymbol{W}\boldsymbol{\phi}(\boldsymbol{x})$ and denote by $\mathcal{L}(\boldsymbol{W}; D)$ the negative log-likelihood loss of $\boldsymbol{f}(\cdot; \boldsymbol{W})$ on a dataset $D$ of inputs $\boldsymbol{x}$ with corresponding class labels $y$: $\mathcal{L}(\boldsymbol{W}; D) \doteq -\sum_{(\boldsymbol{x}, y) \in D} \log s_y(\boldsymbol{f}(\boldsymbol{x}; \boldsymbol{W}))$.

**Uncertainty Estimation**  Our first intermediate goal is to estimate the uncertainty about the response to a given prompt $\boldsymbol{x}^\star$ after having fine-tuned on selected data $D_n$ of size $n$. To this end, we generalize prior work on confidence sets under categorical feedback (i.e., class feedback, Amani & Thrampoulidis, 2020; Zhang & Sugiyama, 2023) to our fine-tuning setting. We consider the function class $\mathcal{W}_B \doteq \{\boldsymbol{W} \in \mathbb{R}^{V \times d} \mid \|\boldsymbol{W} - \boldsymbol{W}^{\mathrm{pre}}\|_{\mathrm{F}} \leq B\}$ where $\|\cdot\|_{\mathrm{F}}$ denotes the Frobenius norm and with $B$ a constant such that $\boldsymbol{W}^\star \in \mathcal{W}_B$. Then given data $D_n$, we can refine the prior estimate $\boldsymbol{W}^{\mathrm{pre}}$ of $\boldsymbol{W}^\star$ by minimizing the regularized negative log-likelihood loss

$$\mathcal{L}^\lambda(\boldsymbol{W}; D_n) \doteq \mathcal{L}(\boldsymbol{W}; D_n) + \frac{\lambda}{2} \|\boldsymbol{W} - \boldsymbol{W}^{\mathrm{pre}}\|_{\mathrm{F}}^2 \tag{1}$$

with regularization coefficient $\lambda > 0$. We write its minimizer as $\boldsymbol{W}_n \doteq \arg\min_{\boldsymbol{W} \in \mathcal{W}_B} \mathcal{L}^\lambda(\boldsymbol{W}; D_n)$. We will further denote the ground truth probability distribution over the response to $\boldsymbol{x}$ by $\boldsymbol{s}^\star(\boldsymbol{x}) \doteq \boldsymbol{s}(\boldsymbol{f}^\star(\boldsymbol{x}))$ and our approximation after selection of $n$ samples by $\boldsymbol{s}_n(\boldsymbol{x}) \doteq \boldsymbol{s}(\boldsymbol{f}(\boldsymbol{x}; \boldsymbol{W}_n))$.

We construct confidence sets of the form $[\boldsymbol{s}_n(\boldsymbol{x}) \pm \beta_n(\delta)\sigma_n(\boldsymbol{x})]$ centered around this prediction, and show their uniform anytime validity. The width of these sets is characterized by our central quantity $\sigma_n(\boldsymbol{x})$ which we define next. We consider the inner-product kernel $k(\boldsymbol{x}, \boldsymbol{x}') \doteq \boldsymbol{\phi}(\boldsymbol{x})^\top \boldsymbol{\phi}(\boldsymbol{x}')$ and define for a set of inputs $X = \{\boldsymbol{x}_1, \ldots, \boldsymbol{x}_n\} \subseteq \mathcal{D}$:

$$\sigma_X^2(\boldsymbol{x}) \doteq k(\boldsymbol{x}, \boldsymbol{x}) - \boldsymbol{k}_X^\top(\boldsymbol{x})(\boldsymbol{K}_X + \lambda\kappa\boldsymbol{I}_n)^{-1}\boldsymbol{k}_X(\boldsymbol{x}) \tag{2}$$

where $\boldsymbol{k}_X(\boldsymbol{x}) = (k(\boldsymbol{x}_1, \boldsymbol{x}), \ldots, k(\boldsymbol{x}_n, \boldsymbol{x})) \in \mathbb{R}^n$, $\boldsymbol{K}_X \in \mathbb{R}^{n \times n}$ is the kernel matrix satisfying $(\boldsymbol{K}_X)_{i,j} = k(\boldsymbol{x}_i, \boldsymbol{x}_j)$, and $\kappa \doteq \sup_{\boldsymbol{x} \in \mathcal{X}, \boldsymbol{W} \in \mathcal{W}_B} 1/\lambda_{\min}(\boldsymbol{A}(\boldsymbol{x}; \boldsymbol{W}))$. Here, $\boldsymbol{A}(\boldsymbol{x}; \boldsymbol{W}) \in \mathbb{R}^{V \times V}$ is the matrix satisfying $(\boldsymbol{A}(\boldsymbol{x}; \boldsymbol{W}))_{i,j} \doteq s_i(\boldsymbol{x}; \boldsymbol{W})(\mathbb{1}\{i = j\} - s_j(\boldsymbol{x}; \boldsymbol{W}))$ which is the proper generalization of the derivative of the sigmoid function, standard in the analysis of binary feedback (Faury et al., 2020; Pásztor et al., 2024). We write $\sigma_n^2(\boldsymbol{x}) \doteq \sigma_{X_n}^2(\boldsymbol{x})$ where $X_n \subseteq \mathcal{D} \subseteq \mathcal{X}$ are the inputs in $D_n$. With this we are ready to state our first result, namely that for careful choice of $\beta_n(\delta)$, the confidence sets contain $\boldsymbol{s}^\star(\boldsymbol{x})$ simultaneously for all $\boldsymbol{x} \in \mathcal{X}$ and $n \geq 1$ with probability at least $1 - \delta$.

**Theorem 3.2** (Confidence Sets). *Let Assumption 3.1 hold and $\boldsymbol{W}^\star \in \mathcal{W}_B$. Let $\delta \in (0, 1)$ and set*

$$\beta_n(\delta) \doteq 2\sqrt{V(1 + 2B)}\left[B + \frac{LV^{3/2}d}{\lambda}\log\left(\frac{2}{\delta}\sqrt{1 + \frac{n}{d\lambda}}\right)\right] \in O(\log(n/\delta)) \tag{3}$$

*where $L \doteq \sup_{\boldsymbol{x} \in \mathcal{X}, \boldsymbol{W} \in \mathcal{W}_B} \lambda_{\max}(\boldsymbol{A}(\boldsymbol{x}; \boldsymbol{W}))$. Then*

$$\mathbb{P}(\forall n \geq 1, \boldsymbol{x} \in \mathcal{X} : d_{\mathrm{TV}}(\boldsymbol{s}_n(\boldsymbol{x}), \boldsymbol{s}^\star(\boldsymbol{x})) \leq \beta_n(\delta)\sigma_n(\boldsymbol{x})) \geq 1 - \delta$$

*where $d_{\mathrm{TV}}(\boldsymbol{s}, \boldsymbol{s}') \doteq \frac{1}{2}\sum_i |s_i - s_i'|$ is the total variation distance.*

We use $\sigma_n(\boldsymbol{x})$ as a proxy to the *uncertainty about the response to $\boldsymbol{x}$* after having fine-tuned on the selected data $D_n$, since it directly governs the size of the confidence sets around our current estimate of response probabilities. This uncertainty is a key quantity not just in classification: In Appendix K.5, we state analogous confidence sets for regression with the standard squared error loss, building on results by Abbasi-Yadkori (2013) and Chowdhury & Gopalan (2017).

**The Close Relationship of Regularized Loss Minimization and Test-Time Fine-Tuning**  Recall that test-time fine-tuning does not solve the regularized objective of Equation (1), but instead takes a single gradient step. So why do we expect the surrogate model $\boldsymbol{f}(\cdot; \boldsymbol{W}_n)$ be closely related to the fine-tuned $\boldsymbol{f}^{\mathrm{pre}}$? To answer this question, we contrast two alternative models:

- $\boldsymbol{W}_\lambda \doteq \arg\min_{\boldsymbol{W}} \mathcal{L}^\lambda(\boldsymbol{W})$, *(minimizer of regularized loss)*

- $\widehat{\boldsymbol{W}}_\eta \doteq \boldsymbol{W}^{\mathrm{pre}} - \eta\boldsymbol{\nabla}\mathcal{L}(\boldsymbol{W}^{\mathrm{pre}})$ with any step size $\eta > 0$, *(single gradient-step fine-tuning)*

where we keep the dataset $D$ fixed and omit the dependency on $D$. Our following proposition shows that both models are close if the loss landscape is relatively smooth and for careful choice of $\lambda \approx \frac{1}{\eta}$.

**Proposition 3.3.** *It holds that $\|\boldsymbol{W}_{1/\eta} - \widehat{\boldsymbol{W}}_\eta\|_{\mathrm{F}} \le \eta\,\|\boldsymbol{\nabla}\mathcal{L}(\boldsymbol{W}_{1/\eta}) - \boldsymbol{\nabla}\mathcal{L}(\boldsymbol{W}^{\mathrm{pre}})\|_{\mathrm{F}}.$*

Recent works have also observed $\boldsymbol{W}_{1/\eta} \approx \widehat{\boldsymbol{W}}_\eta$ empirically (Ali et al., 2019; 2020). Intuitively, with a larger step size, $\widehat{\boldsymbol{W}}_\eta$ is farther away from $\boldsymbol{W}^{\mathrm{pre}}$, and hence corresponds to the regularized estimate with less regularization. This connection between regularized loss minimization and test-time fine-tuning is closely linked to the tight connection between regularization and early stopping (Morgan & Bourlard, 1989; Yao et al., 2007; Li et al., 2020). We will use this connection in the following to derive SIFT in the context of fine-tuning.

## 4  SIFT: Efficiently Reducing Uncertainty about the Response

We introduce SIFT, an algorithm for selecting data for fine-tuning that effectively reduces the uncertainty about the response to the prompt $\boldsymbol{x}^\star \in \mathcal{X}$. Note that we can compute the uncertainty $\sigma_X(\boldsymbol{x}^\star)$ about the response to the prompt $\boldsymbol{x}^\star$ for any selected data $X \subseteq \mathcal{D}$ in closed-form, since its definition (cf. Equation (2)) depends only on the selected inputs $X$. SIFT minimizes this uncertainty about $\boldsymbol{x}^\star$:

$$\boldsymbol{x}_{n+1} \doteq \arg\min_{\boldsymbol{x}\in\mathcal{D}} \sigma^2_{X_n\cup\{\boldsymbol{x}\}}(\boldsymbol{x}^\star) = \arg\max_{\boldsymbol{x}\in\mathcal{D}} \boldsymbol{k}^\top_{X_n\cup\{\boldsymbol{x}\}}(\boldsymbol{x}^\star)(\boldsymbol{K}_{X_n\cup\{\boldsymbol{x}\}} + \lambda'\boldsymbol{I}_{n+1})^{-1}\boldsymbol{k}_{X_n\cup\{\boldsymbol{x}\}}(\boldsymbol{x}^\star).$$
$$(\mathrm{SIFT}(\lambda'))$$

SIFT selects data that minimizes a bound on the approximation error of the surrogate model, and then fine-tunes the full LLM using this data. We discuss the design choices, including the choice of embeddings, that make SIFT efficient in §4.2. In §C.1, we illustrate with an example of how SIFT balances relevance and diversity, where we also see that the free parameter $\lambda' = \lambda\kappa$ controls this trade-off. Larger $\lambda'$ emphasize relevance of selected data, while smaller $\lambda'$ emphasize diversity. Probabilistically, SIFT can be interpreted as maximizing the information gain of the selected data $X_n$ on the response to the prompt $\boldsymbol{x}^\star$ in a tractable model. We formally introduce this interpretation of SIFT in §G.

### 4.1  Uncertainty Provably Vanishes

We prove that unlike with Nearest Neighbor retrieval, the uncertainty about the response to the prompt vanishes if SIFT is used to select data for fine-tuning. We give an informal overview here, and defer the formal treatment to §C.2. Our theoretical analysis shows that test-time fine-tuning can fully reduce uncertainty only if the data space contains sufficient information to determine the correct response. If the data space does not contain all relevant information, the remaining uncertainty is quantified by the limiting uncertainty after seeing "all data in the data space infinitely often", which we call the *irreducible uncertainty* and denote by $\sigma_\infty(\boldsymbol{x}^\star)$. We provide the formal definition in §C.2, but intuitively, the irreducible uncertainty is the largest quantity satisfying $\sigma_X(\boldsymbol{x}^\star) \ge \sigma_\infty(\boldsymbol{x}^\star)$ for all $X \subseteq \mathcal{D}$. We then specialize the result of Hübotter et al. (2024) to show that the uncertainty about the response to the prompt shrinks at the rate $\widetilde{O}(1/\sqrt{n})$ until it reaches the irreducible uncertainty:

**Informal Theorem 4.1** (Convergence Guarantee). *Fix any $\lambda' > 0$ and let $\mathrm{SIFT}(\lambda')$ select $X_n$ from the data space $\mathcal{D}$. Then for all $n \ge 1$ and $\boldsymbol{x}^\star \in \mathcal{X}$,*

$$\sigma^2_n(\boldsymbol{x}^\star) - \sigma^2_\infty(\boldsymbol{x}^\star) \le \frac{O(\lambda'\log n)}{\sqrt{n}}.$$

Naturally, convergence is slower with a larger regularization parameter / smaller step size. Notably, the irreducible uncertainty depends on the data space. With a large and diverse data space, the irreducible uncertainty is typically negligible. This statistical guarantee is a key property of SIFT. As we show in Proposition K.1, Nearest Neighbor retrieval fails to satisfy a guarantee of this kind.

### 4.2  Compute-Efficient Data Selection

We have established how to select informative data for fine-tuning. Next to good statistical efficiency, good computational efficiency is key for selecting data at test-time. In the following, we describe design choices such that SIFT has negligible overhead compared to Nearest Neighbor retrieval.

**Sequence-Level Selection** In the self-supervised paradigm, each sequence of tokens $\boldsymbol{x} \in \mathcal{D}$ corresponds to a dataset of next-token predictions $x_{1:k} \mapsto x_{k+1}$. Rather than selecting individual next-token predictions from the data space of all sub-sequences $x_{1:k}$, we select full sequences $\boldsymbol{x}$ from the significantly smaller data space $\mathcal{D}$, then fine-tune for a single gradient step on each sub-sequence within $\boldsymbol{x}$. This is a common practice in prior works that use Nearest Neighbor retrieval for data selection (e.g., Xia et al., 2024; Hardt & Sun, 2024).

**Surrogate Sequence Embedders** We use a surrogate sequence embedding model to generate embeddings of the data space and prompts. We use the same embedding model as Hardt & Sun (2024) which is a large Roberta model (Liu, 2019) with 355M parameters that was fine-tuned for one pass on the Pile training set. The embedding dimension is 1024. Unlike Hardt & Sun (2024), we additionally normalize the embeddings to unit length, the reasons for which we discuss in §D.

We obtain decent performance with this surrogate model. Nevertheless, our theoretical results indicate that using embeddings extracted from the LLM to be fine-tuned could further improve the performance of SIFT. Empirical neural tangent embeddings (Wei et al., 2022a; Holzmüller et al., 2023) and influence function embeddings (Xia et al., 2024) can be implemented efficiently and offer alternative latent spaces capturing the pre-trained model. We hypothesize that the decent performance of the surrogate model is explained by the similarity of emergent latent spaces of language models that were trained on similar data.

**Efficient Implementation of SIFT** In our experiments, we pre-select 200 candidates via Nearest Neighbor retrieval with Faiss (Johnson et al., 2019) and then apply SIFT to select 50 sequences from this smaller data space. On the Pile dataset, we find that performance can be increased further by pre-selecting more candidates (cf. Figure 18 in §H) but the marginal gains diminish. The precise performance benefit of pre-selecting more candidates may differ on other datasets. We describe in §H how SIFT can be solved iteratively without computing the inverse in every iteration. When a matrix of the size of the pre-selected data space fits in GPU memory, we find that SIFT has a negligible computational overhead compared to Nearest Neighbor retrieval. We report results with an NVIDIA RTX 4090 GPU in Figure 4.[1] While our main implementation of SIFT is fast if the data space is small, it does not scale linearly with the size of the data space $K$. In §H, we show that a priority queue can be used to achieve an almost-linear runtime in $K$.

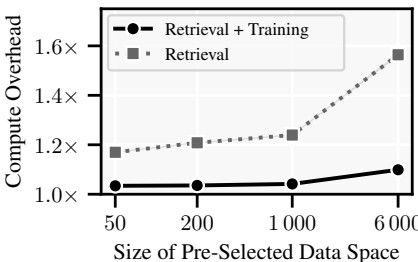

Figure 4: The (multiplicative) computational overhead of SIFT compared to Nearest Neighbor retrieval is minimal. The compute overhead with a 1k data space is less than $1.05\times$.

## 5 RESULTS

We focus on language modeling with causal language models. Following Hardt & Sun (2024), we fine-tune a pre-trained LLM for a single gradient step each on $N = 50$ selected data points in the order that they are selected, most to least relevant. We use the Pile dataset (Gao et al., 2020) for evaluation, restricting our use to data which is obtained and used in compliance with the terms of service of the data host. This version of the Pile contains a diverse set of 17 high-quality sub-datasets, ranging from Q&A to code, scientific publications, math, and more. Concretely, we use the Pile training set containing 210M sequences of total size 1.3TB as data space for data selection, and we evaluate on the Pile test set.[2] We report the *bits per byte* metric as recommended by Gao et al. (2020), which is proportional to the negative log-likelihood loss normalized by a dataset-specific constant. Error bars correspond to 90% confidence intervals computed via bootstrapping with 1'000 samples.

**Base Models and Baselines** We evaluate the GPT-2 model (Radford et al., 2019) with 124M parameters also evaluated by Hardt & Sun (2024), with the default learning rate of the `transformers` library (Wolf et al., 2020). We obtain analogous results with GPT-2-large (774M parameters) and the state-of-the-art Phi-3 (3.8B, Abdin et al., 2024).[3] With Phi-3, we use low-rank adaptation (LoRA, Hu et al., 2022), fine-tuning slightly less than 1% of the model's total parameters. We compare SIFT

---

[1] We use the client-server architecture described by Hardt & Sun (2024) with CPU-only servers.

[2] We evaluate on 1% of the test set (0.1% with Phi-3), corresponding to 1'812 sequences.

[3] We detail hyperparameter choices for larger models in §I.

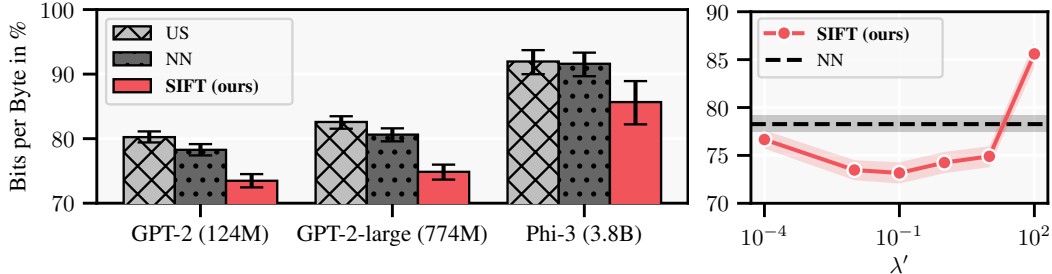

Figure 5: Bits per byte (in % relative to the base model, ↓ better) after 50 test-time iterations. **Left:** Performance gains of SIFT are consistent across models. The failure-mode of Nearest Neighbor consistently performs worse than the base model. Tables 4 and 5 in §F detail our results with GPT-2-large and Phi-3 analogously to Table 1. **Right:** Most choices of $\lambda'$ lead to comparable performance. With $\lambda' \to \infty$, SIFT($\lambda'$) repeatedly selects the nearest neighbor.

with $\lambda' = 0.01$ to Nearest Neighbor retrieval (NN) and the failure mode of Nearest Neighbor retrieval that repeatedly selects the closest neighbor. The failure mode of Nearest Neighbor retrieval (NN-F) corresponds to an extreme case of redundancy in the data space which we suspect to be a realistic scenario in larger or less curated datasets. Finally, we compare to Uncertainty Sampling (US), which is a widely used active learning strategy (Lewis, 1995; Settles, 2009) that selects the data with the highest uncertainty in the model's response by selecting according to $\boldsymbol{x}_{n+1} = \arg\max_{\boldsymbol{x} \in \mathcal{D}} \sigma_n^2(\boldsymbol{x})$. We compare to the heuristic that uses US to choose from the 200 nearest neighbors, in which case US can be understood as finding a diverse cover of this pre-selected data space (see, e.g., Holzmüller et al., 2023; Kirsch et al., 2018). In contrast, SIFT *minimizes* the uncertainty in the model's response to the prompt $\boldsymbol{x}^\star$, leading to a "denser" cover close to $\boldsymbol{x}^\star$ and a "coarser" cover further away from $\boldsymbol{x}^\star$.

**Insight 1: SIFT consistently selects better data for fine-tuning than Nearest Neighbor retrieval.**
We show in Figure 1 that SIFT outperforms NN and avoids its failure mode where the same data point is selected repeatedly. In Figure 5 (left), we show that the performance gains of SIFT are consistent across models. Table 1 compares the performance of SIFT against NN across all datasets of the Pile, using GPT-2 as base model. Overall, we find that SIFT improves performance both on datasets where NN already performs well, such as GitHub, and on datasets where NN performs poorly, such as NIH Grants. On all datasets of the Pile, SIFT performs at least as well as the strongest baseline (within margin of error), suggesting that it is a robust method for data selection. We observe the trend that relative performance gains of SIFT over Nearest Neighbor retrieval *increase with model capability*. That is, with stronger base models, informativeness of selected data appears to become more important.

|  | US | NN | NN-F | SIFT | $\Delta$ |
|---|---|---|---|---|---|
| NIH Grants | 93.1 (1.1) | 84.9 (2.1) | 91.6 (16.7) | **53.8** (8.9) | ↓31.1 |
| US Patents | 85.6 (1.5) | 80.3 (1.9) | 108.8 (6.6) | **62.9** (3.5) | ↓17.4 |
| GitHub | 45.6 (2.2) | 42.1 (2.0) | 53.2 (4.0) | **30.0** (2.2) | ↓12.1 |
| Enron Emails | **68.6** (9.8) | 64.4 (10.1) | 91.6 (20.6) | **53.1** (11.4) | ↓11.3 |
| Wikipedia | 67.5 (1.9) | **66.3** (2.0) | 121.2 (3.5) | **62.7** (2.1) | ↓3.6 |
| Common Crawl | 92.6 (0.4) | 90.4 (0.5) | 148.8 (1.5) | **87.5** (0.7) | ↓2.9 |
| PubMed Abstr. | 88.9 (0.3) | 87.2 (0.4) | 162.6 (1.3) | **84.4** (0.6) | ↓2.8 |
| ArXiv | 85.4 (1.2) | **85.0** (1.6) | 166.8 (6.4) | **82.5** (1.4) | ↓2.5 |
| PubMed Central | **81.7** (2.6) | **81.7** (2.6) | 155.6 (5.1) | **79.5** (2.6) | ↓2.2 |
| Stack Exchange | 78.6 (0.7) | 78.2 (0.7) | 141.9 (1.5) | **76.7** (0.7) | ↓1.5 |
| Hacker News | 80.4 (2.5) | **79.2** (2.8) | 133.1 (6.3) | **78.4** (2.8) | ↓0.8 |
| FreeLaw | **63.9** (4.1) | 64.1 (4.0) | 122.4 (7.1) | 64.0 (4.1) | ↑0.1 |
| DeepMind Math | 69.4 (2.1) | **69.6** (2.1) | 121.8 (3.1) | 69.7 (2.1) | ↑0.3 |
| *All* | 80.2 (0.5) | 78.3 (0.5) | 133.3 (1.2) | **73.5** (0.6) | ↓4.8 |

Table 1: Bits per byte (in % relative to the base model, ↓) after 50 test-time iterations on individual datasets of the Pile. We only include datasets with at least 10 examples in our test set. **Bold** numbers denote the best performing selected subset. Numbers in parentheses are standard errors. $\Delta$ denotes the performance gain of SIFT over the strongest baseline.

**Insight 2: SIFT is robust to the choice of $\lambda'$.** We evaluate SIFT with varying choices of $\lambda'$, and summarize the results in Figure 5 (right). We include extended results in Table 11 of §J, showing that for all evaluated $\lambda'$ between 1e−8 and 10, SIFT performs at least on-par with Nearest Neighbor retrieval on *all* datasets of the Pile, often outperforming it. This suggests that SIFT is robust to the choice of $\lambda'$. Nevertheless, there may be an advantage to adaptively tuning $\lambda'$ (e.g., via cross-validation). In particular, choosing the best $\lambda'$ for each dataset, SIFT outperforms all baselines on every dataset.

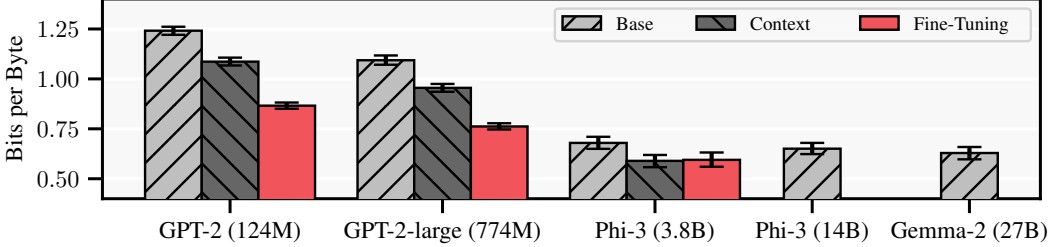

Figure 7: Bits per byte (↓ better), comparing fine-tuning and in-context learning with 50 test-time examples selected by SIFT. We find that fine-tuning systematically outperforms or performs on-par with in-context learning, even when fine-tuning only a LoRA adapter as with Phi-3. Test-time fine-tuning with Phi-3 (3.8B) surpasses the performance of the more than 3× larger Phi-3 (14B) and the 7× larger Gemma-2 (27B).

**Insight 3: SIFT selects data the "right" number of times.** Nearest Neighbor retrieval implicitly relies on non-redundancy within the data space to not select duplicate information, as illustrated in the example of Figure 3. This is almost never the case in practice, and in the extreme case of duplicate data, Nearest Neighbor selects the same data point repeatedly. SIFT does not rely on excluding previously selected data points. Instead, SIFT may select the same data point any number of times, adaptively taking more than one gradient step on it, if beneficial. To ensure that the selected data is maximally informative, SIFT takes into account the redundancy of data points explicitly. This makes SIFT robust to information duplication by design.

We illustrate this in Figure 6 where we evaluate the performance gain of SIFT over Nearest Neighbor and its failure mode. As expected, we find that on all test prompts where SIFT selects many unique points, SIFT outperforms repeatedly selecting the closest neighbor by a large margin. Interestingly, we also find that on all test prompts where SIFT selects only a single point, SIFT outperforms Nearest Neighbor by a large margin. This suggests that in some cases repeatedly taking gradient steps on the closest neighbor is beneficial, and SIFT identifies these cases.

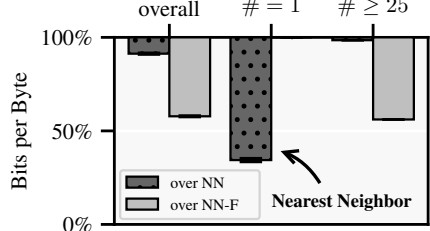

Figure 6: Bits per byte (in % relative to NN / NN-F, ↓ better) after 50 test-time iterations. Error bars correspond to standard errors. The left bars measure the performance gain over all of the Pile. The middle and right bars measure the performance gain for all prompts where SIFT selects # unique points.

**Insight 4: Test-time fine-tuning can significantly improve language modeling ability.** Our results from Figure 7 indicate that test-time fine-tuning improves the performance of the base LLM substantially, surprisingly, even with a state-of-the-art model such as Phi-3. Our Phi-3 with test-time fine-tuning and SIFT achieves 0.595 bits per byte, outperforming the previous leader in the Pile language modeling benchmark, a 30× larger model.[4] We also evaluate the recent Llama-3.2 family of models (Dubey et al., 2024), and with Llama-3.2 (3B) as base model we achieve 0.557 bits per byte, a significant improvement upon the previous state-of-the-art. We compare test-time fine-tuning to the common in-context learning, where we include as much of the data as possible into the context window of the test instance, in addition to its original context, by concatenating text in order of selection. While in-context learning tends to improve the performance of the base model, we find that fine-tuning at test-time tends to outperform or perform on-par with in-context learning. Furthermore, the compute cost of in-context learning grows quadratically with the context window size, meaning that including long texts within large context windows is expensive. Remarkably, test-time fine-tuning consistently outperforms in-context learning by more than 25% on math and coding, tasks that require more complex reasoning (§F).

**Further Insights** In §D, we discuss additional findings on active fine-tuning such as that the performance gains of SIFT over Nearest Neighbor retrieval *grow with dataset size*, and that normalizing embeddings is important for the effectiveness of data selection. In §E, we discuss additional findings on test-time fine-tuning, for example, the trend that *larger models learn faster at test-time*.

---

[4]We compare to prior work in the Pile language modeling benchmark in Table 2 of §A.

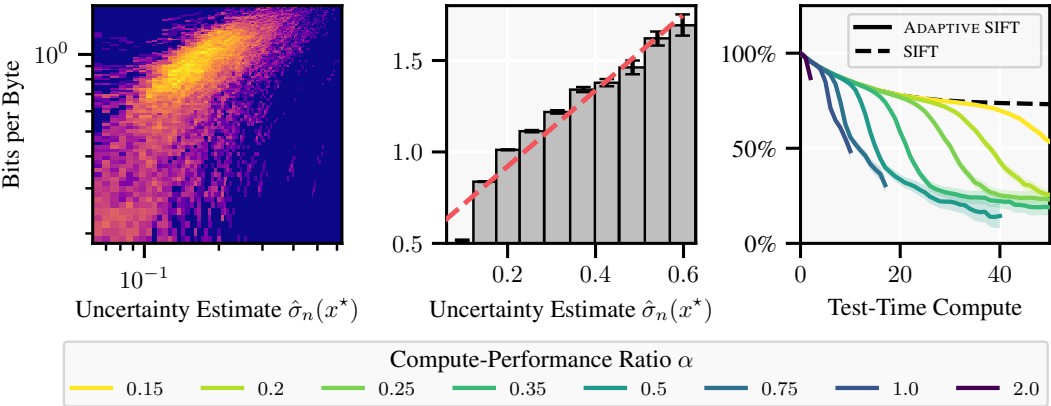

Figure 8: **Left:** We visualize the empirical density of the uncertainty estimates $\hat{\sigma}_n$ wrt. the bits per byte $\text{bpb}_n$. Brighter colors indicate higher density on a logarithmic scale. We observe a strong linear relationship between uncertainty estimates and bits per byte. **Middle:** We construct a "reliability diagram" of uncertainty estimates. Notably, since we evaluate with respect to bits per byte rather than an accuracy, canonical calibration plots are not applicable. In particular, it is well known that bits per byte do not go to zero for perfect models due to irreducible *aleatoric* uncertainty, which is not captured by our *epistemic* uncertainty estimates. Nevertheless, we observe that our epistemic uncertainty estimates are predictive of the model's performance. The red line indicates a linear fit. **Right:** We visualize the bits per byte (in % relative to the base model, ↓ better) of all prompts whose model is fine-tuned at a given iteration. We find that by adaptively stopping with respect to the known uncertainties $\sigma_n$, we can spend test-time compute proportional to realized performance gains (see also Figure 26 in §J). *Remarks:* Results are with GPT-2. In the left and middle plots, we remove the lowest and highest 0.25% of uncertainty estimates (i.e., the outliers) for better visualization. In the left plot, we additionally remove the lowest and highest 0.25% of bits per byte.

## 6 COMPUTE-PROPORTIONAL TEST-TIME FINE-TUNING

We have shown that test-time fine-tuning can improve language modeling ability and that SIFT is a robust method for data selection, outperforming Nearest Neighbor retrieval. However, a key shortcoming of previous approaches to test-time fine-tuning is that they spend a fixed amount of test-time compute, regardless of the nature of the prompt, the available data, or the model. This is not computationally scalable in many practical applications, since a fixed test-time compute budget leads to non-proportionate performance gains. For example, for the prompt "Hello" to a chatbot we would not like to spend any test-time compute, while for a more complex prompt we would like to spend more compute. In this section, we evaluate whether uncertainty estimates can be used to adaptively stop test-time fine-tuning such that the realized performance gain is proportional to the compute used.

**Insight 5: The response uncertainty can *predict* performance gain.** We find that $\sigma_n(\boldsymbol{x}^\star)$ is monotonically and linearly correlated at coefficient $\approx 0.4$ with the model error after $n$ test-time iterations, i.e., the bits per byte $\text{bpb}_n(\boldsymbol{x}^\star)$. This is remarkable because $\sigma_n$ contains information only from the surrogate embedding model, and is normalized such that $\sigma_0(\boldsymbol{x}^\star) = 1$. To determine the importance of the base model, we also evaluate the denormalized uncertainty estimate $\hat{\sigma}_n(\boldsymbol{x}^\star) \doteq \sigma_n(\boldsymbol{x}^\star) \cdot \text{bpb}_0(\boldsymbol{x}^\star)$, which unlike $\sigma_n$ cannot be evaluated at test-time. We multiply $\sigma_n$ by $\text{bpb}_0$ to ensure that the uncertainty measure is in the same units as the performance metric, correcting for the use of normalized surrogate embeddings. We find that $\hat{\sigma}_n(\boldsymbol{x}^\star)$ is strongly correlated at coefficient $\gtrsim 0.5$ with the bits per byte. We summarize correlations in Table 12 of §J and visualize the predictive capability of $\hat{\sigma}_n$ in Figure 8 (left) and Figure 8 (middle). Our findings indicate that approximations of the base model's uncertainty, before test-time fine-tuning, can be beneficial. In future work, we intend to determine whether generating embeddings from the base model can provide such scale-correction.

Recall that SIFT minimizes the response uncertainty $\sigma_n$ to the given prompt. The predictive ability of uncertainty estimates provides an intuitive explanation for the effectiveness of SIFT.

*Compute-Proportional Performance Gains:* **Early stopping at the "right" time.** Motivated by the predictive power of uncertainty estimates, we evaluate whether they can be used to *adaptively*

*stop* test-time fine-tuning such that the realized performance gain is proportional to the compute used. In the following, we propose a such a stopping criterion for SIFT. Using the approximation of the error via uncertainty estimates discussed above and that $\sigma_0(\boldsymbol{x}^\star) = 1$:

$$\text{performance gain} = \frac{\text{bpb}_0(\boldsymbol{x}^\star)}{\text{bpb}_n(\boldsymbol{x}^\star)} \approx \frac{\sigma_0(\boldsymbol{x}^\star)}{\sigma_n(\boldsymbol{x}^\star)} = \frac{1}{\sigma_n(\boldsymbol{x}^\star)}. \tag{4}$$

We would like to stop fine-tuning when further test-time compute does not yield proportional performance gain, i.e., when "performance gain $< \alpha \cdot n$" with $n$ approximating the compute of $n$ iterations and $\alpha$ a constant comparing the units of compute and performance. Plugging in our above approximation of the performance gain, we propose to stop test-time fine-tuning *before* iteration $n$ if

$$\sigma_n(\boldsymbol{x}^\star) > (\alpha n)^{-1}. \qquad\qquad \text{(ADAPTIVE SIFT)}$$

Intuitively, this stops fine-tuning the LLM when its progress in crafting a better response stalls. For complex prompts that benefit from fine-tuning, ADAPTIVE SIFT spends more test-time compute, whereas for prompts where the model is already strong or where the data space is not informative, ADAPTIVE SIFT spends less test-time compute. Figure 8 (right) shows that the performance gains of this approach are proportional to the compute used.

**Towards Scaling Laws of Test-Time Fine-Tuning**  Interestingly, our results bear resemblance to scaling laws of LLM pre-training (Kaplan et al., 2020; Henighan et al., 2020; Hoffmann et al., 2022). These scaling laws express the performance of a model as a function of the compute used for pre-training (e.g., the number of parameters or training tokens). Such scaling laws are crucial for determining how to optimally spend a fixed amount of compute. Recently, scaling laws for "test-time inference" have gained attention, where test-time compute is usually spent on search (e.g., beam search) with a variable number of forward passes of a few-shot prompted base LLM (Brown et al., 2024; Snell et al., 2025). Our results suggest that similar scaling laws exist for test-time fine-tuning, expressing the performance of a model as a function of the compute used for fine-tuning at test-time. Such scaling laws can be an important tool to determine how to spend test-time compute. There are many open questions in this direction, which we do not address in this work. For example, how does model size affect the scaling laws of test-time fine-tuning? Or, can a model be fine-tuned at test-time to build reasoning chains? Based on our results and previous evaluations of fine-tuning and in-context learning (e.g., Hu et al., 2022; Mosbach et al., 2023; Hardt & Sun, 2024), we conjecture that test-time fine-tuning may lead to a more efficient use of compute than repeatedly prompting a base LLM. We believe that these open questions are exciting directions for future work.

## 7  DISCUSSION AND FUTURE WORK

We propose a data selection algorithm, SIFT, unifying ideas from retrieval and active learning. SIFT estimates the uncertainty about the response to a given prompt after having been fine-tuned on some data (§3), and then selects the data that minimizes this uncertainty (§4). This addresses the limitations of Nearest Neighbor retrieval (§2). SIFT can be seen as a generalization of Nearest Neighbor retrieval from a search method to a learning method, which ensures explicitly that the retrieved data is maximally informative. We show on the Pile dataset that SIFT consistently outperforms Nearest Neighbor retrieval in prompt-specific fine-tuning at test-time and that this kind of local learning can be more effective than locally learning from examples in-context (§5). Finally, we observe that our uncertainty estimates can predict the performance gain of test-time fine-tuning, and use this to develop an adaptive algorithm which achieves compute-proportional performance gains (§6).

Test-time fine-tuning addresses a fundamental limitation of in-context learning, namely that in-context learning is typically limited to a fixed and finite context window. In contrast, test-time fine-tuning allows the LLM to dynamically and effectively access a potentially unbounded non-parametric memory. By improving the effectiveness of test-time fine-tuning, this work opens up several exciting directions for future research. Test-time fine-tuning may be used to ground the model on a trusted dataset, mitigate biases against under-represented groups in the training data, or to dynamically include private data depending on user privileges. Particularly interesting would be a broad evaluation on non-perplexity tasks such as code generation or in the life sciences with large-scale medical or protein data. Unlike few-shot in-context learning which is limited in scope to autoregressive models, test-time fine-tuning and SIFT may be extended to other model classes such as diffusion models. Furthermore, SIFT may be used effectively in other settings that require automatic data selection, such as targeted instruction tuning during post-training of LLMs. Finally, our results suggest scaling laws for test-time fine-tuning and we outline several exciting open questions (§6).

## CONTRIBUTIONS

JH conceived and led the project, being involved in all its components and leading the theory, implementation of the SIFT algorithm, design of experiments, and writing. SB set up and ran the first experiments validating the approach, and contributed to running the final ablation studies. IH ran additional experiments, especially those with larger models, and optimized the code. AK advised.

## ACKNOWLEDGEMENTS

We would like to thank Armin Lederer, Vignesh Ram Somnath, Bhavya Sukhija, Scott Sussex, and Lenart Treven for feedback on early versions of the paper. This project was supported in part by the European Research Council (ERC) under the European Union's Horizon 2020 research and Innovation Program Grant agreement no. 815943, and the Swiss National Science Foundation under NCCR Automation, grant agreement 51NF40 180545. Ido Hakimi was supported by an ETH AI Center Postdoctoral fellowship.

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

# APPENDICES

## CONTENTS

# A    COMPARISON TO THE STATE-OF-THE-ART ON THE PILE LANGUAGE MODELING BENCHMARK

Table 2 summarizes the state-of-the-art in the Pile language modeling benchmark.

| Model | Bits per Byte | Bits per Byte (without Wikipedia) |
|---|---|---|
| Jurassic-1 (178B, Lieber et al., 2021) | n/a | 0.601* |
| GLM (130B, Zeng et al., 2022) | n/a | 0.622* |
| GPT-2 (124M, Radford et al., 2019) | 1.241 | |
| GPT-2 (774M, Radford et al., 2019) | 1.093 | |
| Llama-3.2-Instruct (1B, Dubey et al., 2024) | 0.807 | |
| Llama-3.2-Instruct (3B, Dubey et al., 2024) | 0.737 | |
| Gemma-2 (2B, Team et al., 2024) | 0.721 | |
| Llama-3.2 (1B, Dubey et al., 2024) | 0.697 | 0.684 |
| Phi-3.5 (3.8B, Abdin et al., 2024) | 0.690 | |
| Phi-3 (3.8B, Abdin et al., 2024) | 0.679 | 0.678 |
| Phi-3 (7B, Abdin et al., 2024) | 0.678 | |
| Gemma-2 (9B, Team et al., 2024) | 0.670 | |
| GPT-3 (175B, Brown et al., 2020) | 0.666* | |
| Phi-3.5-MoE (16×3.8B, Abdin et al., 2024) | 0.656 | |
| Phi-3 (14B, Abdin et al., 2024) | 0.651 | |
| Llama-3.2 (3B, Dubey et al., 2024) | 0.640 | 0.627 |
| Gemma-2 (27B, Team et al., 2024) | 0.629 | |
| *Test-Time FT with* SIFT + GPT-2 (124M) | 0.862 | |
| *Test-Time FT with* SIFT + GPT-2 (774M) | 0.762 | |
| *Test-Time FT with* SIFT + Llama-3.2 (1B) | 0.606 | 0.607 |
| *Test-Time FT with* SIFT + Phi-3 (3.8B) | 0.595 | 0.599 |
| *Test-Time FT with* SIFT + Llama-3.2 (3B) | **0.557** | **0.559** |

Table 2: Evaluation of state-of-the-art models on the Pile language modeling benchmark, without copyrighted datasets. (*): Results with GPT-3 are from Gao et al. (2020); results with Jurassic-1 and GLM are from Zeng et al. (2022) and do not report on the Wikipedia dataset. For a complete comparison, we also evaluate our Phi-3 and Llama-3.2 with test-time fine-tuning when excluding the Wikipedia dataset. **Bold** numbers denote the best performing model. Underlined numbers denote a model that is better than the previous state-of-the-art.

Due to our dataset being restricted to the non-copyrighted part of the Pile, the data distribution changes slightly. To account for this, we take the reported results of prior work and exclude the datasets that have copyright restrictions from the evaluation. Notably, some prior reported results of state-of-the-art models miss evaluation of the Wikipedia dataset, which we therefore also exclude for a direct comparison. To the best of our knowledge, our results with test-time fine-tuning and SIFT achieve a new state-of-the-art on the Pile benchmark.

## B  EXTENDED RELATED WORK

### B.1  LEARNING AT TEST-TIME

The subject of learning at test-time has a rich history in statistics and machine learning. By "learning at test-time" we refer to models that are constructed specifically for a given test instance, differing from the model used for other test instances. The following discussion provides a brief overview with emphasis on the most recent developments.

$k$-**Nearest Neighbors and Kernel Regression (since 1950s)**  One of the most basic forms of learning at test-time was developed by Fix & Hodges Jr. (1951) and Cover & Hart (1967). Given the supervised data $\mathcal{D} \subseteq \mathcal{X} \times \mathcal{Y}$ with input domain $\mathcal{X} \subseteq \mathbb{R}^d$ and labels $\mathcal{Y} = \{0, \ldots, K\}$, the $k$-NN algorithm predicts the label of a test instance $x^\star \in \mathcal{X}$ by taking the majority vote of the $k$ nearest neighbors of $x^\star$ in $\mathcal{D}$ according to some distance metric on $\mathcal{X}$ such as Euclidean distance. In the case of regression, $\mathcal{Y} = \mathbb{R}$ and the prediction is the average of the labels of the $k$ nearest neighbors. Kernel regression extended upon this idea in the 1960s by weighting neighbors according to their distance to the test instance (Nadaraya, 1964; Watson, 1964). This is a simple and often effective method if the inputs are well-structured and low-dimensional, e.g., if $\mathcal{X}$ is a learned low-dimensional manifold (Geirhos et al., 2024). When $K$ is large, as for example when $\mathcal{Y}$ is the set of all tokens in a language modeling task, naive application of $k$-NNs is difficult, nevertheless they have been shown to be effective when mixed with parametric language models (Khandelwal et al., 2020).

**Local Learning (since 1970s)**  Local learning is the idea of using data "relevant" to the test instance $x^\star$ to train a parametric model. Formally, given a test instance $x^\star$, conventually a model $f$ is used to predict $f(x^\star)$ where $f$ is trained to minimize the average loss over the training data. Instead, local learning trains a model $f_{x^\star}$ specifically for $x^\star$ and predicts $f_{x^\star}(x^\star)$. Original works train a linear model by weighting data according to their proximity to $x^\star$ (Cleveland, 1979; Cleveland & Devlin, 1988; Atkeson et al., 1997). Here, each test instance trains a model from scratch since the optimal solution of linear regression is independent of initialization. This perspective has regained interest recently in the context of neural networks, with Sun et al. (2020) naming it *"test-time training"*.

**Transductive Learning (since 1980s)**  Vladimir Vapnik developed the general principle of *transduction* which he states in Vapnik (2013) as follows:

> Vladimir Vapnik: *"When solving a problem of interest, do not solve a more general problem as an intermediate step. Try to get the answer that you really need but not a more general one."*

This is perhaps the most general principle behind learning at test-time, and directly opposed to the principle of *induction* — extracting the most general rules from data — which has arguably dominated machine learning research over the last decades. In a way, local learning is pushing the principle of transduction to the opposite extreme: Each test instance defines its own learning problem, with the test instance alone being the target of prediction.

**Local Fine-Tuning (since 1990s)**  Bottou & Vapnik (1992) were the first to use local learning in conjunction with a *pre-trained* parametric model. They train (i.e., "fine-tune") the last layer of a convolutional neural network for handwritten digit classification based on the nearest neighbors to the test instance in pixel space. Very recently, Hardt & Sun (2024) applied the same idea to language models, showing that local fine-tuning can significantly improve the performance of large language models on standard benchmarks. Previously, this idea has also been evaluated by Li et al. (2018) and Basu et al. (2023). *"Test-time fine-tuning"* (as well as "active inference") has frequently been used to refer to this approach of locally fine-tuning a pre-trained model. Within the last few years, test-time fine-tuning has regained substantial interest in the context of self-supervised learning, where the pre-trained model is fine-tuned on the *test instance itself*. Notable applications of this approach are in vision (Jain & Learned-Miller, 2011; Shocher et al., 2018; Luo et al., 2020; Sun et al., 2020; Wang et al., 2021b) and in language modeling (Krause et al., 2018; 2019), where it is called *dynamic evaluation*. As one would also naively expect, test-time fine-tuning yields the largest improvements when the prompt is not (well-) represented in the pre-training data, e.g., due to a distribution shift (Gandelsman et al., 2021; Hardt & Sun, 2024). Notably, test-time fine-tuning is the central component of the state-of-the-art approaches to the ARC challenge (Chollet, 2019; Cole & Osman, 2023), a non-saturated benchmark which is intended to test reasoning capabilities based on "core knowledge" rather than mere memorization.

**(Few-Shot) In-Context Learning (since 2020s)**   Very recently, with the advent of large language models (LLMs), learning at test-time has regained interest. Brown et al. (2020) showed that GPT-3 can *learn in-context* from input-label pairs that are appended to the prompt, an emergent phenomenon of LLMs that has been widely studied since (Von Oswald et al., 2023; Kossen et al., 2024; Bhattamishra et al., 2024). In contrast to standard in-weights learning, in-context learning requires no parameter updates. Interestingly, in-context learning adopts the same paradigm as local learning wherein a model is adapted specifically for the test instance $x^\star$, here by skewing the autoregressive distribution towards the data included in the prompt. This is often combined with the automatic sourcing of nearest neighbors to $x^\star$ in an external dataset, which is known as *"retrieval augmented generation"* (RAG, Lewis et al., 2019; Borgeaud et al., 2022), and is akin to the other methods of test-time learning discussed above. A crucial difference between test-time fine-tuning and in-context learning appears to be that learning from context works by *changing the test instance* (Bhargava et al., 2023) whereas in-weights learning works by *changing the model*. With small datasets, in-context learning is therefore often more computationally efficient than test-time fine-tuning, however this ceases to be the case when the dataset grows since the complexity of transformers grows quadratically in the number of context tokens whereas the complexity of test-time fine-tuning grows linearly.

### B.2   DATA SELECTION

Clearly, the choice of data to learn from at test-time is crucial for predictive performance. Selecting uninformative data can increase inference time or even degrade performance (see, e.g., Kolossov et al., 2024). Today, datasets for fine-tuning are often hand-designed, however, this is not possible in a test-time setting. Automatic data selection has a rich history in machine learning, studied extensively in *search*, *experimental design* (Chaloner & Verdinelli, 1995), and *active learning* (Settles, 2009). The following attempts to give a brief overview of the most recent developments.

**(Document) Retrieval (since 1970s)**   Retrieval methods aim to search a dataset $\mathcal{D}$ for the most relevant data to a given query/prompt. The most classical methods such as TF-IDF (Sparck Jones, 1972) and BM25 (Robertson et al., 2009) are based on keyword matching, and were developed alongside the first search engines. Due to their reliance on "bags of words", i.e., sets of one-hot-encoded word vectors, they are known as *sparse retrievers*. An alternative idea is to select the data $x$ that maximizes the likelihood of the query $x^\star$ given the data, i.e., $\arg\max_{x \in \mathcal{D}} p(x^\star \mid x)$, known as *query likelihood retrievers* (Ponte & Croft, 1998; Wang et al., 2023). Here, the conditional probability can be a non-parametric term frequency or a parametric language model. More recently, due to significant advances in representation learning (Devlin et al., 2018; Reimers & Gurevych, 2019), dense retrievers have become popular (e.g., Lewis et al., 2019; Karpukhin et al., 2020; Borgeaud et al., 2022). A *dense retriever* embeds dataset and query into a metric vector space, and retrieves the nearest neighbors to the query. Standard vector-based search methods use cosine similarity or (equivalently[5]) Euclidean distance. Recent advances in algorithms and implementation mean that (approximate) nearest neighbor retrieval can be performed efficiently with databases of billions or even trillions of tokens (e.g., Johnson et al., 2019; Aumüller et al., 2020). The most common metric is cosine distance, which coincides with Euclidean distance when vectors are normalized to unit length. Nearest neighbor retrieval has been the de-facto standard for data selection in RAG and local learning.[6]

**Influence Functions (since 1970s)**   Influence functions measure the change in a model's prediction when a single data point is removed from the training data. First proposed by Cook (1977) for linear regression, they have since been used extensively to *interpret* predictions (Koh & Liang, 2017; Pruthi et al., 2019). Very recently, Xia et al. (2024) applied influence functions to select data that leads to the largest (approximate) reduction in test-loss. Concretely, using a first-order Taylor approximation of the loss $\ell$ and if the model at time $t$ is updated via stochastic gradient descent with step size $\eta_t$ on data $x$, the loss reduction can be approximated as

$$\ell(x^\star; \theta_{t+1}) - \ell(x^\star; \theta_t) \approx -\eta_t \langle \nabla_\theta \ell(x; \theta_t), \nabla_\theta \ell(x^\star; \theta_t) \rangle.$$

That is, the data $x$ whose loss gradient is most aligned with the loss gradient of the test instance $x^\star$, can be expected to lead to the largest loss reduction.[7] Note that this simply leads to nearest neighbor

---

[5]Here we assume that vectors are normalized to unit length, cf. Appendix K.2.

[6]There is substantial literature that investigates selection of "informative" data for RAG (e.g., Ye et al., 2023).

[7]Xia et al. (2024) normalize embeddings before computing the inner product (thus, maximizing cosine similarity) to account for varying gradient norms depending on sequence lengths.

retrieval in an embedding space informed by the model at time $t$. A major limitation of using influence functions for data selection is that they implicitly assume that the influence of selected data adds linearly (i.e., two equally scored data points are expected to doubly improve the model performance, Xu & Kazantsev, 2019, Section 3.2). This assumption does quite obviously not hold in practice as seen, e.g., by simply duplicating data. The same limitation applies to the related approach of *datamodels* (Ilyas et al., 2022). A recent line of work aims to address this limitation by designing simulators that can be probed with datasets to estimate their effect on a prediction requiring less compute than training the full model (Guu et al., 2023), yet, this does not address the data selection problem as the space of possible datasets is exponentially large.

**Coverage & *Inductive* Active Learning** Next we discuss an orthogonal line of work, which takes into account the interaction between selected data, but not the interaction of that data with respect to a test instance. Roughly speaking classical active learning studies how to most effectively select data from a domain $\mathcal{X}$ for learning a model over this domain $\mathcal{X}$. Intuitively, this task can be thought of as selecting a subset $X \subseteq \mathcal{X}$ of fixed size that captures the most "information" about the target function $f$. As such, this task is of an *inductive* nature: we aim to extract general rules from the data that can be applied to unseen data later, without concrete specification of the unseen data. Approaches to (inductive) active learning are broadly aiming to select *diverse* data that covers the data manifold in $\mathcal{X}$ well. Methods include those that maximize the mutual distances between selected data (e.g., CORESET (Sener & Savarese, 2017), BADGE (Ash et al., 2020), and PROBCOVER (Yehuda et al., 2021)) with respect to a latent distance metric and those "uncertainty sampling" methods that select data that the model is most uncertain about (e.g., *D-optimal design* (Wynn, 1970) and BATCHBALD (Kirsch et al., 2018)).[8] Both families of methods can be seen as determining some decent covering of the data manifold in $\mathcal{X}$. In a probabilistic sense, uncertainty sampling can be seen to minimize the "posterior predictive entropy" in expectation over the observed data. Approaches to inductive active learning have frequently been applied to pre-training models, with image classification as the canonical application (e.g, Holzmüller et al., 2023).

## B.3 SIFT UNIFIES WORK ON RETRIEVAL AND WORK ON COVERAGE

Retrieval and inductive active learning fall on to two extreme ends of a spectrum: Retrieval methods search for relevant data without ensuring that data is non-redundant. As such, naive application of search methods is insufficient for a learning task since those generally do not take "distinctiveness" into account (cf. Section 2.1). In contrast, active learning methods select non-redundant data without ensuring that data is relevant. Like SIFT, many active learning methods are based on some measure of "uncertainty", however how this measure is utilized for data selection differs fundamentally in SIFT:

***Transductive* Active Learning: Unifying retrieval & coverage** Transductive active learning is motivated from the central observation that learning and prediction requires synthesizing information that is both relevant and non-redundant. Transductive active learning (Hübotter et al., 2024) bridges this gap by selecting data that is both relevant and non-redundant. In this work, we propose SIFT, an approach to test-time transductive active learning (i.e., transductive active learning with a single prediction target), which extends previously proposed algorithms (MacKay, 1992; Seo et al., 2000; Yu et al., 2006; Hübotter et al., 2024). Similar algorithmic ideas have recently been evaluated empirically in a variety of other settings (Kothawade et al., 2020; Wang et al., 2021a; Kothawade et al., 2022; Bickford Smith et al., 2023) such as Bayesian optimization (Hübotter et al., 2024), multi-task reinforcement learning (Bagatella et al., 2024), and the amortized fine-tuning of neural networks (Hübotter et al., 2024). SIFT aims to select data that is both relevant and non-redundant with respect to the already seen data, whereby the hyperparameter $\lambda'$ controls the trade-off between relevance and redundancy. Hübotter et al. (2024) introduce extensions of SIFT to more than one prediction target, i.e., amortizing learning across multiple prompts. They show that if the prediction targets include *all of* $\mathcal{X}$, then the method reduces to a form of *inductive active learning*.

---

[8]Section 5.2 of Holzmüller et al. (2023) provides a comprehensive overview.

## C    FURTHER DETAILS ON SIFT

### C.1    HOW SIFT BALANCES RELEVANCE AND DIVERSITY

Let us look more closely at the points selected by SIFT. We will assume here for ease of notation that embeddings have unit length.[9] The first point selected by SIFT has the largest (absolute) cosine similarity to the prompt within the latent space:

$$\boldsymbol{x}_1 = \arg\min_{\boldsymbol{x}\in\mathcal{D}} \sigma^2_{\{\boldsymbol{x}\}}(\boldsymbol{x}^\star) = \arg\max_{\boldsymbol{x}\in\mathcal{D}} \frac{(\boldsymbol{\phi}(\boldsymbol{x}^\star)^\top\boldsymbol{\phi}(\boldsymbol{x}))^2}{1+\lambda'} = \arg\max_{\boldsymbol{x}\in\mathcal{D}} \Big( \underbrace{\measuredangle_{\boldsymbol{\phi}}(\boldsymbol{x}^\star,\boldsymbol{x})}_{\text{cosine similarity of }\boldsymbol{\phi}(\boldsymbol{x}^\star),\,\boldsymbol{\phi}(\boldsymbol{x})} \Big)^2. \quad \textbf{(1st point)}$$

This recovers the standard approach of Nearest Neighbor retrieval with respect to cosine similarity, provided cosine similarities are non-negative. However, we show next that selecting more than one point, SIFT not only considers the relevance with respect to the prompt $\boldsymbol{x}^\star$, but also the redundancy with respect to the already seen data $\boldsymbol{x}_1$.

$$\boldsymbol{x}_2 = \arg\min_{\boldsymbol{x}\in\mathcal{D}} \sigma^2_{\{\boldsymbol{x}_1,\boldsymbol{x}\}}(\boldsymbol{x}^\star) = \arg\max_{\boldsymbol{x}\in\mathcal{D}} \begin{bmatrix} \measuredangle_{\boldsymbol{\phi}}(\boldsymbol{x}^\star,\boldsymbol{x}_1) \\ \measuredangle_{\boldsymbol{\phi}}(\boldsymbol{x}^\star,\boldsymbol{x}) \end{bmatrix}^\top \begin{bmatrix} 1+\lambda' & \measuredangle_{\boldsymbol{\phi}}(\boldsymbol{x}_1,\boldsymbol{x}) \\ \measuredangle_{\boldsymbol{\phi}}(\boldsymbol{x}_1,\boldsymbol{x}) & 1+\lambda' \end{bmatrix}^{-1} \begin{bmatrix} \measuredangle_{\boldsymbol{\phi}}(\boldsymbol{x}^\star,\boldsymbol{x}_1) \\ \measuredangle_{\boldsymbol{\phi}}(\boldsymbol{x}^\star,\boldsymbol{x}) \end{bmatrix}.$$
$$\textbf{(2nd point)}$$

To illustrate how SIFT balances relevance and diversity, we compare the value of observing $\boldsymbol{x}_1$ twice to observing a different $\boldsymbol{x}$ with cosine similarity $\measuredangle_{\boldsymbol{\phi}}(\boldsymbol{x}_1,\boldsymbol{x}) = 0$. We show in Appendix K.4 that SIFT($\lambda'$) prefers $\boldsymbol{x}$ over $\boldsymbol{x}_1$ for selecting $\boldsymbol{x}_2$ *if and only if*

$$\measuredangle_{\boldsymbol{\phi}}(\boldsymbol{x}^\star,\boldsymbol{x})^2 > \frac{\lambda'}{2+\lambda'}\measuredangle_{\boldsymbol{\phi}}(\boldsymbol{x}^\star,\boldsymbol{x}_1)^2$$

The hyperparameter $\lambda'$ controls the trade-off between relevance and diversity: if $\lambda' = 1$ then even if $\boldsymbol{x}$ has one third the relevance of $\boldsymbol{x}_1$, it is still preferred. As $\lambda' \to \infty$, SIFT($\lambda'$) performs retrieval by repeatedly selecting the same point; and as $\lambda' \to 0$, SIFT($\lambda'$) aims only to select the most diverse points. We observe the same relationship empirically on the Pile dataset (cf. Figure 9 (left)). Table 3 summarizes the effect of the regularization parameter $\lambda$ and its interpretations.

| Parameter | Relation | Div. |
|---|---|---|
| regularization $\lambda$ | $\lambda$ | ↓ |
| step size $\eta$ | $1/\eta$ | ↑ |
| noise $\rho$ (cf. §G) | $\rho^2$ | ↓ |

Table 3: The effect of $\lambda$ and its other interpretations on diversity of selected data (as the parameter is increased).

### C.2    THE UNCERTAINTY OF SIFT PROVABLY VANISHES

We now formally prove that unlike with Nearest Neighbor retrieval, the uncertainty $\sigma^2_n(\boldsymbol{x}^\star)$ about the response to the prompt vanishes if SIFT is used to select data for fine-tuning. As discussed in §4.1, this requires that the data space contains sufficient information to determine the correct response. In general, there might be an irreducible error remaining. We will denote a basis of the embeddings $\{\boldsymbol{\phi}(\boldsymbol{x}) : \boldsymbol{x} \in \mathcal{D}\}$ within the data space $\mathcal{D}$ by $\boldsymbol{\Phi} \in \mathbb{R}^{m\times d}$ with size $m$ and dimension $d$, and we denote by $\boldsymbol{\Pi_\Phi}$ its orthogonal projection onto the orthogonal complement of the span of $\boldsymbol{\Phi}$. Hübotter et al. (2024) show that for all $X \subseteq \mathcal{D}$,

$$\sigma^2_X(\boldsymbol{x}^\star) \geq \|\boldsymbol{\phi}(\boldsymbol{x}^\star)\|^2_{\boldsymbol{\Pi_\Phi}} \tag{5}$$

where $\|\boldsymbol{v}\|_{\boldsymbol{A}} = \sqrt{\boldsymbol{v}^\top\boldsymbol{A}\boldsymbol{v}}$ denotes the Mahalanobis distance. We call $\sigma^2_\infty(\boldsymbol{x}^\star) \doteq \|\boldsymbol{\phi}(\boldsymbol{x}^\star)\|^2_{\boldsymbol{\Pi_\Phi}}$ the *irreducible uncertainty* about $\boldsymbol{x}^\star$. It can be seen that $\sigma^2_\infty(\boldsymbol{x}^\|) = 0$ for all $\boldsymbol{x}^\| \in \mathcal{X}$ with $\boldsymbol{\phi}(\boldsymbol{x}^\|) \in \text{span } \boldsymbol{\Phi}$. That is, the irreducible uncertainty is zero for points in the span of the data space. In contrast, for points $\boldsymbol{x}^\perp$ with $\boldsymbol{\phi}(\boldsymbol{x}^\perp) \in (\text{span } \boldsymbol{\Phi})^\perp$, the irreducible uncertainty equals the initial uncertainty: $\sigma^2_\infty(\boldsymbol{x}^\perp) = \sigma^2_0(\boldsymbol{x}^\perp)$. The irreducible uncertainty of any prompt $\boldsymbol{x}^\star$ can be computed by simple decomposition of $\boldsymbol{\phi}(\boldsymbol{x}^\star)$ into parallel and orthogonal components. Hence, if the data space is large and includes all relevant information to answer the prompt, the irreducible uncertainty is negligible.

We will denote the *uncertainty reduction* about the prompt $\boldsymbol{x}^\star$ achieved by fine-tuning on $X$ by $\psi_{\boldsymbol{x}^\star}(X) \doteq \sigma^2_0(\boldsymbol{x}^\star) - \sigma^2_X(\boldsymbol{x}^\star)$ and note that SIFT selects $\boldsymbol{x}_{n+1} = \arg\max_{\boldsymbol{x}\in\mathcal{D}} \psi_{\boldsymbol{x}^\star}(X_n \cup \{\boldsymbol{x}\})$. Stating the convergence guarantee of SIFT requires one straightforward assumption.

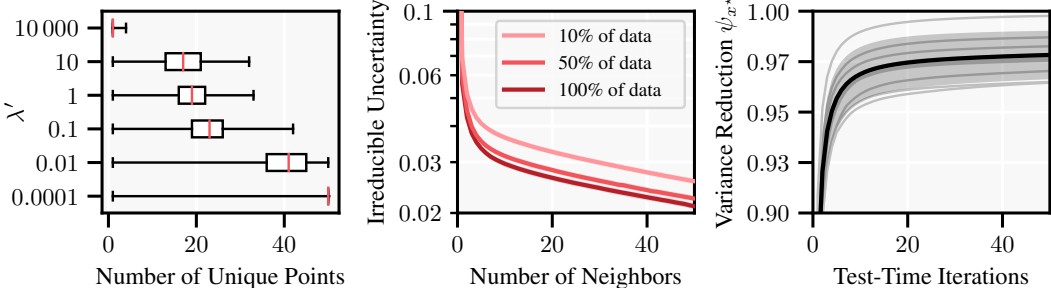

Figure 9: **Left:** The parameter $\lambda'$ controls the trade-off between relevance and diversity of the selected data. As $\lambda' \to \infty$, SIFT selects the same point repeatedly whereas as $\lambda' \to 0$, SIFT selects a diverse set of points. **Middle:** The irreducible uncertainty of test prompts from the Pile given neighbors selected from fractions of the Pile training dataset in the data space. The irreducible uncertainty captures how much information is available, and decays quickly. **Right:** We empirically observe that $\psi_{\boldsymbol{x}^\star}$ is monotone submodular, i.e., its "marginal gains" decrease as the number of iterations increases. The shaded region denotes the standard deviation, gray lines are from 10 randomly selected prompts.

**Assumption C.1.** The uncertainty reduction $\psi_{\boldsymbol{x}^\star}(X)$ is submodular.

Intuitively, Assumption C.1 states that the marginal uncertainty reduction achieved by adding a point to the selected data (i.e., the 'marginal gain') decreases as the size of the selected data increases, which is a common assumption in prior work.[10] Formally Assumption C.1 is satisfied if, for all $\boldsymbol{x} \in \mathcal{D}$ and $X' \subseteq X \subseteq \mathcal{D}$,

$$\Delta_{\boldsymbol{x}^\star}(\boldsymbol{x} \mid X') \geq \Delta_{\boldsymbol{x}^\star}(\boldsymbol{x} \mid X) \tag{6}$$

where $\Delta_{\boldsymbol{x}^\star}(\boldsymbol{x} \mid X) \doteq \psi_{\boldsymbol{x}^\star}(X \cup \{\boldsymbol{x}\}) - \psi_{\boldsymbol{x}^\star}(X)$ is the *marginal uncertainty reduction* of $\boldsymbol{x}$ given $X$.

Though theoretically this assumption may be violated by some instances (Hübotter et al., 2024, Example C.8), we observe that it is satisfied in practice (cf. Figure 9 (right)). Under this assumption, $\psi_{\boldsymbol{x}^\star}(X_n) \geq (1 - 1/e) \max_{X \subseteq \mathcal{D}, |X| \leq n} \psi_{\boldsymbol{x}^\star}(X)$ due to the seminal result on monotone submodular function maximization of Nemhauser et al. (1978). That is, the iterative scheme of SIFT achieves a constant factor approximation of the optimal uncertainty reduction. Moreover, recent work on transductive active learning of Hübotter et al. (2024) which we restate here shows that the uncertainty of SIFT converges to the irreducible uncertainty. We assume w.l.o.g. that $\|\boldsymbol{\phi}(\boldsymbol{x})\|_2^2 \leq 1$ for all $\boldsymbol{x} \in \mathcal{X}$.

**Theorem C.2** (Convergence Guarantee, formalization of Informal Theorem 4.1). *Let Assumption C.1 hold and $X_n$ be selected by* SIFT($\lambda'$) *from the data space $\mathcal{D}$. Then for all $n \geq 1$ and $\boldsymbol{x}^\star \in \mathcal{X}$,*

$$\sigma_n^2(\boldsymbol{x}^\star) \leq \sigma_\infty^2(\boldsymbol{x}^\star) + \frac{d(1 + 2d\lambda'\lambda_{\min}^{-1})\log(1 + \frac{\hat{\lambda}_n}{\lambda'})}{\sqrt{n}}$$

*where $\lambda_{\min}$ is the smallest eigenvalue of $\boldsymbol{\Phi}\boldsymbol{\Phi}^\top$ with $\boldsymbol{\Phi} \in \mathbb{R}^{m \times d}$ a basis of $\{\boldsymbol{\phi}(\boldsymbol{x}) : \boldsymbol{x} \in \mathcal{D}\}$, and where $\hat{\lambda}_n \leq O(n)$ is the largest eigenvalue of $\boldsymbol{\Phi}_n\boldsymbol{\Phi}_n^\top$.*

*Proof.* Theorem C.2 follows from Theorem 3.2 of Hübotter et al. (2024) noting that

- The SIFT objective is a special case of VTL (Variance-based Transductive Active Learning) with "target space" $\mathcal{A} = \{\boldsymbol{x}^\star\}$.

- Theorem 3.2 of Hübotter et al. (2024) can be extended to finite-dimensional reproducing kernel Hilbert spaces (Hübotter et al., 2024, Appendix C.6.4).

- The "maximum information gain of $n$ iterations", $\gamma_n$, in the statement of Hübotter et al. (2024) is bounded as follows (Srinivas et al., 2009, Appendix C.3): $\gamma_n \leq d\log(1 + \hat{\lambda}_n/\lambda')$.

$\square$

---

[9]See Appendix K.4 for the expressions with non-normalized embeddings.
[10]Similar assumptions have been made by Bogunovic et al. (2015) and Kothawade et al. (2020).

# D  FURTHER INSIGHTS ON ACTIVE FINE-TUNING

We expand the analysis of our results that we summarized in §5. We analyze aspects of the two key contributions of our work separately: In the following, we analyze the performance of SIFT in active fine-tuning, and in §E, we analyze the performance of test-time fine-tuning more generally.

**Insight 6: SIFT's improvement over NN grows with dataset size.**   As shown in Figure 10, we find that the relative improvement of SIFT over Nearest Neighbor retrieval grows with dataset size. We suspect that going from a small-size dataset to a medium-size dataset, the additional performance stems mainly from the ability of SIFT to adaptively select the same data for multiple gradient steps. Going from a medium-size dataset to a large-size dataset, we suspect that the additional performance stems mainly from the ability of SIFT to select more diverse data points.

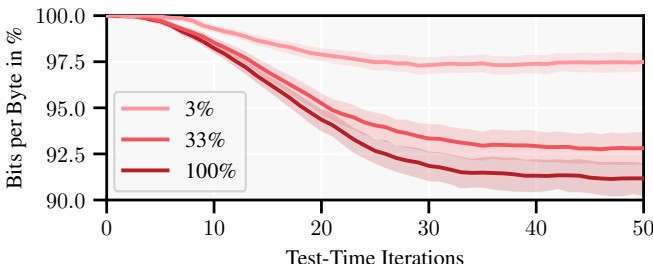

Figure 10: Bits per byte (in % relative to the Nearest Neighbor retrieval baseline, ↓ better). We evaluate data selection from 3%, 33%, and 100% of the Pile training dataset. We see a clear trend that SIFT's improvement over Nearest Neighbor retrieval grows with dataset size — even from 33% to 100% with the highly curated Pile dataset.

**Insight 7: Points with high negative cosine similarity *may* help.**   With the Roberta embedding model, we find that there are no negative cosine similarities in the data (cf. Figure 21 in §J). Choosing different embeddings such as influence embeddings can give negative cosine similarities (Xia et al., 2024, Appendix K.2). Inspection of those points found by Xia et al. (2024) suggests that they can be equally informative as points with high positive cosine similarity. Our derivation of SIFT naturally addresses this by selecting points with large *absolute* cosine similarity. Geometrically, points with positive or negative cosine similarity are both equally "parallel" to the test prompt. Our theoretical results suggest that the informativeness of a data point is closely related to how parallel its embedding is to the test prompt. We leave further investigation to future work.

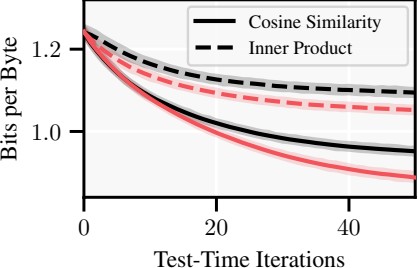

Figure 11: Data selection via SIFT (red) and Nearest Neighbor (black) performs best with normalized embeddings.

**Insight 8: Normalizing embeddings helps.**   We evaluate the performance of Nearest Neighbor retrieval and SIFT with or without explicitly normalized embeddings in Figure 11. We find that for both selection strategies, normalizing embeddings consistently improves performance. Previously, Hardt & Sun (2024) minimized the Euclidean distance between unnormalized embeddings, which we find to perform identically to maximizing cosine similarity.

# E    FURTHER INSIGHTS ON TEST-TIME FINE-TUNING

**Insight 9: Scaling pre-training compute may not be all you need.**  In Table 2 of §A, we compare state-of-the-art LLMs to our test-time fine-tuned models. We show that our Phi-3 with test-time fine-tuning outperforms all evaluated base models, from a wide selection of state-of-the-art LLMs, by a large margin. Notably, we see a clear advantage of using stronger base models, i.e., better initializations. The leading base model Gemma-2 (27B, Team et al., 2024), which is $7\times$ larger and more recent than Phi-3, achieves $0.629$ bits per byte, whereas our test-time fine-tuned Phi-3 achieves $0.595$ bits per byte. This indicates that scaling pre-training compute is not all you need to achieve state-of-the-art performance, and that test-time fine-tuning can be an effective method for improving the performance of a base LLM.

**Insight 10: Test-time fine-tuning outperforms in-context learning in "hard" tasks.**  Interestingly, we observe that across all evaluated models, updating the base model via fine-tuning as opposed to augmenting the models' context leads to large improvements on the DeepMind Math, GitHub, ArXiv, and FreeLaw datasets. We include the per-dataset results in §F.2. These datasets contain school-level math problems, code, scientific papers, and court opinions, which are often colloquially understood as tasks that require "understanding" or "reasoning". In the case of DeepMind Math and ArXiv, augmenting the models' context does consistently not improve the performance of the base model at all, whereas test-time fine-tuning can lead to significant performance improvements.

**Insight 11: Test-time fine-tuning yields largest gains at the boundary of the data distribution.**  In Figure 12, we plot the improvement of test-time fine-tuning with SIFT over the base model against the weight of a dataset in the Pile. We observe the trend that test-time fine-tuning yields largest performance improvements for datasets that have a smaller weight in the Pile. We hypothesize that this trend occurs because the weight of a dataset in the Pile corresponds roughly to the weight of similar data in the pre-training dataset of GPT-2, in which case the performance gains would be largest for prompts that are at the "boundary" of the data distribution. Notable is the outlier of the large GitHub dataset where test-time fine-tuning leads to large performance gains. We hypothesize that this is because coding is relatively dissimilar to other data in the Pile, and therefore the GitHub dataset can be seen as "small" relative to the rest of the data.

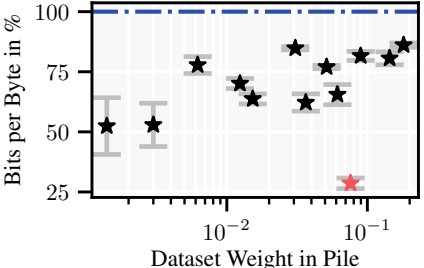

Figure 12: Improvement of $50$ test-time iterations over the base model (blue; ↓ better) with SIFT against the percentage of bytes occupied by the dataset in the Pile. Error bars correspond to standard errors. We observe the trend that test-time fine-tuning benefits prompts at the "boundary" of the data distribution most. The "outlier" GitHub dataset is highlighted in red.

We make the observation that if the problem domain is large (like general language modeling), almost every sub-task can be seen as at the "boundary" / as an "outlier". We see that datasets closest to the center of mass of the data distribution do not benefit as much from test-time fine-tuning as datasets that are further away from the center of mass. Therefore, we expect test-time fine-tuning to benefit those models most that are learning a diverse data distribution as opposed to models that are learning a very concentrated data distribution.

**Insight 12: The order of fine-tuning data does not matter.**  In Figure 13, we evaluate the performance of test-time fine-tuning with Nearest Neighbor retrieval when taking gradient steps in the order of selected data compared to reversed order. We find that the order of gradient steps does not affect the final performance. This indicates that sequentially fine-tuning on selected data is not necessary, and that batched gradient steps can be used to further speed up test-time fine-tuning. We leave a detailed exploration of batched updates to future work.

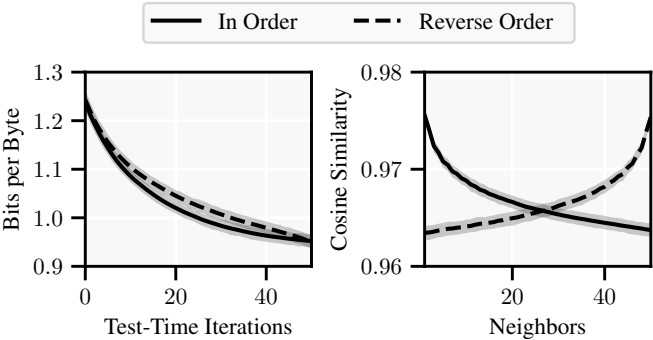

Figure 13: Taking gradient steps in order of selected data compared to reversed order. Data is selected using Nearest neighbor retrieval. We observe that the order of gradient steps does not affect the final performance.

**Insight 13: Test-time fine-tuning works also when fine-tuning only the last linear layer.** Motivated by the linear representation hypothesis (cf. Assumption 3.1) which informs SIFT's surrogate model for data selection, we evaluate whether we can fine-tune this surrogate model directly instead of fine-tuning the full model. Concretely, we fine-tune only the last linear layer of the LLM, keeping its latent space fixed. The gradients for this linear surrogate model can be computed efficiently at almost no cost. Remarkably, we find in Figure 14 that large gains of test-time fine-tuning can already be realized by fine-tuning only the last linear layer. Given these preliminary results with GPT-2 it would be interesting to evaluate the performance gains of fine-tuning the linear head of larger base models.

**Insight 14: Test-time fine-tuning works also with parameter-efficient fine-tuning.** In our experiments

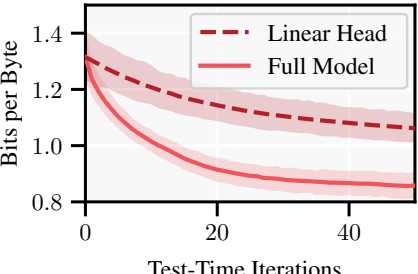

Figure 14: Bits per byte (↓ better) against the number of test-time iterations. We compare fine-tuning only the linear head to fine-tuning the full model. We use learning rate 1e−4 and evaluate on 0.1% of the full test set.

with Phi-3, we use Low-Rank Adaptation (Lora, Hu et al., 2022) with a rank of 64. We find that LoRA converges slower than fine-tuning the full model, and therefore use the learning rate 5e−4, which is a factor 10 larger than the learning rate used for fine-tuning the full model. In Figure 15, we evaluate the performance of LoRA compared to fine-tuning the full model. On the smaller GPT-2 and GPT-2-large we use a rank of 32. We generally observe that fine-tuning with LoRA can recover roughly the same performance as fine-tuning the full model. We expect that with more careful tuning of the learning rate, learning curves could be made more similar.

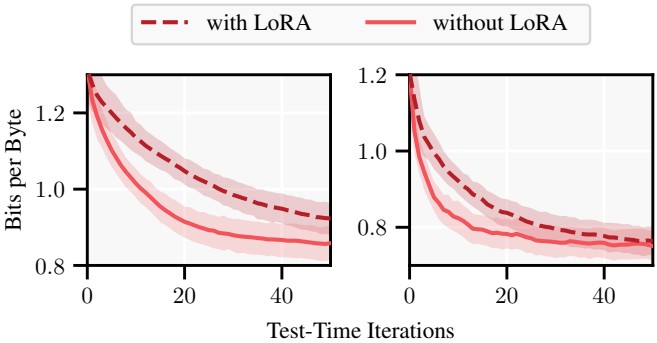

Figure 15: Bits per byte (↓ better) against the number of test-time iterations. We compare parameter-efficient fine-tuning with LoRA and fine-tuning the full model. We use 0.1% of the full test set.

**Insight 15: Larger models appear to learn faster at test-time.** We find that with a larger model (e.g., GPT-2-large vs GPT-2), a smaller $\lambda'$ tends to be more beneficial. For example, keeping the learning rate fixed at $5e-5$, using $\text{SIFT}(0.1)$ is the best choice for GPT-2, but leads to slight overfitting at later iterations for GPT-2-large as shown in Figure 16. Recall that a smaller $\lambda'$ leads to more diverse sampling of the data space. Thus, this observed trend indicates that larger models learn faster, and therefore benefit more from less redundant training data. The same trend can also be observed from the behavior of NN-F from Figure 17: GPT-2-large overfits much faster with NN-F than GPT-2. This offers a potential explanation why the advantage of SIFT over Nearest Neighbor retrieval grows with larger models (cf. §F.1).

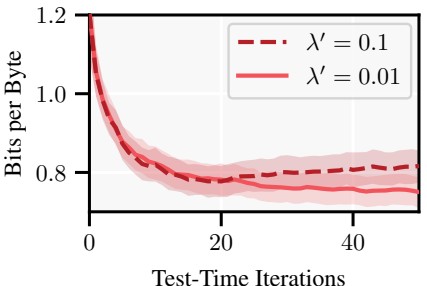

Figure 16: Bits per byte ($\downarrow$ better) with GPT-2-large and varying $\lambda'$. A larger $\lambda'$ can lead to overfitting in later iterations. We use 0.1% of the full test set.

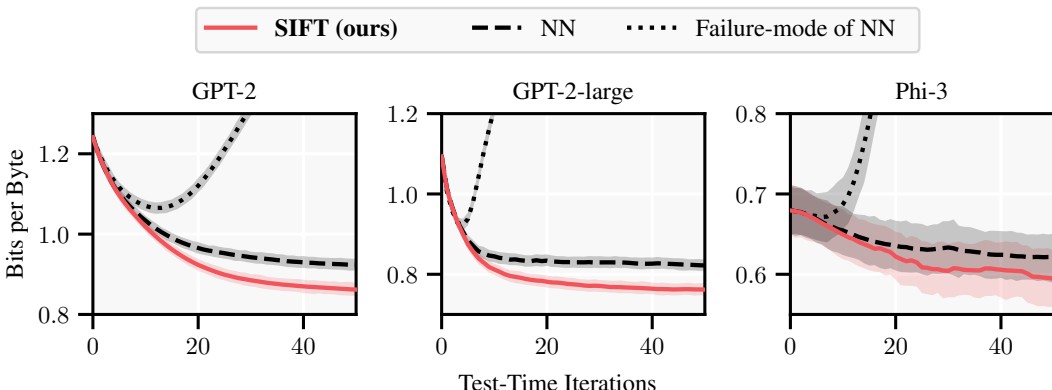

Figure 17: Bits per byte ($\downarrow$ better) against the number of test-time iterations with various base models.

# F    EXTENDED RESULTS

This section includes additional per-dataset results to support our findings on active fine-tuning and test-time fine-tuning.

## F.1    ACTIVE FINE-TUNING

We compare SIFT against the data selection baselines Uncertainty Sampling (US), Nearest Neighbor retrieval (NN), and the failure-mode of Nearest Neighbor retrieval (with information duplication) that repeatedly retrieves the same point (NN-F). Our results with GPT-2 as base model are summarized in the main text in Table 1.

- In Table 4, we include the comparison with **GPT-2-large**.
- In Table 5, we include the comparison with **Phi-3**.

We find that our results on GPT-2 are consistent across all models. In particular, test-time fine-tuning with SIFT improves the base model on *all* datasets of the Pile, often significantly. SIFT outperforms Uncertainty Sampling and Nearest Neighbor retrieval consistently. Notably, we find that the improvement of SIFT over Nearest Neighbor retrieval is larger with stronger base models, indicating that informativeness of data becomes more important the stronger the base model.

## F.2    TEST-TIME FINE-TUNING

We compare the in-context baseline against test-time fine-tuning.

- In Table 6, we include the comparison with **GPT-2**.
- In Table 7, we include the comparison with **GPT-2-large**.
- In Table 8, we include the comparison with **Phi-3**.

We find that test-time fine-tuning consistently outperforms in-context learning with GPT-2 and GPT-2-large. With Phi-3, in-context learning and test-time fine-tuning have roughly matching performance, though test-time fine-tuning is more computationally efficient (cf. Figure 7). Interestingly, we observe that test-time fine-tuning leads to large gains on math ("DeepMind Math") and coding ("GitHub") on all models, two tasks that require more complex reasoning.

|  | US | NN | NN-F | SIFT | $\Delta$ |
|---|---|---|---|---|---|
| NIH Grants | 96.6 (1.6) | 77.9 (4.8) | 107.6 (19.8) | **51.9** (9.3) | ↓26.0 |
| US Patents | 86.8 (2.3) | 78.9 (2.6) | 129.1 (7.7) | **64.7** (3.8) | ↓14.2 |
| Enron Emails | **73.9** (12.3) | **68.6** (13.6) | 102.9 (23.1) | 55.5 (12.2) | ↓13.1 |
| GitHub | 45.2 (2.4) | 42.8 (2.2) | 62.0 (4.5) | **31.0** (2.2) | ↓11.8 |
| Wikipedia | 71.0 (2.0) | 71.5 (2.0) | 141.3 (3.5) | **64.4** (2.2) | ↓6.6 |
| PubMed Abstr. | 94.5 (0.4) | 93.7 (0.6) | 202.6 (1.6) | **87.8** (0.7) | ↓5.9 |
| ArXiv | 90.6 (1.8) | 90.2 (2.0) | 175.8 (5.7) | **84.8** (2.1) | ↓5.4 |
| Hacker News | **79.4** (2.6) | **79.0** (2.9) | 138.7 (4.4) | 75.6 (3.6) | ↓3.4 |
| Stack Exchange | 84.1 (0.7) | 84.6 (0.8) | 165.2 (1.8) | **80.7** (0.9) | ↓3.4 |
| Common Crawl | 93.7 (0.6) | 89.9 (0.7) | 163.6 (2.1) | **87.1** (1.0) | ↓2.8 |
| PubMed Central | **87.9** (2.7) | **87.6** (2.7) | 157.8 (4.6) | 85.4 (3.1) | ↓2.2 |
| FreeLaw | **66.8** (4.2) | **67.4** (4.1) | 132.0 (6.4) | 68.3 (4.2) | ↑1.5 |
| DeepMind Math | **71.2** (2.2) | **72.2** (2.0) | 186.1 (4.1) | 74.2 (2.3) | ↑3.0 |
| *All* | 82.6 (0.6) | 80.6 (0.6) | 153.3 (1.4) | **74.9** (0.7) | ↓5.7 |

Table 4: Results with **GPT-2-large**. Bits per byte (in % relative to the base model, ↓) after 50 test-time iterations on individual datasets of the Pile. We only include datasets with at least 10 examples in our test set. **Bold** numbers denote the best performing selected subset. Numbers in parentheses are standard errors. $\Delta$ denotes the performance gain of SIFT over the strongest baseline.

| | US | NN | NN-F | SIFT | Δ |
|---|---|---|---|---|---|
| GitHub | 80.6 | 80.8 | 105.2 | 46.5 | ↓34.1 |
| US Patents | 95.4 | 94.2 | 274.6 | 83.7 | ↓10.5 |
| Enron Emails | 113.6 | 86.6 | 319.9 | 78.7 | ↓7.9 |
| Wikipedia | 84.6 | 85.5 | 263.2 | 79.2 | ↓5.4 |
| PubMed Abstr. | 93.5 | 93.3 | 301.8 | 89.5 | ↓3.8 |
| NIH Grants | 100.4 | 100.1 | 327.6 | 98.6 | ↓1.5 |
| ArXiv | 95.5 | 96.5 | 282.4 | 94.3 | ↓1.2 |
| Common Crawl | 95.3 | 94.9 | 257.0 | 93.7 | ↓1.2 |
| PubMed Central | 80.3 | 82.1 | 204.9 | 79.7 | ↓0.6 |
| DeepMind Math | 76.4 | 75.5 | 221.4 | 75.3 | ↓0.2 |
| Hacker News | 95.1 | 94.8 | 243.8 | 95.0 | ↑0.2 |
| FreeLaw | 66.9 | 67.8 | 178.0 | 67.2 | ↑0.3 |
| Stack Exchange | 99.7 | 98.7 | 309.9 | 99.4 | ↑0.7 |
| *All* | 92.0 (1.1) | 91.6 (1.1) | 256.6 (7.1) | **85.7** (2.0) | ↓5.9 |

Table 5: Results with **Phi-3**. Bits per byte (in % relative to the base model, ↓) after 50 test-time iterations on individual datasets of the Pile. **Bold** numbers denote the best performing selected subset. Numbers in parentheses are standard errors. Δ denotes the performance gain of SIFT over the strongest baseline.

| | Context | Fine-Tuning | Δ |
|---|---|---|---|
| GitHub | 74.5 (2.5) | **28.6** (2.2) | ↓45.9 |
| DeepMind Math | 100.4 (0.1) | **70.1** (2.1) | ↓30.3 |
| US Patents | 86.8 (2.5) | **62.2** (3.6) | ↓24.6 |
| Enron Emails | **73.3** (9.8) | **52.4** (11.8) | ↓20.9 |
| FreeLaw | 85.5 (4.0) | **65.5** (4.2) | ↓20.0 |
| Stack Exchange | 96.7 (0.3) | **77.0** (0.7) | ↓19.7 |
| ArXiv | 99.2 (1.4) | **81.6** (1.9) | ↓17.6 |
| Wikipedia | 77.4 (2.1) | **63.7** (2.1) | ↓13.7 |
| PubMed Central | 92.8 (3.1) | **80.6** (2.7) | ↓12.2 |
| Hacker News | 89.0 (3.8) | **77.8** (3.5) | ↓11.2 |
| NIH Grants | **63.7** (9.5) | **52.9** (9.0) | ↓10.8 |
| Common Crawl | 93.4 (0.7) | **86.1** (0.9) | ↓7.3 |
| PubMed Abstr. | 91.8 (0.6) | **84.8** (0.7) | ↓7.0 |
| *All* | 89.3 (0.5) | **73.2** (0.7) | ↓16.1 |

Table 6: Comparison between the in-context baseline and test-time fine-tuning with **GPT-2**. Bits per byte (in % relative to the base model, ↓) after 50 test-time iterations on individual datasets of the Pile. We only include datasets with at least 10 examples in our test set. **Bold** numbers denote the best performing selected subset. Numbers in parentheses are standard errors. Δ denotes the performance gain of test-time fine-tuning over in-context learning.

|  | Context | Fine-Tuning | Δ |
|---|---|---|---|
| GitHub | 74.6 (2.5) | **31.0** (2.2) | ↓43.6 |
| DeepMind Math | 100.2 (0.7) | **74.2** (2.3) | ↓26.0 |
| US Patents | 87.4 (2.5) | **64.7** (3.8) | ↓22.7 |
| FreeLaw | 87.2 (3.6) | **68.3** (4.2) | ↓18.9 |
| Hacker News | 92.6 (2.7) | **75.6** (3.6) | ↓17.0 |
| Stack Exchange | 97.2 (0.4) | **80.7** (0.9) | ↓16.5 |
| NIH Grants | **67.7** (9.4) | **51.9** (9.3) | ↓15.8 |
| Enron Emails | **71.9** (10.2) | **55.5** (12.2) | ↓15.5 |
| ArXiv | 98.8 (1.8) | **84.8** (2.1) | ↓14.0 |
| Wikipedia | 76.6 (2.1) | **64.4** (2.2) | ↓12.2 |
| PubMed Central | 92.3 (3.3) | **85.4** (3.1) | ↓6.9 |
| Common Crawl | 93.5 (0.7) | **87.1** (1.0) | ↓6.4 |
| PubMed Abstr. | 91.6 (0.6) | **87.8** (0.7) | ↓3.8 |
| *All* | 89.4 (0.5) | **74.9** (0.7) | ↓14.5 |

Table 7: Comparison between the in-context baseline and test-time fine-tuning with **GPT-2-large**. Bits per byte (in % relative to the base model, ↓) after 50 test-time iterations on individual datasets of the Pile. We only include datasets with at least 10 examples in our test set. **Bold** numbers denote the best performing selected subset. Numbers in parentheses are standard errors. Δ denotes the performance gain of test-time fine-tuning over in-context learning.

|  | Context | Fine-Tuning | Δ |
|---|---|---|---|
| DeepMind Math | 100.8 | 75.3 | ↓25.5 |
| GitHub | 71.3 | 46.5 | ↓24.8 |
| FreeLaw | 78.2 | 67.2 | ↓11.0 |
| ArXiv | 101.0 | 94.3 | ↓6.4 |
| Enron Emails | 81.8 | 78.7 | ↓3.1 |
| Hacker News | 97.6 | 95.0 | ↓2.6 |
| Stack Exchange | 100.9 | 99.4 | ↓1.4 |
| PubMed Central | 79.9 | 79.7 | ↓0.2 |
| US Patents | 83.3 | 83.7 | ↑0.4 |
| Wikipedia | 77.1 | 79.2 | ↑2.1 |
| NIH Grants | 95.1 | 98.6 | ↑3.5 |
| Common Crawl | 89.9 | 93.7 | ↑3.8 |
| PubMed Abstr. | 85.7 | 89.5 | ↑3.8 |
| *All* | **87.1** (1.7) | **85.7** (2.0) | ↓1.4 |

Table 8: Comparison between the in-context baseline and test-time fine-tuning with **Phi-3**. Bits per byte (in % relative to the base model, ↓) after 50 test-time iterations on individual datasets of the Pile. **Bold** numbers denote the best performing selected subset. Numbers in parentheses are standard errors. Δ denotes the performance gain of test-time fine-tuning over in-context learning.

## G    SIFT MAXIMIZES INFORMATION GAIN

We discuss here briefly that SIFT can be interpreted as maximizing the information gain of data $X_n$ on the response to the prompt $\boldsymbol{x}^\star$.

This probabilistic interpretation takes the perspective that the sequence model predicting the next token is a *probabilistic model with a prior belief* over its state $\boldsymbol{W}$ which induces an epistemic prior belief over what might be the next token.[11] Our main text describes a closed loop where this sequence model interacts with a non-parametric memory (i.e., the data space) to update its epistemic beliefs about $\boldsymbol{W}$, obtaining posterior beliefs $\boldsymbol{W} \mid D$ conditional on the selected data $D$. Again, these posterior epistemic beliefs induce an epistemic uncertainty over what might be the next token. We discuss in the following how SIFT can be interpreted probabilistically; as the model interacting with the non-parametric memory with the goal of reducing its posterior uncertainty about the next token.

Our brief overview will proceed as follows:

- We establish fundamentals from information theory and Gaussian processes, which are a tractable probabilistic model (§G.1).

- We define the prior belief and probabilistic observation model and derive the posterior belief (§G.2).

- We show that, in this probabilistic model, SIFT can be interpreted as maximizing the information gain of the data about the response to the prompt $\boldsymbol{x}^\star$ (§G.3).

- We show that balancing relevance and diversity of data is a natural consequence of maximizing information gain (§G.4).

SIFT uses relatively simple probabilistic surrogate models that are tractable, and which remarkably lead to strong empirical performance. Hübotter et al. (2024) cover the probabilistic interpretation in greater detail.

### G.1    PRELIMINARIES: INFORMATION THEORY AND GAUSSIAN PROCESSES

**Information Theory**    We briefly recap several important concepts from information theory. The (differential) entropy $\mathrm{H}[\boldsymbol{f}] \doteq \mathbb{E}_{p(\boldsymbol{f})}[-\log p(\boldsymbol{f})]$ of a random vector $\boldsymbol{f}$ is one possible measure of uncertainty about $\boldsymbol{f}$. Here, $-\log p(\boldsymbol{f})$ is also called the suprisal about an event with density $p(\boldsymbol{f})$. The entropy can be interpreted as the expected suprisal about $\boldsymbol{f}$ upon realization. The conditional entropy $\mathrm{H}[\boldsymbol{f} \mid \boldsymbol{y}] \doteq \mathbb{E}_{p(\boldsymbol{f},\boldsymbol{y})}[-\log p(\boldsymbol{f} \mid \boldsymbol{y})]$ is the (expected) posterior uncertainty about $\boldsymbol{f}$ after observing the random vector $\boldsymbol{y}$. The information gain $\mathrm{I}(\boldsymbol{f}; \boldsymbol{y}) = \mathrm{H}[\boldsymbol{f}] - \mathrm{H}[\boldsymbol{f} \mid \boldsymbol{y}]$ measures the (expected) reduction in uncertainty about $\boldsymbol{f}$ due to $\boldsymbol{y}$. Refer to Cover (1999) for more details.

**Gaussian Processes**    The stochastic process $f$ is a Gaussian process (GP, Williams & Rasmussen (2006)), denoted $f \sim \mathcal{GP}(\mu, k)$, with mean function $\mu$ and kernel $k$ if for any finite subset $X = \{\boldsymbol{x}_1, \ldots, \boldsymbol{x}_n\} \subseteq \mathcal{X}$, $\boldsymbol{f}_X \sim \mathcal{N}(\boldsymbol{\mu}_X, \boldsymbol{K}_X)$ is jointly Gaussian with mean vector $(\boldsymbol{\mu}_X)_i = \mu(\boldsymbol{x}_i)$ and covariance matrix $(\boldsymbol{K}_X)_{i,j} = k(\boldsymbol{x}_i, \boldsymbol{x}_j)$. A Gaussian process can be interpreted as capturing an epistemic functional belief, i.e., a belief over functions. Our linear surrogate model from Assumption 3.1 leads to a Gaussian process with the linear kernel described in the main text. That is, our surrogate model assumption can be interpreted as the *prior belief* that the ground truth function predicting the next token is a logit-linear function in a latent representation space. This is closely linked to the hypothesis that LLMs learn linear representations of high-level concepts, which is widely known as the "linear representation hypothesis" (e.g., Park et al., 2024; Mikolov et al., 2013; Arora et al., 2016; Elhage et al., 2022). There are two lenses through which to view such linear Gaussian processes: the *weight-space* view which considers a belief about weights $\boldsymbol{W}$, or the *function-space* view which directly considers the belief about functions $f$. Both views are equivalent, and we will focus on the function-space view in the following.

For Gaussian random vectors $\boldsymbol{f}$ and $\boldsymbol{y}$, the entropy is $\mathrm{H}[\boldsymbol{f}] = \frac{d}{2}\log(2\pi e) + \frac{1}{2}\log\det\mathrm{Var}(\boldsymbol{f})$ and the information gain is $\mathrm{I}(\boldsymbol{f}; \boldsymbol{y}) = \frac{1}{2}(\log\det\mathrm{Var}(\boldsymbol{f}) - \log\det\mathrm{Var}(\boldsymbol{f} \mid \boldsymbol{y}))$.

---

[11]This *epistemic* uncertainty is distinct from the irreducible *aleatoric* uncertainty of natural language, such as uncertainty about the continuation of "I love . . .".

### G.2 PROBABILISTIC OBSERVATION MODEL

We will focus in the following on the case of regression, which we introduced in Appendix K.5. We suppose that observations of $f$ follow the probabilistic model

$$y_{\boldsymbol{x}} = f_{\boldsymbol{x}} + \varepsilon_{\boldsymbol{x}},$$

where we make the following assumptions about the prior distribution of $f$ and the noise $\varepsilon_{\boldsymbol{x}}$:

**Assumption G.1** (Gaussian prior). We assume that $f \sim \mathcal{GP}(\mu, k)$ with known mean function $\mu$ and kernel $k$.

**Assumption G.2** (Gaussian noise). We assume that the noise $\varepsilon_{\boldsymbol{x}}$ is mutually independent and zero-mean Gaussian with known variance $\rho^2 > 0$.

Under Assumptions G.1 and G.2, the posterior distribution of $f$ after observing points $X$ with values $\boldsymbol{y}_X$ is $\mathcal{GP}(\mu_n, k_n)$ with

$$\mu_n(\boldsymbol{x}) = \mu(\boldsymbol{x}) + \boldsymbol{k}_X^\top(\boldsymbol{x})(\boldsymbol{K}_{XX} + \rho^2 \boldsymbol{I})^{-1}(\boldsymbol{y}_X - \boldsymbol{\mu}_X),$$
$$k_n(\boldsymbol{x}, \boldsymbol{x}') = k(\boldsymbol{x}, \boldsymbol{x}') - \boldsymbol{k}_X^\top(\boldsymbol{x})(\boldsymbol{K}_{XX} + \rho^2 \boldsymbol{I})^{-1}\boldsymbol{k}_X(\boldsymbol{x}'),$$
$$\sigma_n^2(\boldsymbol{x}) = k_n(\boldsymbol{x}, \boldsymbol{x}).$$

### G.3 THE PROBABILISTIC INTERPRETATION OF SIFT

Observe that the above definition of $\sigma_n^2$ matches the definition from Equation (2).[12] That is, under the above probabilistic model,

$$\sigma_n^2(\boldsymbol{x}) = \mathrm{Var}(f(\boldsymbol{x}) \mid y_{1:n}).$$

As such, SIFT($\rho^2$) is minimizing the variance of the response to the prompt $\boldsymbol{x}^\star$ after observing the data $X_n$:

$$\boldsymbol{x}_{n+1} = \underset{\boldsymbol{x} \in \mathcal{D}}{\arg\min} \, \mathrm{Var}(f(\boldsymbol{x}^\star) \mid y_{1:n}, y(\boldsymbol{x})).$$

By simple algebraic manipulation this can be seen to be equivalent to maximizing the information gain of the data on the response to the prompt $\boldsymbol{x}^\star$:

$$\boldsymbol{x}_{n+1} = \underset{\boldsymbol{x} \in \mathcal{D}}{\arg\max} \, \frac{1}{2}\Big( \underbrace{\log \mathrm{Var}(f(\boldsymbol{x}^\star) \mid y_{1:n})}_{\text{const}} - \log \mathrm{Var}(f(\boldsymbol{x}^\star) \mid y_{1:n}, y(\boldsymbol{x})) \Big)$$
$$= \underset{\boldsymbol{x} \in \mathcal{D}}{\arg\max} \, \mathrm{I}(f(\boldsymbol{x}^\star); y(\boldsymbol{x}) \mid y_{1:n}). \tag{7}$$

**Discussion** The above offers a very intuitive probabilistic interpretation of SIFT($\rho^2$). In this probabilistic interpretation, the regularization parameter $\lambda'$ of SIFT is equal to the observation noise $\rho^2$. Intuitively, larger observation noise leads to slower convergence of the estimate of $f$, analogously to our discussion of larger regularization parameter and smaller step size in Proposition 3.3.

The reason why SIFT($\rho^2$) can be interpreted *both* as minimizing the variance and as minimizing the entropy of the response to the prompt $\boldsymbol{x}^\star$ is that for Gaussians, variance is proportional to the entropy of the response to the prompt $\boldsymbol{x}^\star$. As observed by Hübotter et al. (2024), if learning is amortized with respect to multiple prompts $\{\boldsymbol{x}_1^\star, \ldots, \boldsymbol{x}_m^\star\} = \mathcal{A}$, this ceases to be the case and the two objectives lead to different data selection schemes. It appears to be a special property of non-amortized transductive active learning that measures of uncertainty and resulting data selection schemes are interchangeable.

A quick remark is in order. SIFT does not only maximize the marginal information gain as shown in Equation (7), if Assumption C.1 is satisfied, it also maximizes the joint information gain $\mathrm{I}(f(\boldsymbol{x}^\star); y_{1:n})$. That is, in this case the "entropy reduction" of data $X_n$ selected by SIFT achieves a constant factor approximation of the maximum possible joint information gain $\max_{X \subseteq \mathcal{D}, |X| \le n} \mathrm{I}(f(\boldsymbol{x}^\star); \boldsymbol{y}(X))$ due to the seminal result on monotone submodular function maximization of Nemhauser et al. (1978).

---

[12]Notably, it can also be shown that $\mu_n$ is the closed-form solution to the regularized loss from Equation (10).

### G.4 HOW SIFT BALANCES RELEVANCE AND DIVERSITY

In §C.1, we discussed how SIFT chooses data that is both relevant and diverse. The probabilistic interpretation offers a simple explanation for how this behavior naturally emerges from selecting the most informative data. To this end, observe that the information gain from Equation (7) can be expressed as

$$\mathrm{I}(f(\boldsymbol{x}^\star); y(\boldsymbol{x}) \mid y_{1:n}) = \underbrace{\mathrm{I}(f(\boldsymbol{x}^\star); y(\boldsymbol{x}))}_{\text{relevance}} - \underbrace{\mathrm{I}(f(\boldsymbol{x}^\star); y(\boldsymbol{x}); y_{1:n})}_{\text{redundancy}} \tag{8}$$

where $\mathrm{I}(\boldsymbol{f}; \boldsymbol{x}; \boldsymbol{y}) \doteq \mathrm{I}(\boldsymbol{f}; \boldsymbol{x}) - \mathrm{I}(\boldsymbol{f}; \boldsymbol{x} \mid \boldsymbol{y}) = \mathrm{I}(\boldsymbol{f}; \boldsymbol{x}) + \mathrm{I}(\boldsymbol{f}; \boldsymbol{y}) - \mathrm{I}(\boldsymbol{f}; \boldsymbol{x}, \boldsymbol{y})$ denotes the multivariate information gain (Murphy, 2023). The multivariate information gain is a measure of the redundancy of $\boldsymbol{x}$ and $\boldsymbol{y}$ in predicting $\boldsymbol{f}$, and is therefore often called simply "redundancy" (which is the opposite of "synergy"). Equation (8) shows that the balancing of relevance and non-redundancy (i.e., diversity) arises naturally from maximizing the information gain. Note that the tradeoff between relevance and diversity is governed by the noise parameter $\rho^2 \approx \lambda'$ of the probabilistic model.

### G.5 THE PERSPECTIVE OF CLASSIFICATION

The above interpretation takes the perspective of regression. However, the above interpretation can be extended to classification. We will focus here on the case of binary classification for notational convenience, but the same argument can be made for multi-class classification (Williams & Rasmussen, 2006, Section 3.5).

In (binary) Gaussian Process Classification the logit $f \sim \mathcal{GP}(\mu, k)$ is modeled as a Gaussian process, and the likelihood follows the model introduced in Section 3: $y(\boldsymbol{x}) \sim \mathrm{Bern}(s(f(\boldsymbol{x})))$ where we have Bernoulli rather than categorical feedback and use the logistic function $s(a) \doteq 1/(1 + e^{-a})$ rather than the softmax by virtue of restricting to binary classification.

The standard approach (Williams & Rasmussen, 2006, Section 3.4) is to approximate the posterior distribution of the latent function $f$ given observations $y_{1:n}$ by a Gaussian using Laplace's method. This Gaussian can be shown to have covariance $(\boldsymbol{K}_{X_n}^{-1} + \boldsymbol{W})^{-1}$ with $\boldsymbol{W} \succeq \kappa^{-1} \boldsymbol{I}_n$ where $\kappa \doteq \sup_{a \leq B} 1/\dot{s}(a)$ and $\dot{s}(a) = s(a)(1 - s(a))$ denotes the derivative of the logistic function.[13] It is then straightforward to derive that

$$\sigma_n^2(\boldsymbol{x}^\star) = k(\boldsymbol{x}^\star, \boldsymbol{x}^\star) - \boldsymbol{k}_{X_n}^\top(\boldsymbol{x}^\star)(\boldsymbol{K}_{X_n} + \boldsymbol{W}^{-1})^{-1}\boldsymbol{k}_{X_n}(\boldsymbol{x}^\star)$$
$$\leq k(\boldsymbol{x}^\star, \boldsymbol{x}^\star) - \boldsymbol{k}_{X_n}^\top(\boldsymbol{x}^\star)(\boldsymbol{K}_{X_n} + \kappa \boldsymbol{I}_n)^{-1}\boldsymbol{k}_{X_n}(\boldsymbol{x}^\star)$$

Thus, SIFT minimizes a tight upper bound to the (approximate) posterior variance of the latent function $f$ at the prompt $\boldsymbol{x}^\star$. The same relationship to maximizing information gain that was discussed above applies.

---

[13]In the binary case, this is equal to the more general $\kappa$ from the main text.

# H EFFICIENT COMPUTATION OF SIFT

In the following, we show how to select data via SIFT at low computational cost. Our implementation extends the Faiss library (Johnson et al., 2019; Douze et al., 2024) for Nearest Neighbor retrieval. We open-source the `activeft` (Active Fine-Tuning) library which can be used as a drop-in replacement for Nearest Neighbor retrieval.

In our runtime analysis, we will denote by $K$ the size of the data space $\mathcal{D}$, and by $N$ the number of points to be selected. We describe two implementations of SIFT:

1. The first exact implementation has sequential computation cost $O(K^2 N)$, however, computation can be effectively parallelized on a GPU.
2. The second "fast" implementation assumes submodularity (i.e., Assumption C.1) and has computation cost $\widetilde{O}(K + N^3)$ where $\widetilde{O}(\cdot)$ suppresses log-factors. This cost is only marginally above the cost of Nearest Neighbor retrieval.

Both implementations achieve virtually identical performance gains (cf. Figure 19 (right)), which is further evidence that Assumption C.1 is satisfied in our language modeling setting.

## H.1 EXACT IMPLEMENTATION

The central object of the first implementation is the conditional kernel matrix of the data space given the selected points $X_n$:

$$\boldsymbol{K}_n \doteq \boldsymbol{K}_{\mathcal{D}} - \boldsymbol{K}_{\mathcal{D}, X_n}(\boldsymbol{K}_{X_n} + \lambda' \boldsymbol{I}_n)^{-1} \boldsymbol{K}_{X_n, \mathcal{D}}.$$

The entries $k_n(\boldsymbol{x}, \boldsymbol{x}')$ of this matrix can be updated efficiently via the following relation (Chowdhury & Gopalan, 2017, Appendix F) arising from properties of the Schur complement:

$$k_n(\boldsymbol{x}, \boldsymbol{x}') = k_{n-1}(\boldsymbol{x}, \boldsymbol{x}') - \frac{k_{n-1}(\boldsymbol{x}, \boldsymbol{x}_n) k_{n-1}(\boldsymbol{x}_n, \boldsymbol{x}')}{k_{n-1}(\boldsymbol{x}_n, \boldsymbol{x}_n) + \lambda'}. \tag{9}$$

The implementation is detailed in Algorithm 1. The computation of the objective value in line 4 and the kernel matrix update in line 5 can be parallelized on a GPU. Thus, the main bottleneck of this implementation is the requirement that the kernel matrix of size $K \times K$ fits onto a GPU. In case this is not possible, such as with large data spaces, the following two sections detail methods to reduce the computational cost.

---

**Algorithm 1** SIFT($\lambda'$)

1: **Input:** prompt $\boldsymbol{x}^\star$, data space $\mathcal{D}$, (initial) kernel matrix $k_0(\boldsymbol{x}, \boldsymbol{x}') = \phi(\boldsymbol{x})^\top \phi(\boldsymbol{x}')$, $\boldsymbol{x}, \boldsymbol{x}' \in \mathcal{D}$,
   number of points to select $N$
2: **Output:** set of selected points $\{\boldsymbol{x}_1, \ldots, \boldsymbol{x}_N\}$
3: **for** $n$ from 1 to $N$ **do**
4:     $\boldsymbol{x}_n \leftarrow \arg\max_{\boldsymbol{x} \in \mathcal{D}} \frac{k_{n-1}^2(\boldsymbol{x}^\star, \boldsymbol{x})}{k_{n-1}(\boldsymbol{x}, \boldsymbol{x}) + \lambda'}$                                  {Select next point}
5:     **for** each $\boldsymbol{x}, \boldsymbol{x}' \in \mathcal{D}$ **do**
6:         Update $k_n(\boldsymbol{x}, \boldsymbol{x}') \leftarrow k_{n-1}(\boldsymbol{x}, \boldsymbol{x}') - \frac{k_{n-1}(\boldsymbol{x}, \boldsymbol{x}_n) k_{n-1}(\boldsymbol{x}_n, \boldsymbol{x}')}{k_{n-1}(\boldsymbol{x}_n, \boldsymbol{x}_n) + \lambda'}$     {Update kernel matrix}
7:     **end for**
8: **end for**

---

## H.2 FAST (EXACT) IMPLEMENTATION

The following "fast" implementation of SIFT rests on the assumption that the objective function optimized by SIFT is submodular (cf. Assumption C.1). Recall that this objective function can be expressed as $\boldsymbol{x}_{n+1} = \arg\max_{\boldsymbol{x} \in \mathcal{D}} \psi_{\boldsymbol{x}^\star}(X_n \cup \{\boldsymbol{x}\})$ where $\psi_{\boldsymbol{x}^\star}(X) = \sigma_0^2(\boldsymbol{x}^\star) - \sigma_X^2(\boldsymbol{x}^\star)$ denotes the *uncertainty reduction* about $\boldsymbol{x}^\star$ upon fine-tuning the model on data $X$.

The "trick" of the fast implementation is to use a max-heap (with $O(1)$ lookup and $O(\log K)$ insertion) to keep track of upper bounds of $\psi_{\boldsymbol{x}^\star}(X_n \cup \{\boldsymbol{x}\})$ for each $\boldsymbol{x} \in \mathcal{D}$. The upper bounds come directly from the submodularity assumption:

$$\psi_{\boldsymbol{x}^\star}(X_i \cup \{\boldsymbol{x}\}) \geq \psi_{\boldsymbol{x}^\star}(X_j \cup \{\boldsymbol{x}\}) \quad \forall j \geq i.$$

At iteration $n$, we evaluate $\psi_{\boldsymbol{x}^\star}(X_{n-1} \cup \{\boldsymbol{x}\})$ for $\boldsymbol{x}$ in max-heap order. As soon as we find a $\boldsymbol{x}$ whose re-computed upper bound is smaller than a previously re-computed upper bound, we stop the evaluation. In the worst case, one might iterate through all $K$ points in each iteration, but in practice, it can sometimes be reasonable to assume that one only needs to consider $O(1)$ points per iteration. This algorithm is known as the "lazy greedy algorithm" in submodular function maximization (Minoux, 1978) where it is typically seen to result in large speed-ups.

We summarize the fast implementation in Algorithm 2. The kernel matrix $\boldsymbol{K}$ tracks the conditional kernel matrix of the prompt $\boldsymbol{x}^\star$ and the previously selected data $X_{n-1}$. $\boldsymbol{\Lambda}$ tracks the (regularized) inverse of the kernel matrix of the previously selected data $X_{n-1}$. Whenever necessary, the cached kernel matrix and cached inverse are updated. We denote by $\boldsymbol{\Phi} \in \mathbb{R}^{(n-1) \times d}$ the matrix of embeddings of previously selected points and by $\tilde{\boldsymbol{\Phi}} \in \mathbb{R}^{n \times d}$ the same matrix extended by $\boldsymbol{\phi}(\boldsymbol{x}^\star)$ as the first row.

Initializing the max-heap takes time $\widetilde{O}(K)$ and is analogous to standard Nearest Neighbor retrieval. Additionally, SIFT-FAST performs a data selection loop for $N$ iterations where each operation takes $O(N^2)$ time requiring persistent memory of size $O(N^2)$. Notably, only the kernel matrix of the prompt and the previously selected data is kept in memory.

---

**Algorithm 2** SIFT-FAST($\lambda'$)

1: **Input:** prompt $\boldsymbol{x}^\star$, data space $\mathcal{D}$, number of points to select $N$
2: **Output:** set of selected points $\{\boldsymbol{x}_1, \ldots, \boldsymbol{x}_N\}$

    {Initializing max-heap ("Nearest Neighbor retrieval")}
3: **for** $\boldsymbol{x} \in \mathcal{D}$ **do**
4:     $\alpha_{\boldsymbol{x}} \leftarrow \frac{(\boldsymbol{\phi}(\boldsymbol{x}^\star)^\top \boldsymbol{\phi}(\boldsymbol{x}))^2}{\|\boldsymbol{\phi}(\boldsymbol{x})\|_2^2 + \lambda'}$
5:     Insert $(\boldsymbol{x}, \alpha_{\boldsymbol{x}})$ into max-heap
6: **end for**

    {Data selection}
7: Initialize $\boldsymbol{K} = \left[\|\boldsymbol{\phi}(\boldsymbol{x}^\star)\|_2^2\right]$ and $\boldsymbol{\Lambda}$ as an empty square matrix
8: **for** $n$ from 1 to $N$ **do**
9:     Initialize lower bound $\alpha^\star \leftarrow -\infty$
10:     **for** each popped $(\boldsymbol{x}, \alpha)$ in max-heap order **do**
11:         **if** $\alpha = \alpha^\star$ **then**
12:             $\boldsymbol{x}_n \leftarrow \boldsymbol{x}$                                       {$\boldsymbol{x}$ maximizes the SIFT($\lambda'$) objective}
13:             **break**
14:         **end if**
15:         $\alpha_{\boldsymbol{x}}, \boldsymbol{\Lambda}, \boldsymbol{K}' \leftarrow \text{RECOMPUTE}(\boldsymbol{x}, \boldsymbol{K}, \boldsymbol{\Lambda})$                   {Recompute objective value}
16:         $\alpha^\star \leftarrow \max\{\alpha^\star, \alpha_{\boldsymbol{x}}\}$
17:         Insert $(\boldsymbol{x}, \alpha_{\boldsymbol{x}})$ into max-heap
18:     **end for**
19:     $\boldsymbol{K} \leftarrow \text{UPDATESTATE}(\boldsymbol{x}_n, \boldsymbol{K}')$                            {Update cached kernel matrix}
20: **end for**

---

### H.3 PRE-SELECTING DATA VIA NEAREST NEIGHBOR RETRIEVAL

The reason for SIFT-FAST being so efficient is that it effectively "discards" all points in $\mathcal{D}$ that are completely irrelevant to the prompt. Whereas SIFT recomputes the objective value of every point in $\mathcal{D}$ at each iteration, SIFT-FAST only reevaluates points that are potentially relevant. An alternative to make SIFT fast is therefore simply to preemptively discard irrelevant points. In our experiments we do so by pre-selecting a subset of size $K = 200$ via Nearest Neighbor retrieval within $\mathcal{D}$ (cf. Appendix I for more details). This step aims to eliminate all points from the data space that SIFT would not end up picking anyway while retaining a diverse set of relevant points. Figure 18 shows the effect of $K$ on statistical performance and Figure 4 shows the effect on computational performance.

### H.4 FUTURE WORK: IMPROVING GPU UTILIZATION OF SIFT-FAST

In our experiments on the Pile dataset, we find that SIFT-FAST is less efficient than SIFT (cf. Figure 19 (left)). We attribute this to the fact that for any given prompt, the closest neighbors in the

---

**Algorithm 3** SIFT-FAST($\lambda'$): RECOMPUTE

---

1: **Input:** prompt $\boldsymbol{x}^{\star}$, current iteration $n$, candidate $\boldsymbol{x}$, cached kernel matrix $\boldsymbol{K}$, cached inverse $\boldsymbol{\Lambda}$
2: **Output:** objective value $\alpha_{\boldsymbol{x}}$, updated cached inverse $\boldsymbol{\Lambda}$, expanded kernel matrix $\boldsymbol{K}$

   {Expand cached kernel matrix $\boldsymbol{K}$ (if required)}
3: **if** $\boldsymbol{x}$ is has not been selected yet **then**
4:     {Update $\boldsymbol{\Lambda}$ with the Sherman-Morrison-Woodbury formula (Sherman & Morrison, 1950)}
5:     Let $i$ denote the size of $\boldsymbol{\Lambda}$
6:     **if** $i < n - 1$ **then**
7:         $\boldsymbol{A} \leftarrow \boldsymbol{\Phi}_i \boldsymbol{\Phi}_{i+1:n-1}^{\top}$
8:         $\boldsymbol{B} \leftarrow \boldsymbol{\Phi}_{i+1:n-1} \boldsymbol{\Phi}_{i+1:n-1}^{\top}$
9:         $\boldsymbol{C} \leftarrow (\boldsymbol{B} - \boldsymbol{A}^{\top} \boldsymbol{\Lambda} \boldsymbol{A})^{-1}$
10:         $\boldsymbol{\Lambda} \leftarrow \begin{bmatrix} \boldsymbol{\Lambda} + \boldsymbol{\Lambda} \boldsymbol{A} \boldsymbol{C} \boldsymbol{A}^{\top} \boldsymbol{\Lambda} & -\boldsymbol{\Lambda} \boldsymbol{A} \boldsymbol{C} \\ -\boldsymbol{C} \boldsymbol{A}^{\top} \boldsymbol{\Lambda} & \boldsymbol{C} \end{bmatrix}$
11:     **end if**

   {Expand kernel matrix $\boldsymbol{K}$}
12:     $\boldsymbol{A} \leftarrow \boldsymbol{I} - \boldsymbol{\Phi}^{\top} \boldsymbol{\Lambda} \boldsymbol{\Phi}$
13:     $\boldsymbol{k} \leftarrow \tilde{\boldsymbol{\Phi}} \boldsymbol{A} \boldsymbol{\phi}(\boldsymbol{x})$
14:     $\boldsymbol{K} \leftarrow \begin{bmatrix} \boldsymbol{K} & \boldsymbol{k} \\ \boldsymbol{k}^{\top} & \|\boldsymbol{\phi}(\boldsymbol{x})\|_{\boldsymbol{A}}^2 \end{bmatrix}$
15: **end if**
16: $\alpha_{\boldsymbol{x}} \leftarrow \frac{k^2(\boldsymbol{x}^{\star}, \boldsymbol{x})}{k(\boldsymbol{x}, \boldsymbol{x}) + \lambda'}$          {Compute objective value using the relation from Equation (9)}

---

**Algorithm 4** SIFT-FAST($\lambda'$): UPDATESTATE

---

1: **Input:** selected point $\boldsymbol{x}_n$, expanded kernel matrix $\boldsymbol{K}'$
2: **Output:** new conditional kernel matrix $\boldsymbol{K}$

   {Update kernel matrix using the relation from Equation (9)}
3: **for** each $\boldsymbol{x}, \boldsymbol{x}' \in \{\boldsymbol{x}^{\star}\} \cup X_n$ **do**
4:     Update $k(\boldsymbol{x}, \boldsymbol{x}') \leftarrow k'(\boldsymbol{x}, \boldsymbol{x}') - \frac{k'(\boldsymbol{x}, \boldsymbol{x}_n) k'(\boldsymbol{x}_n, \boldsymbol{x}')}{k'(\boldsymbol{x}_n, \boldsymbol{x}_n) + \lambda'}$
5: **end for**

---

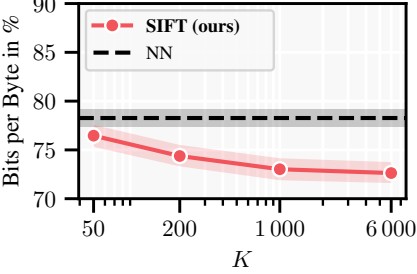

Figure 18: We run SIFT ($\lambda' = 1$) with various values of $K$ and report the bits per byte ($\downarrow$ better) after 50 test-time iterations. We find that performance on the Pile plateaus after $K = 1\text{'}000$. Even at $K = 50$, which equals the number of points selected, SIFT outperforms Nearest Neighbor retrieval due to being able to select the same points multiple times.

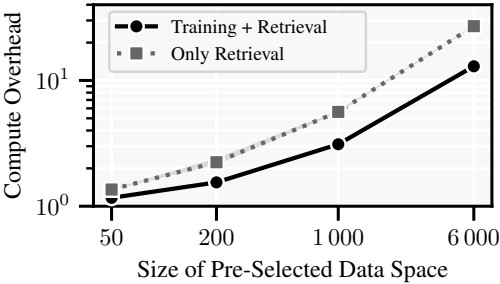 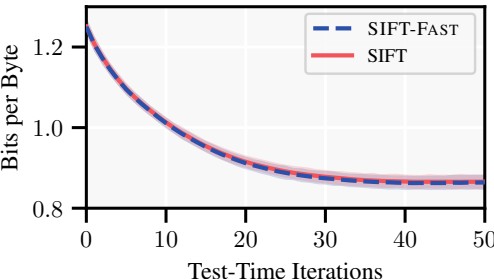

Figure 19: **Left:** Computational overhead of SIFT-FAST over Nearest Neighbor retrieval. This overhead is larger than the overhead of SIFT over Nearest Neighbor retrieval (cf. Figure 4). **Right:** SIFT-FAST achieves identical statistical performance to SIFT, which is further evidence that Assumption C.1 is satisfied in our language modeling setting.

data space are all relatively similar to the prompt (cf. Figure 20), meaning that each iteration of SIFT-FAST has to loop (sequentially) over the entire priority queue. In contrast, SIFT performs this operation in parallel on a GPU.

We believe that a promising computational approach is to combine the advantages of the SIFT and SIFT-FAST implementations. This could be achieved by keeping a large sub-selected kernel matrix on the GPU (akin to the SIFT implementation) and selectively using the SIFT-FAST implementation if points on the priority queue that are not in the sub-selected kernel matrix may be selected. This would allow for a more efficient use of the GPU memory of SIFT-FAST, which we expect to yield comparable computational performance to the SIFT implementation in most cases, while still being able to handle large data spaces.

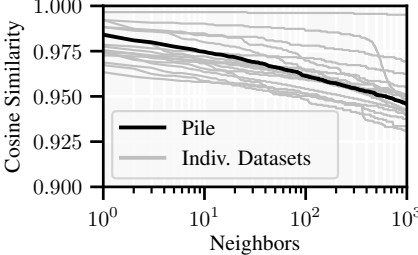

Figure 20: Average cosine similarities of test prompts to closest 1'000 neighbors in the data space of the Pile; with the Roberta embedding model.

# I    EXPERIMENT DETAILS

We fine-tune the pre-trained model for a single gradient step each on $N = 50$ selected data points. We evaluate the performance on 1% of the test instances of the Pile. We use the Pile training dataset as data space for data selection, which notably does *not* include data from the validation and test sets.

**Evaluation**    We use the standard implementation of the `lm-evaluation-harness` library (Gao et al., 2024) for computing the bits per byte. This implementation computes the log-likelihood of a document using a rolling-window approach, which ensures that the model's maximum context window is fully utilized.

**Truncation of Long Sequences**    Analogously to Hardt & Sun (2024), to generate embeddings, we naively truncate long sequences to the maximum sequence length of the embedding model, that is, we only consider the prefixes of long sequences for data selection.

**Learning Rate and Optimizer**    Following Hardt & Sun (2024), we use the Adam optimizer (Kingma & Ba, 2014) with $\epsilon$-value 1e−8. We use the default learning rate 5e−5 of the `transformers` library (Wolf et al., 2020) unless noted otherwise. Hardt & Sun (2024) used a learning rate of 2e−5 for their experiments. We show in Figure 24 that 5e−5 leads to strictly better performance of the Nearest Neighbor baseline. In our ablation study over metrics for Nearest Neighbor retrieval (cf. Figure 11), which was conducted concurrently, we still used learning rate 2e−5 of Hardt & Sun (2024).

**Low-Rank Adaptation (LoRA)**    We use LoRA (Hu et al., 2022) for fine-tuning Phi-3, and also evaluate the performance of LoRA with GPT-2 and GPT-2-large (cf. §E). We use LoRAs with rank 64, output scaling 16, without dropout and bias. When fine-tuning with LoRA, we use the learning rate 5e−4.

**Gradient Checkpointing**    We additionally use gradient checkpointing (Chen et al., 2016) for fine-tuning Phi-3 to reduce memory footprint and allow fine-tuning on our hardware.

**Uncopyrighted Pile Dataset**    We use only those datasets of the Pile where our use is in compliance with the terms of service of the data host (Gao et al., 2020). This excludes the Books3, BookCorpus2, OpenSubtitles, YTSubtitles, and OWT2 datasets.

We provide an overview of all hyperparameters of test-time fine-tuning in Table 9.

| Model family | GPT-2 | Phi-3 | Llama-3.2 |
|---|---|---|---|
| $\lambda'$ | 0.01 | 0.01 | 0.01 |
| Learning rate | 5e−5 | 5e−4 | 1e−4 |
| Adam's $\epsilon$-value | 1e−8 | 1e−8 | 1e−8 |
| Max. sequence length (in tokens) | 1024 | 4096 | 4096 |
| LoRA | no | yes | yes |
| Gradient checkpointing | no | yes | yes |

Table 9: Hyperparameters during test-time fine-tuning, unless noted otherwise.

## I.1 PROPERTIES OF THE PILE DATASET

Figure 21 shows the average cosine similarities of test prompts to neighbors in the data space of the Pile. Table 10 shows the weight of each dataset in the Pile.

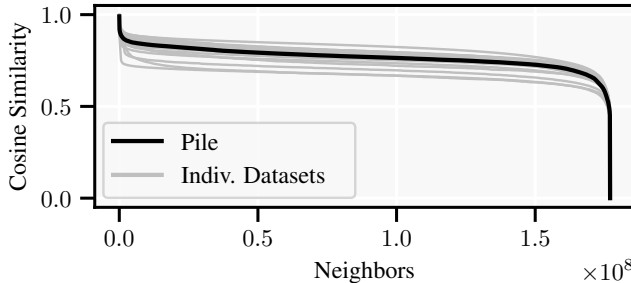

Figure 21: Average cosine similarities of test prompts to neighbors in the data space of the Pile; with the Roberta embedding model.

|  | Weight |
| --- | --- |
| Common Crawl | 24.14% |
| PubMed Central | 19.19% |
| ArXiv | 11.94% |
| GitHub | 10.12% |
| FreeLaw | 8.18% |
| Stack Exchange | 6.84% |
| US Patents | 4.87% |
| PubMed Abstracts | 4.09% |
| Project Gutenberg | 2.89% |
| Wikipedia | 2.04% |
| DeepMind Math | 1.65% |
| Ubuntu IRC | 1.17% |
| EuroParl | 0.97% |
| Hacker News | 0.83% |
| PhilPapers | 0.51% |
| NIH ExPorter Grants | 0.40% |
| Enron Emails | 0.19% |

Table 10: Overview of datasets in the (uncopyrighted) Pile. Weight is the percentage of bytes in the final dataset occupied by each dataset. Numbers are taken from Gao et al. (2020) and renormalized.

## I.2 IN-CONTEXT BASELINE

In our evaluation of in-context learning, we use the following format to insert the selected data into the context of the model: We separate all retrieved token sequences with the string "\n\n" which can be seen as a paragraph separator, and additionally add this string between the data string and the prompt.

Notably, our results with in-context learning on GPT-2-large outperform the results previously reported by Hardt & Sun (2024). We suspect that this is due to a combination of a more reasonable evaluation and using SIFT as opposed to Nearest Neighbor retrieval for data selection.

**Evaluation of Inference Cost of In-Context Baseline** We estimate the inference cost of in-context learning as follows. We evaluate the time it takes compute the rolling log-likelihood of the test instance with context included and subtract the time it takes to compute the rolling log-likelihood of

the test instance without context. This is a lower-bound of the inference cost of in-context learning, as unlike autoregressive generation, computing the log-likelihood is partially parallelized.

To compute the token throughput of the in-context baseline, we divide the total compute time by the number of tokens added to the context.

### I.3 INFERENCE COST WITH TEST-TIME FINE-TUNING

Figure 22 evaluates the inference cost of test-time fine-tuning on all the Pile and the largest datasets.

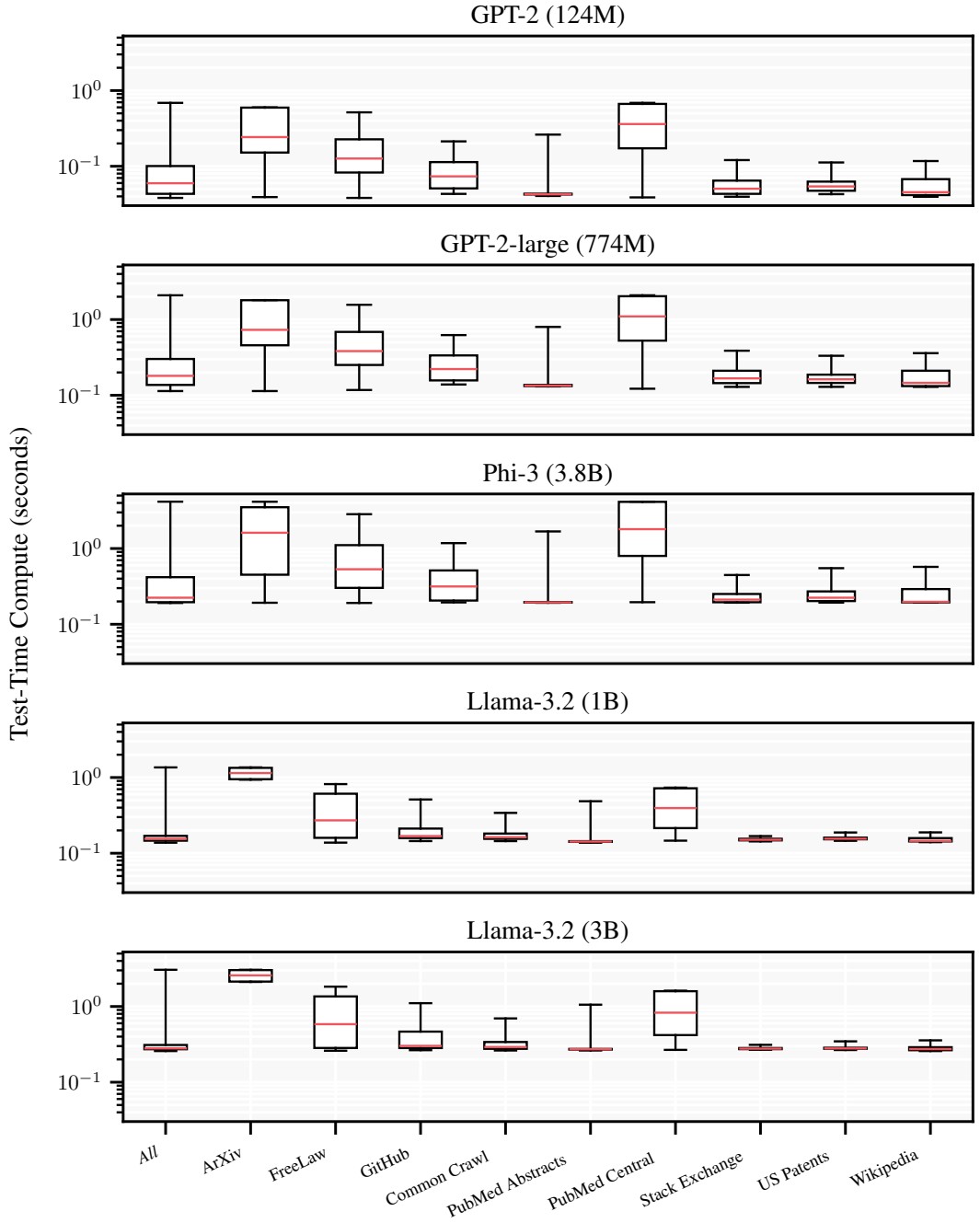

Figure 22: Cost of taking a single gradient step. Results are with an NVIDIA GH200.

## J    ABLATIONS

This section summarizes ablations that we conducted to investigate test-time fine-tuning and SIFT.

- **Hyperparameter** $\lambda'$: Table 11
- **Learning Curves for Individual Datasets of the Pile**: Figure 23
- **Learning Rate**: Figure 24
- **Uncertainty Estimation**:
    - Summary of correlations (Table 12)
    - Visualization of $\sigma_n$ (Figure 25)
- **Compute-proportional Performance Gain**:
    - Details on ADAPTIVE SIFT (Figure 26)

| | 1e−12 | 1e−8 | 1e−4 | 0.01 | 0.1 | 1 | 10 | 100 | 10'000 | NN | NN-F | Δ |
|---|---|---|---|---|---|---|---|---|---|---|---|---|
| NIH Grants | 123.9 (6.9) | 79.0 (6.4) | 70.2 (6.7) | 53.8 (8.9) | 52.9 (9.0) | 53.3 (9.1) | 54.2 (9.1) | 64.5 (10.9) | 93.5 (16.9) | 84.9 (2.1) | 91.6 (16.7) | ↓32.0 |
| US Patents | 119.9 (3.9) | 82.9 (2.7) | 70.2 (3.1) | 62.9 (3.5) | 62.2 (3.6) | 62.7 (3.7) | 63.2 (3.7) | 72.9 (4.2) | 105.4 (6.4) | 80.3 (1.9) | 108.8 (6.6) | ↓18.1 |
| GitHub | 54.6 (3.1) | 41.4 (2.2) | 35.9 (2.3) | 30.0 (2.2) | 28.6 (2.2) | 28.6 (2.2) | 29.2 (2.2) | 36.1 (2.6) | 51.3 (4.0) | 42.1 (2.0) | 53.2 (4.0) | ↓13.5 |
| Enron Emails | 87.1 (16.5) | 68.6 (9.4) | 63.1 (9.1) | 53.1 (11.4) | 52.4 (11.8) | 53.8 (12.2) | 54.1 (12.2) | 59.6 (13.4) | 89.4 (20.4) | 64.4 (10.1) | 91.6 (20.6) | ↓12.0 |
| Common Crawl | 117.9 (1.3) | 91.0 (0.5) | 90.7 (0.5) | 87.5 (0.7) | 86.1 (0.9) | 87.8 (0.9) | 88.3 (0.9) | 99.3 (1.0) | 146.2 (1.6) | 90.4 (0.5) | 148.8 (1.5) | ↓4.3 |
| ArXiv | 145.9 (7.0) | 83.5 (1.3) | 83.6 (1.3) | 82.5 (1.4) | 81.6 (1.9) | 81.2 (1.8) | 82.8 (1.9) | 94.6 (2.8) | 158.0 (6.1) | 85.0 (1.6) | 166.8 (6.4) | ↓3.8 |
| Wikipedia | 104.2 (3.0) | 64.9 (2.1) | 63.9 (2.2) | 62.7 (2.1) | 63.7 (2.1) | 64.8 (2.2) | 65.6 (2.3) | 77.5 (2.5) | 118.1 (3.7) | 66.3 (2.0) | 121.2 (3.5) | ↓3.6 |
| PubMed Abstr. | 132.3 (1.6) | 87.0 (0.4) | 87.0 (0.4) | 84.4 (0.6) | 84.8 (0.7) | 86.4 (0.7) | 86.7 (0.7) | 102.0 (0.9) | 158.9 (1.4) | 87.2 (0.4) | 162.6 (1.3) | ↓2.8 |
| PubMed Central | 131.9 (4.9) | 80.5 (2.5) | 80.0 (2.7) | 79.5 (2.6) | 80.6 (2.7) | 82.0 (2.7) | 83.8 (2.9) | 98.6 (3.7) | 151.6 (5.5) | 81.7 (2.6) | 155.6 (5.1) | ↓2.2 |
| Stack Exchange | 118.0 (1.7) | 77.6 (0.7) | 77.6 (0.7) | 76.7 (0.7) | 77.0 (0.7) | 77.8 (0.7) | 78.1 (0.7) | 85.9 (0.9) | 136.9 (1.6) | 78.2 (0.7) | 141.9 (1.5) | ↓1.5 |
| Hacker News | 113.9 (7.2) | 78.8 (2.7) | 78.9 (2.7) | 78.4 (2.8) | 77.8 (3.5) | 78.1 (3.6) | 78.4 (3.6) | 86.2 (3.3) | 131.3 (6.2) | 79.2 (2.8) | 133.1 (6.3) | ↓1.4 |
| DeepMind Math | 104.7 (6.2) | 69.3 (2.1) | 69.1 (2.1) | 69.7 (2.1) | 70.1 (2.1) | 69.0 (2.0) | 70.1 (2.1) | 71.9 (2.2) | 103.5 (5.6) | 69.6 (2.1) | 121.8 (3.1) | ↓0.6 |
| FreeLaw | 102.5 (6.3) | 64.0 (3.9) | 63.5 (4.0) | 64.0 (4.1) | 65.5 (4.2) | 65.7 (4.1) | 67.0 (4.2) | 80.3 (5.0) | 114.1 (7.1) | 64.1 (4.0) | 122.4 (7.1) | ↓0.6 |
| All | 112.9 (0.9) | 78.5 (0.6) | 76.7 (0.6) | 73.5 (0.6) | 73.2 (0.7) | 74.3 (0.7) | 74.9 (0.7) | 85.6 (0.8) | 129.8 (1.2) | 78.3 (0.5) | 133.3 (1.2) | ↓5.4 |

Table 11: Percentage of bits per byte after 50 test-time iterations for varying $\lambda'$, relative to the bits per byte of the base model. We only include datasets with at least 10 examples in our test set. **Bold** numbers denote the best performing selected subset. Underlined numbers denote better or on-par performance with Nearest Neighbor retrieval. Δ denotes the performance gain of SIFT with the strongest $\lambda'$ *per dataset* over Nearest Neighbor retrieval. Numbers in parentheses are standard errors. We remark that $\lambda'$ is on a logarithmic scale. For any choice of $\lambda' \in [1e{-}8, 10]$, SIFT *always* performs at least on-par with Nearest Neighbor retrieval.

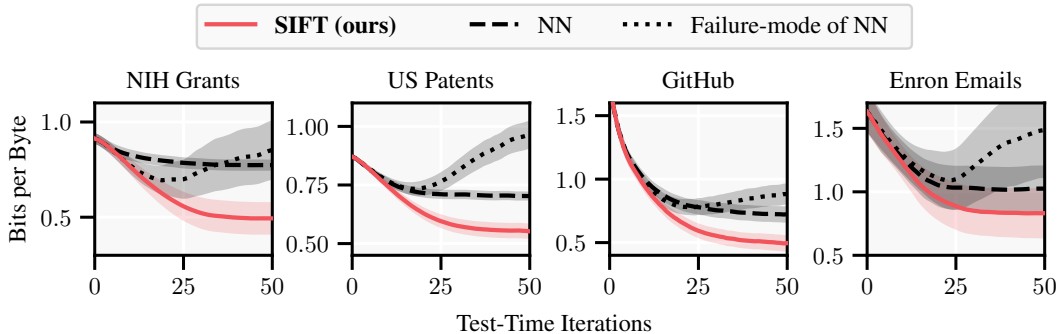

Figure 23: Performance in some of the datasets of the Pile, with GPT-2 as base model. Error bars correspond to standard errors.

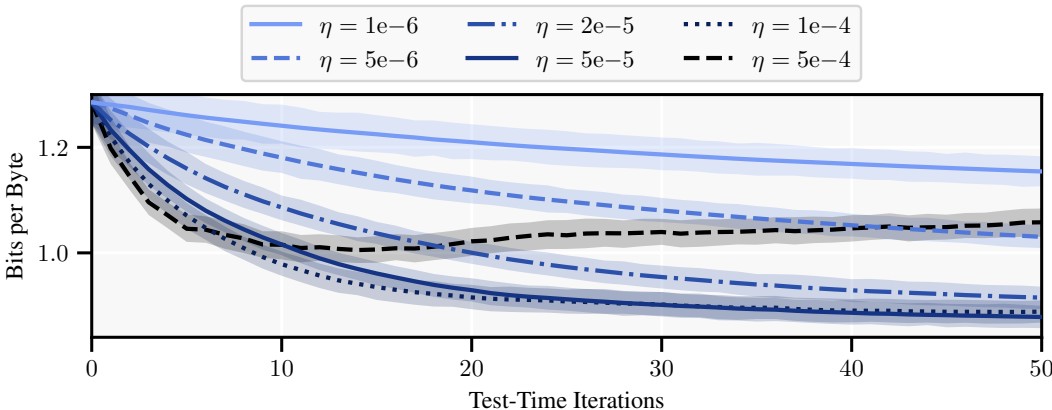

Figure 24: Ablation of the learning rate with data selected by Nearest Neighbor retrieval. We find that the default learning rate $5e{-}5$ of the `transformers` library (Wolf et al., 2020) works best, and conduct our other experiments with this learning rate unless noted otherwise. Hardt & Sun (2024) had previously used $2e{-}5$ which we find to be suboptimal.

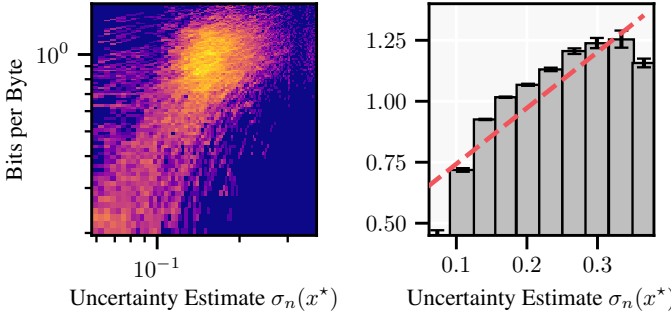

Figure 25: We visualize the predictive ability of the uncertainty estimates $\sigma_n$ analogously to Figure 8.

|  |  | Spearman | Pearson |
|---|---|---|---|
| $\sigma_n$ | all steps | 0.485 | 0.421 |
|  | final step | 0.496 | 0.443 |
| $\hat{\sigma}_n$ | all steps | 0.722 | 0.581 |
|  | final step | 0.682 | 0.482 |
| $\log \sigma_n$ | all steps | 0.485 | 0.468 |
|  | final step | 0.496 | 0.466 |
| $\log \hat{\sigma}_n$ | all steps | 0.722 | 0.618 |
|  | final step | 0.682 | 0.526 |

Table 12: We find a strong / moderate correlation between the uncertainty estimates $\hat{\sigma}_n$ / $\sigma_n$ and bits per byte. We further consider the correlation at all test-time iterations (from 0 to 50) as well as only at the final iteration. We report both the Spearman and Pearson correlation coefficients, measuring monotonic and linear relationships, respectively. Before determining the Pearson correlation, we exclude the 0.25% of the data points with the lowest and highest uncertainty estimates to avoid the influence of outliers. The p-value of all correlations is below $1e-5$ due to the large sample size.

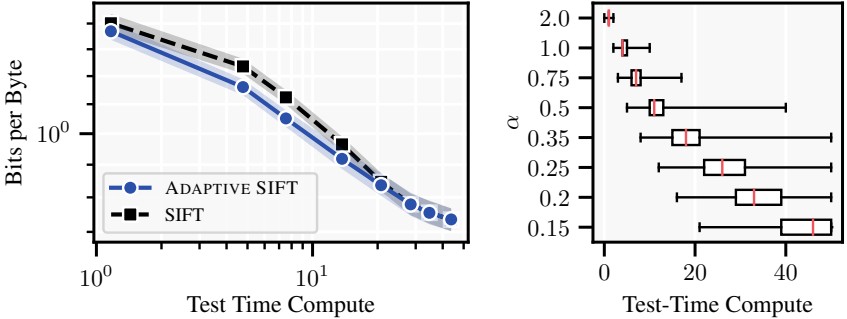

Figure 26: We evaluate ADAPTIVE SIFT with the same choices of $\alpha$ as in Figure 8 (right). **Left:** Bits per byte of ADAPTIVE SIFT ($\downarrow$ better) against test-time compute. Every marker corresponds to the performance of ADAPTIVE SIFT with a given $\alpha$, where the associated test-time compute is the average number of test-time iterations on prompts. We compare ADAPTIVE SIFT to SIFT, where we spend the same test-time compute on every prompt. We see a slight advantage of ADAPTIVE SIFT over SIFT, due to adaptively stopping depending on the prompt. Our current experiment exhibits a bias as test-time compute approaches 50, since we force-stop the compute at 50 iterations. This biases ADAPTIVE SIFT to perform similarly to SIFT. We hypothesize that the initial advantage of ADAPTIVE SIFT over SIFT may grow with more test-time compute if compute is not force-stopped at 50 iterations. **Right:** Frequency of stopping at a given iteration for given values of $\alpha$.

# K   PROOFS

This section provides the formal proofs of the results presented in the main text.

- §K.2 proves the insufficiency of Nearest Neighbor retrieval (Informal Proposition 2.1).
- §K.3 shows the close relationship of regularized loss minimization and test-time fine-tuning (Proposition 3.3).
- §K.4 details how SIFT balances relevance and diversity (§C.1).
- §K.5 states confidence sets for fine-tuning regression models that are analogous to the confidence sets for classification from the main text.
- §K.6 derives the confidence sets from the main text (Theorem 3.2).

## K.1   NOTATION

Throughout this work, $\log$ denotes the natural logarithm. Unless noted otherwise $\{\ldots\}$ denotes a multiset. We define the feature map $\boldsymbol{\Phi}_n \doteq (\boldsymbol{\phi}(\boldsymbol{x}_1), \ldots, \boldsymbol{\phi}(\boldsymbol{x}_n)) \in \mathbb{R}^{n \times d}$, which gives rise to the kernel matrix $\boldsymbol{K}_n \doteq \boldsymbol{K}_{X_n} = \boldsymbol{\Phi}_n \boldsymbol{\Phi}_n^\top \in \mathbb{R}^{n \times n}$ and the covariance operator $\boldsymbol{\Sigma}_n \doteq \boldsymbol{\Phi}_n^\top \boldsymbol{\Phi}_n \in \mathbb{R}^{d \times d}$.

## K.2   INSUFFICIENCY OF NEAREST NEIGHBOR RETRIEVAL (INFORMAL PROPOSITION 2.1)

We refer to §C.2 for the formal definition of the irreducible uncertainty $\sigma_\infty(\boldsymbol{x}^\star; \mathcal{D})$.

We remark that if embeddings are of unit length, the cosine similarity scoring function is equivalent to the (negative) Euclidean distance scoring function:

$$\|\boldsymbol{x}^\star - \boldsymbol{x}\|_2^2 = (\boldsymbol{x}^\star - \boldsymbol{x})^\top (\boldsymbol{x}^\star - \boldsymbol{x}) = \|\boldsymbol{x}^\star\|_2^2 + \|\boldsymbol{x}\|_2^2 - 2\boldsymbol{x}^{\star\top} \boldsymbol{x} = 2 - 2\cos(\boldsymbol{x}^\star, \boldsymbol{x}).$$

We henceforth consider the Euclidean distance scoring function.

**Proposition K.1** (Insufficiency of Nearest Neighbor Retrieval). *Suppose w.l.o.g. that $\boldsymbol{\phi}(\boldsymbol{x}) = \boldsymbol{x}$. Consider the data space $\mathcal{D} = \bigcup_{i=1}^d \mathcal{D}_i$ where $\mathcal{D}_i = \{\boldsymbol{e}_i \mid j \in \mathbb{N}\}$ with $\boldsymbol{e}_i$ the $i$-th basis vector of $\mathbb{R}^d$.[14] Let $\boldsymbol{x}^\star = \frac{1}{\sqrt{4+(d-1)}}(2, 1, 1, \ldots, 1) \in \mathbb{R}^d$.*

*Then, for all $n \geq 1$:*

1. *If $X_n$ are the $n$ nearest neighbors of $\boldsymbol{x}^\star$ in $\mathcal{D}$, $\sigma_n^2(\boldsymbol{x}^\star) \geq \sigma_\infty^2(\boldsymbol{x}^\star; \mathcal{D}_1) \gg 0$.*

2. *If $X_n$ is selected by SIFT, $\sigma_n^2(\boldsymbol{x}^\star) \xrightarrow{n \to \infty} \sigma_\infty^2(\boldsymbol{x}^\star; \mathcal{D}) = 0$.*

*Proof.*

1. Clearly, $\|\boldsymbol{x}^\star - \boldsymbol{e}_1\|_2^2 < \|\boldsymbol{x}^\star - \boldsymbol{e}_i\|_2^2$ for all $i > 1$. Hence, $X_n = \{\boldsymbol{e}_1 \mid i \in [n]\} \subset \mathcal{D}_1$. This is as if the data space was restricted to $\mathcal{D}_1$, and hence $\sigma_n^2(\boldsymbol{x}^\star) \geq \sigma_\infty^2(\boldsymbol{x}^\star; \mathcal{D}_1)$.

2. This follows readily from Theorem C.2 and noting that $\operatorname{span}\mathcal{D} = \mathbb{R}^d$, implying $\sigma_\infty^2(\boldsymbol{x}^\star; \mathcal{D}) = 0$.

$\square$

**Discussion**   The setting examined in Proposition K.1 is an extreme case (where data exists with exact duplication), yet we deem that it illustrates a realistic scenario. Particularly nowadays that similar information is accessible from many sources in different forms, it is crucial to explicitly select diverse data for fine-tuning. We show here theoretically and in Appendix L.1 qualitatively that SIFT does not have this limitation.

---

[14]We remark that $\{\ldots\}$ denotes a multiset.

### K.3 THE CLOSE RELATIONSHIP OF REGULARIZED LOSS MINIMIZATION AND TEST-TIME FINE-TUNING (PROPOSITION 3.3)

*Proof.* We note that the regularized negative log-likelihood loss $\mathcal{L}^\lambda$ from Equation (1),

$$\mathcal{L}^\lambda(\boldsymbol{W}; D) = \underbrace{-\sum_{(\boldsymbol{x},y) \in D} \log s_y(\boldsymbol{W}\boldsymbol{\phi}(\boldsymbol{x}))}_{\mathcal{L}(\boldsymbol{W}; D)} + \frac{\lambda}{2}\|\boldsymbol{W} - \boldsymbol{W}^{\text{pre}}\|_{\text{F}}^2,$$

is strictly convex in $\boldsymbol{W}$ and has a unique minimizer $\boldsymbol{W}_\lambda$ which satisfies

$$\boldsymbol{\nabla}\mathcal{L}^\lambda(\boldsymbol{W}_\lambda; D) = \boldsymbol{\nabla}\mathcal{L}(\boldsymbol{W}_\lambda; D) + \lambda(\boldsymbol{W}_\lambda - \boldsymbol{W}^{\text{pre}}) = \boldsymbol{0}.$$

It follows that $\boldsymbol{W}_\lambda = \boldsymbol{W}^{\text{pre}} - \frac{1}{\lambda}\boldsymbol{\nabla}\mathcal{L}(\boldsymbol{W}_\lambda; D)$.

Finally, recall that $\widehat{\boldsymbol{W}}_\eta = \boldsymbol{W}^{\text{pre}} - \eta\boldsymbol{\nabla}\mathcal{L}(\boldsymbol{W}^{\text{pre}}; D)$. We obtain

$$\|\boldsymbol{W}_{1/\eta} - \widehat{\boldsymbol{W}}_\eta\|_{\text{F}} = \|\eta\boldsymbol{\nabla}\mathcal{L}(\boldsymbol{W}^{\text{pre}}; D) - \eta\boldsymbol{\nabla}\mathcal{L}(\boldsymbol{W}_{1/\eta}; D)\|_{\text{F}}$$
$$= \eta\|\boldsymbol{\nabla}\mathcal{L}(\boldsymbol{W}^{\text{pre}}; D) - \boldsymbol{\nabla}\mathcal{L}(\boldsymbol{W}_{1/\eta}; D)\|_{\text{F}}.$$

$\square$

### K.4 HOW SIFT BALANCES RELEVANCE AND DIVERSITY

**1st point** For non-unit length embeddings, the first selected point can be expressed as follows:

$$\boldsymbol{x}_1 = \arg\min_{\boldsymbol{x} \in \mathcal{D}} \sigma_{\{\boldsymbol{x}\}}^2(\boldsymbol{x}^\star) = \arg\max_{\boldsymbol{x} \in \mathcal{D}} \frac{(\boldsymbol{\phi}(\boldsymbol{x}^\star)^\top \boldsymbol{\phi}(\boldsymbol{x}))^2}{\|\boldsymbol{\phi}(\boldsymbol{x})\|_2^2 + \lambda'} = \arg\max_{\boldsymbol{x} \in \mathcal{D}} \begin{cases} \measuredangle_\phi(\boldsymbol{x}^\star, \boldsymbol{x})^2 & \text{as } \lambda' \to 0 \\ (\boldsymbol{\phi}(\boldsymbol{x}^\star)^\top \boldsymbol{\phi}(\boldsymbol{x}))^2 & \text{as } \lambda' \to \infty. \end{cases}$$

**2nd point** Next, we consider the second selected point. We derive the results in terms of the dot product kernel $k(\boldsymbol{x}, \boldsymbol{x}') = \boldsymbol{\phi}(\boldsymbol{x})^\top \boldsymbol{x}'$ which is such that $k(\boldsymbol{x}, \boldsymbol{x}') = \measuredangle_\phi(\boldsymbol{x}, \boldsymbol{x}')$ for unit length embeddings. Let $\boldsymbol{x}$ be such that $k(\boldsymbol{x}_1, \boldsymbol{x}) = 0$. We have

$$\psi_{\boldsymbol{x}^\star}(\{\boldsymbol{x}_1, \boldsymbol{x}_1\}) = \begin{bmatrix} k(\boldsymbol{x}^\star, \boldsymbol{x}_1) \\ k(\boldsymbol{x}^\star, \boldsymbol{x}_1) \end{bmatrix}^\top \begin{bmatrix} 1+\lambda' & 1 \\ 1 & 1+\lambda' \end{bmatrix}^{-1} \begin{bmatrix} k(\boldsymbol{x}^\star, \boldsymbol{x}_1) \\ k(\boldsymbol{x}^\star, \boldsymbol{x}_1) \end{bmatrix}$$
$$= \frac{1}{(1+\lambda')^2 - 1} \begin{bmatrix} k(\boldsymbol{x}^\star, \boldsymbol{x}_1) \\ k(\boldsymbol{x}^\star, \boldsymbol{x}_1) \end{bmatrix}^\top \begin{bmatrix} 1+\lambda' & -1 \\ -1 & 1+\lambda' \end{bmatrix} \begin{bmatrix} k(\boldsymbol{x}^\star, \boldsymbol{x}_1) \\ k(\boldsymbol{x}^\star, \boldsymbol{x}_1) \end{bmatrix}$$
$$= \frac{2\lambda' k(\boldsymbol{x}^\star, \boldsymbol{x}_1)^2}{(1+\lambda')^2 - 1}$$
$$= \frac{2k(\boldsymbol{x}^\star, \boldsymbol{x}_1)^2}{2+\lambda'}.$$

For $\boldsymbol{x}$, we have

$$\psi_{\boldsymbol{x}^\star}(\{\boldsymbol{x}_1, \boldsymbol{x}\}) = \begin{bmatrix} k(\boldsymbol{x}^\star, \boldsymbol{x}_1) \\ k(\boldsymbol{x}^\star, \boldsymbol{x}) \end{bmatrix}^\top \begin{bmatrix} 1+\lambda' & 0 \\ 0 & 1+\lambda' \end{bmatrix}^{-1} \begin{bmatrix} k(\boldsymbol{x}^\star, \boldsymbol{x}_1) \\ k(\boldsymbol{x}^\star, \boldsymbol{x}) \end{bmatrix}$$
$$= \frac{1}{(1+\lambda')^2} \begin{bmatrix} k(\boldsymbol{x}^\star, \boldsymbol{x}_1) \\ k(\boldsymbol{x}^\star, \boldsymbol{x}) \end{bmatrix}^\top \begin{bmatrix} 1+\lambda' & 0 \\ 0 & 1+\lambda' \end{bmatrix} \begin{bmatrix} k(\boldsymbol{x}^\star, \boldsymbol{x}_1) \\ k(\boldsymbol{x}^\star, \boldsymbol{x}) \end{bmatrix}$$
$$= \frac{k(\boldsymbol{x}^\star, \boldsymbol{x}_1)^2 + k(\boldsymbol{x}^\star, \boldsymbol{x})^2}{1+\lambda'}.$$

We see that $\boldsymbol{x}$ is preferred over $\boldsymbol{x}^\star$ if and only if

$$\frac{k(\boldsymbol{x}^\star, \boldsymbol{x}_1)^2 + k(\boldsymbol{x}^\star, \boldsymbol{x})^2}{1+\lambda'} > \frac{2k(\boldsymbol{x}^\star, \boldsymbol{x}_1)^2}{2+\lambda'} \iff k(\boldsymbol{x}^\star, \boldsymbol{x})^2 > \underbrace{\frac{\lambda'}{2+\lambda'}}_{c(\lambda')} k(\boldsymbol{x}^\star, \boldsymbol{x}_1)^2.$$

As $\lambda' \to \infty$, $c(\lambda') \to 1$; whereas as $\lambda' \to 0$, $c(\lambda') \to 0$.

We interpret the expressions extensively in Section 4.

## K.5 CONFIDENCE SETS FOR REGRESSION

Before moving on to deriving confidence sets for the setting with categorical feedback, we state analogous results for the regression setting under the following standard assumptions. For ease of notation, we consider the scalar case.

**Assumption K.2** (Linear function in a known latent space). We assume $f^\star(\boldsymbol{x}) = \boldsymbol{\phi}(\boldsymbol{x})^\top \boldsymbol{w}^\star$ with $\boldsymbol{w}^\star \in \mathbb{R}^d$ and where $\boldsymbol{\phi}(\cdot) \in \mathbb{R}^d$ denotes known embeddings. We assume that $\boldsymbol{w}^\star$ has bounded norm, i.e., $\|\boldsymbol{w}^\star - \boldsymbol{w}^{\mathrm{pre}}\|_2 \le B$ for some finite $B \in \mathbb{R}$.

**Assumption K.3** (Sub-Gaussian Noise). We assume that the data follows

$$y_n = f^\star(\boldsymbol{x}_n) + \varepsilon_n$$

where each $\varepsilon_n$ from the noise sequence $\{\varepsilon_n\}_{n=1}^\infty$ is conditionally zero-mean $\rho$-sub-Gaussian with known constant $\rho > 0$. Formally,

$$\forall n \ge 1, \lambda \in \mathbb{R}: \quad \mathbb{E}\big[e^{\lambda \epsilon_n} \mid D_{n-1}\big] \le \exp\left(\frac{\lambda^2 \rho^2}{2}\right)$$

where $D_{n-1}$ corresponds to the $\sigma$-algebra generated by the random variables $\{\boldsymbol{x}_i, \epsilon_i\}_{i=1}^{n-1}$ and $\boldsymbol{x}_n$.

We consider the standard squared loss $\mathcal{L}(\boldsymbol{w}; D) \doteq \frac{1}{2} \sum_{(\boldsymbol{x},y) \in D} (f(\boldsymbol{x}; \boldsymbol{w}) - y)^2$ where we write $f(\boldsymbol{x}; \boldsymbol{w}) \doteq \boldsymbol{\phi}(\boldsymbol{w})^\top \boldsymbol{w}$. The regularized loss with minimizer $\boldsymbol{w}_n$ is then

$$\mathcal{L}^\lambda(\boldsymbol{w}; D_n) \doteq \mathcal{L}(\boldsymbol{w}; D_n) + \frac{\lambda}{2} \|f - \boldsymbol{w}^{\mathrm{pre}}\|_2^2 \tag{10}$$

where $\lambda > 0$ is the regularization parameter. In the following, we write $f_n(\boldsymbol{x}) \doteq f(\boldsymbol{x}; \boldsymbol{w}_n)$ and $f^{\mathrm{pre}}(\boldsymbol{x}) \doteq f(\boldsymbol{x}; \boldsymbol{w}^{\mathrm{pre}})$. The closed-form solution to the optimization problem from Equation (10) is well-known (see, e.g., Williams & Rasmussen, 2006, Section 6.2.2) to be

$$f_n(\boldsymbol{x}) = f^{\mathrm{pre}}(\boldsymbol{x}) + \boldsymbol{k}_{X_n}^\top(\boldsymbol{x})(\boldsymbol{K}_{X_n} + \lambda \boldsymbol{I}_n)^{-1}(\boldsymbol{y}_n - \boldsymbol{f}_n^{\mathrm{pre}})$$

where $\boldsymbol{f}_n^{\mathrm{pre}}$ is the vector of predictions of $f^{\mathrm{pre}}$ at $X_n$ and $\boldsymbol{y}_n$ is the vector of observations in $D_n$.

The below result is an almost immediate consequence of the results of Abbasi-Yadkori (2013) and Chowdhury & Gopalan (2017).

**Theorem K.4** (Confidence Sets for Regression). *Pick $\delta \in (0,1)$ and let Assumptions K.2 and K.3 hold. Let*

$$\beta_n(\delta) \doteq B + \rho\sqrt{2(\gamma_n + 1 + \log(1/\delta))}$$

*where $\gamma_n \doteq \max_{\boldsymbol{x}_1,\dots,\boldsymbol{x}_n} \frac{1}{2} \log \det\left(\boldsymbol{I}_n + \lambda^{-1}\boldsymbol{K}_{X_n}\right)$. Then*

$$\mathbb{P}(\forall n \ge 1, \boldsymbol{x} \in \mathcal{X}: |f^\star(\boldsymbol{x}) - f_n(\boldsymbol{x})| \le \beta_n(\delta)\sigma_n(\boldsymbol{x})) \ge 1 - \delta.$$

*Proof.* Let us define the *residual* of the ground truth and pre-trained model as $\tilde{f}^\star(\boldsymbol{x}) \doteq f^\star(\boldsymbol{x}) - f^{\mathrm{pre}}(\boldsymbol{x})$ with corresponding weight vector $\tilde{\boldsymbol{w}}$. Analogously, let $\tilde{y}_n = \tilde{f}^\star(\boldsymbol{x}_n) + \varepsilon_n$ be the observed error. We have that $\tilde{\boldsymbol{w}}^\star \doteq \boldsymbol{w}^\star - \boldsymbol{w}^{\mathrm{pre}} \in \mathbb{R}^d$ with norm $\|\boldsymbol{w}^\star - \boldsymbol{w}^{\mathrm{pre}}\|_k$. The unbiased estimate of the remaining error is

$$\tilde{f}_n = \boldsymbol{k}_{X_n}^\top(\boldsymbol{x})(\boldsymbol{K}_{X_n} + \lambda \boldsymbol{I}_n)^{-1}\tilde{\boldsymbol{y}}_n.$$

By Theorem 2 of Chowdhury & Gopalan (2017), for all $\boldsymbol{x} \in \mathcal{X}$ and $n \ge 1$, jointly with probability at least $1 - \delta$, $|\tilde{f}^\star(\boldsymbol{x}) - \tilde{f}_n(\boldsymbol{x})| \le \beta_n(\delta)\sigma_n(\boldsymbol{x})$. It remains now only to observe that

$$|\tilde{f}^\star(\boldsymbol{x}) - \tilde{f}_n(\boldsymbol{x})| = |f^\star(\boldsymbol{x}) - f_n(\boldsymbol{x})|.$$

$\square$

### K.6 CONFIDENCE SETS FOR CLASSIFICATION (THEOREM 3.2)

We begin by re-stating Corollary 1 of Amani & Thrampoulidis (2020). Analogous results can be obtained from Theorem 1 of Zhang & Sugiyama (2023). Substantial work has studied the special case of binary feedback, $K = 2$ Faury et al. (2020); Pásztor et al. (2024).

Let $\boldsymbol{A}(\boldsymbol{x}; \boldsymbol{W}) \in \mathbb{R}^{K \times K}$ be the matrix satisfying $(\boldsymbol{A}(\boldsymbol{x}; \boldsymbol{W}))_{i,j} \doteq s_i(\boldsymbol{x}; \boldsymbol{W})(\mathbb{1}\{i = j\} - s_j(\boldsymbol{x}; \boldsymbol{W}))$. Equivalently, $\boldsymbol{A}(\boldsymbol{x}; \boldsymbol{W}) = \operatorname{diag}\{\boldsymbol{s}(\boldsymbol{x}; \boldsymbol{W})\} - \boldsymbol{s}(\boldsymbol{x}; \boldsymbol{W})\boldsymbol{s}(\boldsymbol{x}; \boldsymbol{W})^\top$. Based on this matrix, we define $L \doteq \sup_{\boldsymbol{x} \in \mathcal{X}, \boldsymbol{W} \in \mathcal{W}_B} \lambda_{\max}(\boldsymbol{A}(\boldsymbol{x}; \boldsymbol{W}))$ and $\kappa \doteq \sup_{\boldsymbol{x} \in \mathcal{X}, \boldsymbol{W} \in \mathcal{W}_B} 1/\lambda_{\min}(\boldsymbol{A}(\boldsymbol{x}; \boldsymbol{W}))$.

**Lemma K.5** (Corollary 1 of Amani & Thrampoulidis (2020)). *Assume $\boldsymbol{W}^\star \in \mathcal{W}_B$ and $\boldsymbol{W}^{\mathrm{pre}} = \boldsymbol{0}$. Let $\delta \in (0, 1)$ and set*

$$\tilde{\beta}_n(\delta) \doteq \sqrt{\lambda}\left(B + \frac{1}{2\sqrt{K}}\right) + \frac{2K^{3/2}d}{\sqrt{\lambda}} \log\left(\frac{2}{\delta}\sqrt{1 + \frac{n}{d\lambda}}\right). \tag{11}$$

*Then,*

$$\mathbb{P}\left(\forall n \geq 1, \boldsymbol{x} \in \mathcal{X} : \|\boldsymbol{s}_n(\boldsymbol{x}) - \boldsymbol{s}^\star(\boldsymbol{x})\|_2 \leq 2L\tilde{\beta}_n(\delta)\sqrt{\kappa(1 + 2B)}\|\boldsymbol{\phi}(\boldsymbol{x})\|_{\boldsymbol{V}_n^{-1}}\right) \geq 1 - \delta,$$

*where $\boldsymbol{V}_n \doteq \boldsymbol{\Sigma}_n + \kappa\lambda\boldsymbol{I}_d$.*

Our result follows from two auxiliary lemmas.

**Lemma K.6.** *For any $\boldsymbol{s}, \boldsymbol{s}' \in \mathbb{R}^K$, $d_{\mathrm{TV}}(\boldsymbol{s}, \boldsymbol{s}') \leq \frac{\sqrt{K}}{2}\|\boldsymbol{s} - \boldsymbol{s}'\|_2$.*

*Proof.* We have

$$d_{\mathrm{TV}}(\boldsymbol{s}, \boldsymbol{s}') = \frac{1}{2}\|\boldsymbol{s} - \boldsymbol{s}'\|_1 = \frac{1}{2}\sum_{i=1}^{K}|s_i - s_i'| \leq \frac{1}{2}\sqrt{K}\sqrt{\sum_{i=1}^{K}(s_i - s_i')^2} = \frac{\sqrt{K}}{2}\|\boldsymbol{s} - \boldsymbol{s}'\|_2$$

where the inequality follows from Cauchy-Schwarz. □

The following lemma is a standard result in the literature (Srinivas et al., 2009; Chowdhury & Gopalan, 2017; Pásztor et al., 2024), which we include here for completeness.

**Lemma K.7.** *Let $\sigma_n$ be as defined in Equation (2). Then, $\sqrt{\kappa\lambda}\|\boldsymbol{\phi}(\boldsymbol{x})\|_{\boldsymbol{V}_n^{-1}} = \sigma_n(\boldsymbol{x})$ for any $\boldsymbol{x} \in \mathcal{X}$.*

*Proof.* Note that $(\boldsymbol{\Sigma}_n + \kappa\lambda\boldsymbol{I}_d)\boldsymbol{\Phi}_n^\top = \boldsymbol{\Phi}_n^\top(\boldsymbol{K}_n + \kappa\lambda\boldsymbol{I}_n)$ which implies

$$(\boldsymbol{\Sigma}_n + \kappa\lambda\boldsymbol{I}_d)^{-1}\boldsymbol{\Phi}_n^\top = \boldsymbol{\Phi}_n^\top(\boldsymbol{K}_n + \kappa\lambda\boldsymbol{I}_n)^{-1}. \tag{12}$$

Further, by definition of $\boldsymbol{k}_n$, $\boldsymbol{k}_n(\boldsymbol{x}) = \boldsymbol{\Phi}_n\boldsymbol{\phi}(\boldsymbol{x})$ which permits writing

$$(\boldsymbol{\Sigma}_n + \kappa\lambda\boldsymbol{I}_d)\boldsymbol{\phi}(\boldsymbol{x}) = \boldsymbol{\Phi}_n^\top\boldsymbol{k}_n(\boldsymbol{x}) + \kappa\lambda\boldsymbol{\phi}(\boldsymbol{x})$$

and implies

$$\begin{aligned}
\boldsymbol{\phi}(\boldsymbol{x}) &= (\boldsymbol{\Sigma}_n + \kappa\lambda\boldsymbol{I}_d)^{-1}\boldsymbol{\Phi}_n^\top\boldsymbol{k}_n(\boldsymbol{x}) + \kappa\lambda(\boldsymbol{\Sigma}_n + \kappa\lambda\boldsymbol{I}_d)^{-1}\boldsymbol{\phi}(\boldsymbol{x}) \\
&\stackrel{(12)}{=} \boldsymbol{\Phi}_n^\top(\boldsymbol{K}_n + \kappa\lambda\boldsymbol{I}_n)^{-1}\boldsymbol{k}_n(\boldsymbol{x}) + \kappa\lambda(\boldsymbol{\Sigma}_n + \kappa\lambda\boldsymbol{I}_d)^{-1}\boldsymbol{\phi}(\boldsymbol{x})
\end{aligned} \tag{13}$$

We have

$$\begin{aligned}
k(\boldsymbol{x}, \boldsymbol{x}) &= \boldsymbol{\phi}(\boldsymbol{x})^\top\boldsymbol{\phi}(\boldsymbol{x}) \\
&\stackrel{(13)}{=} \left(\boldsymbol{\Phi}_n^\top(\boldsymbol{K}_n + \kappa\lambda\boldsymbol{I}_n)^{-1}\boldsymbol{k}_n(\boldsymbol{x}) + \kappa\lambda(\boldsymbol{\Sigma}_n + \kappa\lambda\boldsymbol{I}_d)^{-1}\boldsymbol{\phi}(\boldsymbol{x})\right)^\top\boldsymbol{\phi}(\boldsymbol{x}) \\
&= \boldsymbol{k}_n(\boldsymbol{x})^\top(\boldsymbol{K}_n + \kappa\lambda\boldsymbol{I}_n)^{-1}\boldsymbol{k}_n(\boldsymbol{x}) + \kappa\lambda\boldsymbol{\phi}(\boldsymbol{x})^\top(\boldsymbol{\Sigma}_n + \kappa\lambda\boldsymbol{I}_d)^{-1}\boldsymbol{\phi}(\boldsymbol{x}) \\
&= \boldsymbol{k}_n(\boldsymbol{x})^\top(\boldsymbol{K}_n + \kappa\lambda\boldsymbol{I}_n)^{-1}\boldsymbol{k}_n(\boldsymbol{x}) + \kappa\lambda\boldsymbol{\phi}(\boldsymbol{x})^\top\boldsymbol{V}_n^{-1}\boldsymbol{\phi}(\boldsymbol{x}).
\end{aligned}$$

Reordering this equation, we obtain

$$\kappa\lambda\|\boldsymbol{\phi}(\boldsymbol{x})\|_{\boldsymbol{V}_n^{-1}}^2 = \kappa\lambda\boldsymbol{\phi}(\boldsymbol{x})^\top\boldsymbol{V}_n^{-1}\boldsymbol{\phi}(\boldsymbol{x}) = k(\boldsymbol{x}, \boldsymbol{x}) - \boldsymbol{k}_n(\boldsymbol{x})^\top(\boldsymbol{K}_n + \kappa\lambda\boldsymbol{I}_n)^{-1}\boldsymbol{k}_n(\boldsymbol{x}) = \sigma_n^2(\boldsymbol{x}),$$

concluding the proof. □

We now proceed to prove a version of Theorem 3.2 with $\boldsymbol{W}^{\mathrm{pre}} = \boldsymbol{0}$.

**Theorem K.8.** *Assume $\boldsymbol{W}^\star \in \mathcal{W}_B$ and $\boldsymbol{W}^{\mathrm{pre}} = \boldsymbol{0}$. Let $\delta \in (0,1)$ and $\beta_n(\delta)$ as in Equation (3). Then*

$$\mathbb{P}(\forall n \geq 1, \boldsymbol{x} \in \mathcal{X} : d_{\mathrm{TV}}(\boldsymbol{s}_n(\boldsymbol{x}), \boldsymbol{s}^\star(\boldsymbol{x})) \leq \beta_n(\delta) \cdot \sigma_n(\boldsymbol{x})) \geq 1 - \delta.$$

*Proof.* We have

$$d_{\mathrm{TV}}(\boldsymbol{s}_n(\boldsymbol{x}), \boldsymbol{s}^\star(\boldsymbol{x})) \leq \frac{\sqrt{K}}{2} \|\boldsymbol{s}_n(\boldsymbol{x}) - \boldsymbol{s}^\star(\boldsymbol{x})\|_2 \qquad \text{(Lemma K.6)}$$

$$\overset{\text{w.h.p.}}{\leq} L\tilde{\beta}_n(\delta)\sqrt{K\kappa(1+2B)} \, \|\phi(\boldsymbol{x})\|_{\boldsymbol{V}_n^{-1}} \qquad \text{(Lemma K.5)}$$

$$= L\tilde{\beta}_n(\delta)\sqrt{\frac{K(1+2B)}{\lambda}}\sigma_n(\boldsymbol{x}). \qquad \text{(Lemma K.7)}$$

It remains to note that

$$L\tilde{\beta}_n(\delta)\sqrt{\frac{K(1+2B)}{\lambda}} = L\sqrt{K(1+2B)}\left(B + \frac{1}{2\sqrt{K}}\right) + \frac{2LK^2 d\sqrt{1+2B}}{\lambda}\log\left(\frac{2}{\delta}\sqrt{1+\frac{n}{d\lambda}}\right)$$

$$\leq 2\sqrt{K(1+2B)}\left[B + \frac{LK^{3/2}d}{\lambda}\log\left(\frac{2}{\delta}\sqrt{1+\frac{n}{d\lambda}}\right)\right] = \beta_n(\delta).$$

$\square$

With this we are ready to prove Theorem 3.2.

*Proof of Theorem 3.2.* We will proceed analogously to the proof of Theorem K.4. That is, our objective will be to bound the deviation of our biased model, which we refer to as $\boldsymbol{W}_n = \arg\min_{\boldsymbol{W}\in\mathcal{W}_B}\mathcal{L}^\lambda(\boldsymbol{W}; D_n)$, to $\boldsymbol{W}^\star$. Let

$$\tilde{\mathcal{L}}(\boldsymbol{W}'; D) \doteq -\sum_{(\boldsymbol{x},y)\in D} \log s_y((\boldsymbol{W}' + \boldsymbol{W}^{\mathrm{pre}})\phi(\boldsymbol{x})) \quad \text{and} \quad \tilde{\mathcal{L}}^\lambda(\boldsymbol{W}'; D) \doteq \tilde{\mathcal{L}}(\boldsymbol{W}'; D) + \frac{\lambda}{2}\|\boldsymbol{W}'\|_{\mathrm{F}}^2$$

with minimizer $\boldsymbol{W}_n' \doteq \arg\min_{\boldsymbol{W}':\|\boldsymbol{W}'\|_{\mathrm{F}}\leq B}\tilde{\mathcal{L}}^\lambda(\boldsymbol{W}'; D_n)$. We further define the residual weights $\tilde{\boldsymbol{W}}^\star \doteq \boldsymbol{W}^\star - \boldsymbol{W}^{\mathrm{pre}}$.

Next, we make the following observation: In their proof of Lemma K.5, Amani & Thrampoulidis (2020) bound

$$\|\boldsymbol{s}(\boldsymbol{f}(\boldsymbol{x}; \boldsymbol{W}_n')) - \boldsymbol{s}(\boldsymbol{f}(\boldsymbol{x}; \tilde{\boldsymbol{W}}^\star))\|_2 \leq \mathrm{const} \cdot \|\mathrm{vec}(\tilde{\boldsymbol{W}}^\star) - \mathrm{vec}(\boldsymbol{W}_n')\|_{\tilde{\boldsymbol{G}}(\tilde{\boldsymbol{W}}^\star, \boldsymbol{W}_n')} \quad (14)$$

where $\mathrm{const}$ is independent of $\boldsymbol{W}^\star, \boldsymbol{W}^{\mathrm{pre}}, \boldsymbol{W}_n'$ and the matrix $\tilde{\boldsymbol{G}}(\tilde{\boldsymbol{W}}^\star, \boldsymbol{W}_n')$ is invariant to a change of variables, i.e., $\tilde{\boldsymbol{G}}(\tilde{\boldsymbol{W}}^\star, \boldsymbol{W}_n') = \boldsymbol{G}(\boldsymbol{W}^\star, \boldsymbol{W}_n' + \boldsymbol{W}^{\mathrm{pre}})$ with $\tilde{\boldsymbol{G}}$ defined with respect to the loss $\tilde{\mathcal{L}}^\lambda$ and $\boldsymbol{G}$ defined with respect to the loss $\mathcal{L}^\lambda$. Theorem K.8 applies to $\boldsymbol{s}(\boldsymbol{f}(\boldsymbol{x}; \boldsymbol{W}_n'))$ and $\boldsymbol{s}(\boldsymbol{f}(\boldsymbol{x}; \boldsymbol{W}^\star - \boldsymbol{W}^{\mathrm{pre}}))$ since the regularization of $\tilde{\mathcal{L}}^\lambda$ is unbiased and the residual weights satisfy $\|\tilde{\boldsymbol{W}}^\star\|_{\mathrm{F}} = \|\boldsymbol{W}^\star - \boldsymbol{W}^{\mathrm{pre}}\|_{\mathrm{F}} \leq B$ by assumption.

Since $\tilde{\boldsymbol{W}}^\star - \boldsymbol{W}_n' = \boldsymbol{W}^\star - (\boldsymbol{W}_n' + \boldsymbol{W}^{\mathrm{pre}})$, the bounds of Equation (14) as well as Theorem K.8 then also apply to $\boldsymbol{s}(\boldsymbol{f}(\boldsymbol{x}; \boldsymbol{W}_n' + \boldsymbol{W}^{\mathrm{pre}})), \boldsymbol{s}(\boldsymbol{f}(\boldsymbol{x}; \boldsymbol{W}^\star))$. Observing that $\boldsymbol{W}_n = \boldsymbol{W}_n' + \boldsymbol{W}^{\mathrm{pre}}$ as a direct consequence of the change of variables completes the proof. $\square$

## L  QUALITATIVE EXAMPLES

### L.1  BALANCING RELEVANCE AND DIVERSITY

The following details the data space and prompt used in the qualitative example of Figure 3. We evaluate SIFT with $\lambda' = 1e-4$ and normalized embeddings, using the same embedding model as in our main experiments.

| Prompt |
| --- |
| What is the age of Michael Jordan and how many kids does he have? |

| Data space | |
| --- | --- |
| 1 | Michael Jordan was born on February 17, 1963, in Brooklyn, New York. |
| 2 | The age of Michael Jordan is 61 years. |
| 3 | Michael Jordan has five children. |
| 4 | Michael Jordan has 5 kids. |

Table 13: Query and information about Michael Jordan within data space

### L.2  EXAMPLES FROM THE PILE

The following provides examples of the data selected by SIFT for some queries from the Pile dataset.

---

**DeepMind Math**

**Query**

Find the second derivative of -222966*l*s**2 + 152*l*s - 8111*l + s**2 + 2 wrt s.
-445932*l + 2
What is the third derivative of 175*s**5 - 5*s**4 - 6106*s**3 + 53*s**2 + 169*s - 1753?
10500*s**2 - 120*s - 36636
What is the third derivative of 23679631*b**5 - 2*b**3 + 8*b**2 + 2*b - 6771326 wrt b?
1420777860*b**2 - 12
Find the second derivative of 3263785*m**4 + 141*m + 11251.
39165420*m**2
What is the second derivative of -47089*k*z**3 - 30997*k*z + 59*z**2 + 295*z wrt z?
. . .

**1st example**

What is the second derivative of 333510825*p**3 - 292254*p + 96 wrt p?
2001064950*p
What is the third derivative of -2862429*f**5 - 5*f**2 + 439*f - 557?
-171745740*f**2
What is the derivative of 32081*i**4 + 10*i**3 - 2*i - 9371139?
128324*i**3 + 30*i**2 - 2
Find the third derivative of -439900344*z**5 - 675051939*z**2 wrt z.
-26394020640*z**2
. . .

**2nd example**

What is the third derivative of 2322809*k**3 + 38*k**2 + 105*k + 236 wrt k?
13936854
What is the third derivative of 1242810*p**4 - 5*p**3 + 8382*p**2 + 491*p wrt p?
29827440*p - 30
Differentiate -23915071*o**4 + 25970708.
-95660284*o**3
Find the first derivative of -73333026*k - 218757639 wrt k.
-73333026

---

What is the second derivative of -9350*n**4 + 2047*n**2 - n - 42762066?
-112200*n**2 + 4094
. . .

---

**Enron Emails**

**Query**

Patti,

What do I do with this now? How do I get the $50? Can I wait and get a
series of months reimbursed later or do I have to go through this every month?

Fletch Sturm

**1st example**

Lucy,

Here is a rentroll for this week.

What is the outstanding balance on #1. It looks like 190 + 110(this week)=
300. I don't think we should make him pay late fees if can't communicate
clearly.

#2 still owe deposit?

#9 What day will she pay and is she going to pay monthly or biweekly.

Have a good weekend. I will talk to you next week.

In about two weeks we should know for sure if these buyers are going to buy
the property. I will keep you informed.

Phillip

**2nd example**

Kim,

I am getting parking deducted twice from my pay check. Who do I contact to
straighten that out?

Thanx

Chris

---

**FreeLaw**

**Query**

In the United States Court of Federal Claims
OFFICE OF SPECIAL MASTERS
No. 15-349V
Filed: August 20, 2015
Unpublished

*************************

ARIKA BROWNE, *
*
Petitioner, * Ruling on Entitlement; Concession;
* Influenza; Shoulder Injury ("SIRVA")

* Special Processing Unit ("SPU")
SECRETARY OF HEALTH *
AND HUMAN SERVICES, *
*
Respondent. *
*
**************************
Andrew Downing, Van Cott & Talamante, PLLC, Phoenix, AZ, for petitioner.
Claudia Barnes Gangi, U.S. Department of Justice, Washington, DC, for respondent.

RULING ON ENTITLEMENT 1

Vowell, Chief Special Master:

On April 7, 2015, Arika Browne filed a petition for compensation under the
National Vaccine Injury Compensation Program, 42 U.S.C. §300aa-10, et seq., 2 [the
"Vaccine Act" or "Program"]. Petitioner alleges that she suffered a left shoulder injury as
a result of the administration of an influenza vaccine. Petition at 1. The case was
assigned to the Special Processing Unit of the Office of Special Masters.

On August 20, 2015, respondent filed her Rule 4(c) report in which she concedes
. . .

**1st example**

In the United States Court of Federal Claims
OFFICE OF SPECIAL MASTERS
No. 15-349V
Filed: October 5, 2015
Unpublished

**************************
ARIKA BROWNE, *
*
Petitioner, * Damages Decision Based on Proffer;
* Influenza; Shoulder Injury ("SIRVA")
* Special Processing Unit ("SPU")
SECRETARY OF HEALTH *
AND HUMAN SERVICES, *
*
Respondent. *
*
**************************
Andrew Downing, Van Cott & Talamante, PLLC, Phoenix, AZ, for petitioner.
Claudia Barnes Gangi, U.S. Department of Justice, Washington, DC for respondent.

DECISION AWARDING DAMAGES 1

Dorsey, Chief Special Master:

On April 7, 2015, Arika Browne filed a petition for compensation under the
National Vaccine Injury Compensation Program, 42 U.S.C. §300aa-10, et seq., 2 [the
"Vaccine Act" or "Program"]. Petitioner alleges that she suffered a left shoulder injury as
a result of the administration of an influenza vaccine. Petition at 1. The case was
assigned to the Special Processing Unit of the Office of Special Masters.

On August 20, 2015, a ruling on entitlement was issued, finding petitioner entitled
. . .

**2nd example**

In the United States Court of Federal Claims
OFFICE OF SPECIAL MASTERS
No. 15-936V
Filed: November 23, 2015
Unpublished

\*\*\*\*\*\*\*\*\*\*\*\*\*\*\*\*\*\*\*\*\*\*\*\*\*\*
JENNIFER SIEKIERSKI, \*
\*
Petitioner, \* Ruling on Entitlement; Concession;
\* Influenza;
\* Shoulder Injury ("SIRVA");
SECRETARY OF HEALTH \* Special Processing Unit ("SPU")
AND HUMAN SERVICES, \*
\*
Respondent. \*
\*
\*\*\*\*\*\*\*\*\*\*\*\*\*\*\*\*\*\*\*\*\*\*\*\*\*\*
Katheryn Lee Bruns, Faraci Lange, LLP, Rochester, NY, for petitioner.
Julia Wernett McInerny, U.S. Department of Justice, Washington, DC, for respondent.

RULING ON ENTITLEMENT 1

Dorsey, Chief Special Master:

On August 26, 2015, Petitioner filed a petition for compensation under the
National Vaccine Injury Compensation Program, 42 U.S.C. §300aa-10, et seq., 2 [the
"Vaccine Act" or "Program"]. Petitioner alleges that she experienced a shoulder injury
related to vaccine administration ("SIRVA") as a result of her receipt of an influenza
vaccine on November 4, 2014. Petition at 1. The case was assigned to the Special
Processing Unit of the Office of Special Masters.

On November 23, 2015, respondent filed her Rule 4(c) report in which she
. . .

---

**GitHub**

**Query**

```php
<?php

/*
 * This file is part of PHPExifTool.
 *
 * (c) 2012 Romain Neutron ¡imprec@gmail.com¿
 *
 * For the full copyright and license information, please view the LICENSE
 * file that was distributed with this source code.
 */

namespace PHPExiftool
Driver
Tag
QuickTime;

use JMS
Serializer
Annotation
ExclusionPolicy;
```

```
use PHPExiftool
Driver
AbstractTag;

/**
 * @ExclusionPolicy("all")
 */
class UserDataDji extends AbstractTag
{

protected $Id = 'xa9dji';

protected $Name = 'UserData_dji';

protected $FullName = 'QuickTime::UserData';

protected $GroupName = 'QuickTime';

protected $g0 = 'QuickTime';

protected $g1 = 'QuickTime';

protected $g2 = 'Video';

protected $Type = 'undef';

protected $Writable = false;

protected $Description = 'User Data dji';

protected $flag_Binary = true;
}
```

**1st example**

```php
<?php

/*
 * This file is part of PHPExifTool.
 *
 * (c) 2012 Romain Neutron ¡imprec@gmail.com¿
 *
 * For the full copyright and license information, please view the LICENSE
 * file that was distributed with this source code.
 */

namespace PHPExiftool
Driver
Tag
QuickTime;

use JMS
Serializer
Annotation
ExclusionPolicy;
use PHPExiftool
Driver
AbstractTag;
```

```php
/**
* @ExclusionPolicy("all")
*/
class UserDataUid extends AbstractTag
{

protected $Id = 'xa9uid';

protected $Name = 'UserData_uid';

protected $FullName = 'QuickTime::UserData';

protected $GroupName = 'QuickTime';

protected $g0 = 'QuickTime';

protected $g1 = 'QuickTime';

protected $g2 = 'Video';

protected $Type = 'undef';

protected $Writable = false;

protected $Description = 'User Data uid';

protected $flag_Binary = true;
}
```

**2nd example**
```php
<?php

/*
* This file is part of PHPExifTool.
*
* (c) 2012 Romain Neutron ¡imprec@gmail.com¿
*
* For the full copyright and license information, please view the LICENSE
* file that was distributed with this source code.
*/

namespace PHPExiftool
Driver
Tag
QuickTime;

use JMS
Serializer
Annotation
ExclusionPolicy;
use PHPExiftool
Driver
AbstractTag;

/**
* @ExclusionPolicy("all")
*/
```

```
class MovieData extends AbstractTag
{
protected $Id = 'mdat';

protected $Name = 'MovieData';

protected $FullName = 'QuickTime::Main';

protected $GroupName = 'QuickTime';

protected $g0 = 'QuickTime';

protected $g1 = 'QuickTime';

protected $g2 = 'Video';

protected $Type = '?';

protected $Writable = false;

protected $Description = 'Movie Data';

protected $flag_Binary = true;
}
```

