# OpenReview forum: "Efficiently Learning at Test-Time: Active Fine-Tuning of LLMs"
_ICLR.cc/2025/Conference — ICLR 2025 Poster_

### Official Review · Reviewer_RGVx · 2024-10-31

**Soundness:** 3
**Presentation:** 2
**Contribution:** 3
**Rating:** 6
**Confidence:** 3

**Summary:**

This paper presents SIFT, a data selection algorithm aimed at improving test-time fine-tuning of LLMs by reducing redundancy in data selection. Unlike traditional Nearest Neighbor retrieval, which often selects redundant examples, SIFT combines retrieval and active learning principles to minimize model uncertainty for each prompt. This approach also enables adaptive fine-tuning, dynamically adjusting compute based on anticipated performance gains. Experiments on the Pile dataset indicate that SIFT achieves better efficiency and robustness than Nearest Neighbor retrieval, with minimal computational overhead. The authors provide an open-source library, activeft, for integration and reproducibility.

**Strengths:**

- The paper introduces SIFT, a well-motivated algorithm that combines retrieval and active learning, effectively addressing data redundancy issues in traditional Nearest Neighbor retrieval for LLM fine-tuning.

- SIFT’s adaptive fine-tuning, which adjusts test-time compute based on predicted performance gains, is an efficient and practical innovation that can optimize resource use, especially in computationally constrained environments.

- The paper provides both theoretical proofs and empirical evidence to demonstrate that SIFT reduces model uncertainty and improves fine-tuning outcomes, adding robustness and credibility to the proposed approach.

- The release of the *activeft* library as a drop-in replacement for Nearest Neighbor retrieval supports transparency and facilitates future research in prompt-specific fine-tuning methods.

**Weaknesses:**

1. The motivation and definition of the task lack clarity, particularly in distinguishing test-time fine-tuning from standard fine-tuning on selected data. It remains unclear why test-time fine-tuning is necessary in this context and how it fundamentally differs from simply fine-tuning on pre-selected data, which may impact understanding of the novelty and importance of SIFT’s approach.

2. The paper does not fully specify scenarios where test-time fine-tuning with SIFT would be most beneficial. This omission makes it difficult to assess the generalizability and practical applications of the method, particularly for those who unfamiliar with prompt-specific fine-tuning needs.

3. Although SIFT is described as having minimal overhead, the adaptive fine-tuning process could introduce additional complexity in real-time settings. A more thorough breakdown of the computational costs associated with adaptive adjustments would improve understanding of its efficiency.

**Questions:**

See weakness.

---

> ### Author Response · Authors · 2024-11-14
>
> Thank you for reviewing our paper and your detailed comments! Please find our responses to the concerns raised.
> Please let us know if you have any further concerns or suggestions.
>
> > The release of the `activeft` library as a drop-in replacement for Nearest Neighbor retrieval supports transparency and facilitates future research in prompt-specific fine-tuning methods.
>
> Thank you for highlighting this!
>
> > The motivation and definition of the task lack clarity, particularly in distinguishing test-time fine-tuning from standard fine-tuning on selected data. It remains unclear why test-time fine-tuning is necessary in this context and how it fundamentally differs from simply fine-tuning on pre-selected data, which may impact understanding of the novelty and importance of SIFT’s approach.
>
> We added a number of additional results on state-of-the-art models in our revised paper, and added Figure 7, showing the absolute performance gain of test-time fine-tuning over the base model. Please see our main official comment for more details.
> We hope that our inclusion of additional results showing the large absolute performance gains that can be achieved by test-time fine-tuning of a base model (even on "in-distribution data" such as the Pile), further supports the setting of `"learning at test-time".
> It seems interesting to us that test-time fine-tuning improves the performance of a state-of-the-art model like Phi-3 (3.8B) much more than going to the largest Phi-3 (14B) model. We make the same observation with Llama-3.2 models.
>
> We see test-time fine-tuning as the extreme end of a spectrum, with pre-training (where we aim to fit the entire data distribution) on the opposite end.
> Loosely speaking: In the same way that we would expect a model fine-tuned on coding to do better on competitive programming than the base model, we would expect a model fine-tuned on competitive programming to do better on competitive programming than the model fine-tuned more broadly on coding. Within this example, our work studies models fine-tuned to a specific competitive programming task. We would expect that such a fine-tuned model tailored specifically to the task at hand can perform outperform a model that is learning to "solve all competitive programming tasks at once" since it has to use its weights only to encode the task-specific information, while being able to ignore all other information.
> For example, recently, *test-time* fine-tuning has been applied in state-of-the-art methods for the ARC challenge [e.g., 1].

---

> > ### Author Response · Authors · 2024-11-14
> >
> > > The paper does not fully specify scenarios where test-time fine-tuning with SIFT would be most beneficial. This omission makes it difficult to assess the generalizability and practical applications of the method, particularly for those who unfamiliar with prompt-specific fine-tuning needs.
> >
> > Thank you for highlighting this! In our revised paper, we tried to address this by adding further empirical evidence supporting the setting of "learning at test-time".
> > Please find our detailed response below.
> > - **Scenarios where SIFT improves *most* over NN:** We appreciate the concern to identify the cases where SIFT is *most* beneficial.
> >     The main improvement of SIFT over NN lies in SIFT's robustness to duplication of information. We would argue that virtually every dataset has some degree of information duplication, and indeed, even with the carefully designed Pile dataset, SIFT leads to a significant improvement over NN. As we show in Figure 6, an additional advantage of SIFT is that it adaptively determines the correct number of times to fine-tune on a given data point: SIFT identifies the rare cases where fine-tuning on the same data point repeatedly is beneficial.
> >     We would like to emphasize that our experiments suggest that SIFT is *always* beneficial (across all datasets in the Pile).
> >
> >     Finally, we would like to mention that we see the introduction of SIFT as our main contribution, with test-time fine-tuning as a testbed. We expect that SIFT may be applicable more broadly to the fine-tuning of LLMs as a drop-in replacement for NN (as, for example, used by [2]), which we believe to be an exciting direction for future work.
> >
> > - **Scenarios where test-time fine-tuning improves over the base model:** We would like to refer you to our updated results, which we summarized in the main official comment. These results show that test-time fine-tuning (with SIFT) always improves upon the base model (on all datasets of the Pile), and often significantly — even with state-of-the-art models.
> >     We find that test-time fine-tuning with SIFT (and Llama-3.2-3B as base model) achieves a new state-of-the-art on the Pile language modeling benchmark.
> >     Additionally, we compare test-time fine-tuning to adding data (selected by SIFT) to the context.
> >     Interestingly, we find that test-time fine-tuning tends to have larger performance gains than "learning from context" on certain datasets such as "DeepMind Math", "GitHub", and "FreeLaw" (court opinions). For this, please see our extended results in Appendix F.2 of the revised paper.
> >
> > > Although SIFT is described as having minimal overhead, the adaptive fine-tuning process could introduce additional complexity in real-time settings. A more thorough breakdown of the computational costs associated with adaptive adjustments would improve understanding of its efficiency.
> >
> > Thank you for highlighting this important consideration! We added results for the compute cost with all evaluated models in Figure 22 of Appendix I.3 of the updated manuscript.
> > While this compute cost is slightly higher than the cost of adding examples to an LLMs context (with a small context window), we believe that test-time fine-tuning might be an interesting alternative to RAG systems (based on adding data to context) in tasks where latency is not the primary consideration such as in a medical or financial setting.
> > Moreover, it seems interesting that the cost of test-time fine-tuning grows linearly with the amount of data, whereas the cost of adding data to the context of a transformer model grows quadratically.
> > Finally, and as you also highlight, our work suggests new approaches to dynamic compute allocation, which we believe to be effective tools for reducing the computational cost of learning at test-time.
> >
> >
> > ---
> >
> > We hope to have addressed your concerns.
> > We would greatly appreciate it if you could reconsider the score based on our response and results.
> >
> > ---
> >
> > References:
> > 1. Akyürek et al. "The Surprising Effectiveness of Test-Time Training for Abstract Reasoning."
> > 2. Xia et al. "LESS: Selecting Influential Data for Targeted Instruction Tuning."

---

### Official Review · Reviewer_RwvN · 2024-11-03

**Soundness:** 3
**Presentation:** 3
**Contribution:** 3
**Rating:** 8
**Confidence:** 3

**Summary:**

The paper introduces SIFT, a method for selecting informative data to fine-tune large language models (LLMs). It critiques the current Nearest Neighbors retrieval approach which selects redundant data and proposes SIFT, inspired by transductive active learning, to select relevant and diverse data for effective fine-tuning. Additionally, it offers a library as a drop-in replacement for Nearest Neighbor retrieval.

**Strengths:**

1. Theoretical Foundation: The paper is theoretically robust and well-motivated, offering comprehensive analysis to demonstrate its effectiveness.
2. Organization and Insight: It is well-organized and self-consistent, providing thorough discussions on the research topic and outlining both current and future research directions.

**Weaknesses:**

1. Inference Cost: Comparison with Nearest Neighbor, this method may require more inference time. It would be beneficial to compare inference times with Nearest Neighbor across different datasets to quantify this.
2. Broader Evaluation: The paper could explore effectiveness on more datasets and larger models, such as LLaMA-3, to validate its scalability and generalizability.

**Questions:**

See weaknesses.

**Details Of Ethics Concerns:**

No ethics concerns

---

> ### Author Response · Authors · 2024-11-14
>
> Thank you for reviewing our paper! You can find below our response to your questions and concerns. Please let us know if you have any further concerns or suggestions.
>
> > Organization and Insight: It is well-organized and self-consistent, providing thorough discussions on the research topic and outlining both current and future research directions.
>
> Thank you! We are glad to hear that you enjoyed reading our paper.
>
> > Inference Cost: Comparison with Nearest Neighbor, this method may require more inference time. It would be beneficial to compare inference times with Nearest Neighbor across different datasets to quantify this.
>
> Thank you for highlighting this! We conducted a comparison in Figure 4, which shows the compute overhead of using SIFT as opposed to Nearest Neighbor (NN) retrieval.
> These results are averaged across all test examples, with bootstrap error bars.
> While SIFT costs slightly more than NN for retrieval, relative to the combined cost for retrieval and training, SIFT costs essentially as much as NN (i.e., incurs almost no additional overhead).
> We hope this addresses your question adeptly.
>
> > Broader Evaluation: The paper could explore effectiveness on more datasets and larger models, such as LLaMA-3, to validate its scalability and generalizability.
>
> The Pile dataset is a huge meta-dataset that contains a large variety of datasets, each of which testing different domains of human language. These include web text, Q&A, Wikipedia, scientific papers, code, math, court opinions, patent applications, emails, and numerous more.
> We therefore believe that the Pile dataset offers a very rich testbed for evaluation of language modeling performance.
> Nevertheless, we agree that it would be interesting to evaluate SIFT on downstream non-perplexity tasks, and we are inclined to leave such an evaluation to future work.
>
> Regarding the tested models, we added additional evaluations on Llama-3.2 (1B \& 3B). Moreover, we evaluated the base performance (without test-time fine-tuning) of many state-of-the-art models (including Phi-3-14B, Gemma-2-27B), and find that our test-time fine-tuned models achieve vastly greater performance.
> We find that Llama-3.2-3B with test-time fine-tuning achieves a new state-of-the-art on the Pile language modeling benchmark.
> We describe these new results in greater detail in our main official comment above.
>
> ---
>
> We hope to have addressed your concerns and are thankful for your recognition of our contributions.
> We would be happy to answer any remaining questions or concerns.

---

> ### Comment · Reviewer_RwvN · 2024-11-26
> **Official Comment by Reviewer RwvN**
>
> Thanks for all your responses, Addressing most of my concerns, I keep my score.

---

> > ### Author Response · Authors · 2024-11-27
> >
> > Thank you for supporting acceptance of our paper! We very much appreciate your efforts in reviewing and reading the rebuttal.

---

### Official Review · Reviewer_p5NR · 2024-11-04

**Soundness:** 3
**Presentation:** 4
**Contribution:** 4
**Rating:** 8
**Confidence:** 4

**Summary:**

This paper introduces SIFT, a data selection algorithm aimed at improving the fine-tuning of language models by addressing the limitations of Nearest Neighbors retrieval, which often selects redundant data. SIFT reduces uncertainty about model responses by optimizing overall information gain and accounting for information duplication. The authors demonstrate that SIFT outperforms Nearest Neighbor retrieval in fine-tuning at test time in experiments, with minimal computational overhead. Additionally, they show that their uncertainty estimates can predict performance gains, leading to an adaptive algorithm that optimally allocates computational resources.

**Strengths:**

1. This paper verifies a meaningful yet little explored topic in LLM application, giving comprehensive and solid discussion on the challenge and the proposed solution. The whole paper is clearly organized and easy to follow.
2. The SIFT strategy is sopported with solid theoretical induction as well as experimental evidences.
3. The experimental study is convincing and covers a wide range of datasets.

**Weaknesses:**

1. The interpretability of this work might be improved by giving some instances of data selection.
2. How is the possibility and gain of combining this strategy with orthogonal methods of LLM finetuning?

**Questions:**

See weaknesses.

---

> ### Author Response · Authors · 2024-11-14
>
> Thank you for reviewing our paper! Please find our response to your question.
> Please let us know if you have any further concerns or suggestions.
>
> > This paper verifies a meaningful yet little explored topic in LLM application, giving comprehensive and solid discussion on the challenge and the proposed solution. The whole paper is clearly organized and easy to follow.
>
> Thank you!
>
> > The experimental study is convincing and covers a wide range of datasets.
>
> Thank you!
>
> > The interpretability of this work might be improved by giving some instances of data selection.
>
> Thank you for this suggestion! We very much agree and added examples to Appendix L.2 (end of the pdf). We believe that this makes our results easier to interpret.
>
> > How is the possibility and gain of combining this strategy with orthogonal methods of LLM finetuning?
>
> Also thank you for highlighting this! We agree that many techniques from LLM fine-tuning might also be applicable to our setting.
> Indeed, our work shows that parameter-efficient fine-tuning via low-rank adaptation (LoRA) is effective for test-time fine-tuning.
> We believe that the evaluation of further techniques is a great idea for future work.
>
> ---
>
> We are glad to see your recognition of our contribution.
> We would like to highlight the results added with our revision, which achieve a new state-of-the-art on the Pile language modeling benchmark (please see our main official comment above).
> We would be happy to answer any remaining questions or concerns!

---

### Official Review · Reviewer_Uxgn · 2024-11-04

**Soundness:** 3
**Presentation:** 3
**Contribution:** 3
**Rating:** 6
**Confidence:** 4

**Summary:**

The paper proposes a novel data selection algorithm called SIFT (Select Informative data for Fine-Tuning) aimed at improving test-time fine-tuning for large language models (LLMs). The paper addresses the limitations of Nearest Neighbor retrieval in data selection by combining ideas from retrieval and active learning. Unlike traditional Nearest Neighbor methods that can select redundant data, SIFT optimizes for information gain, reducing redundancy in the selected examples and enhancing the performance of LLMs during test-time fine-tuning. Evaluated on the Pile dataset, SIFT shows consistent improvement over traditional methods with minimal computational overhead, achieving a robust balance between relevance and diversity in data selection.

**Strengths:**

Effective Data Selection with SIFT: SIFT combines uncertainty and diversity to select non-redundant data, enhancing test-time fine-tuning efficiency compared to traditional methods.
Comprehensive Experiments: Wide experiments on the Pile dataset demonstrate SIFT’s effectiveness, consistently outperforming traditional Nearest Neighbor and other baseline methods in fine-tuning efficiency and model performance.

**Weaknesses:**

This paper has the following drawbacks.

**Complexity and Clarity of Method Presentation**: The paper’s explanation of the SIFT algorithm, especially on pages 4 and 5, could benefit from clearer descriptions and simplification of symbols. The complex notation and detailed mathematical formulation may obscure understanding for readers, especially those less familiar with active learning or information-theoretic approaches. Providing a more accessible walkthrough or visual aids could improve clarity.

**Sensitivity to Hyperparameters**: The method relies on certain hyperparameters, such as the regularization parameter (λ′), which can significantly impact SIFT’s performance. Techniques for automatic tuning or guidelines for parameter selection would make the approach more user-friendly and robust.

**Relevance and Diversity**. It is strange to mention that "we provide an example of how SIFT balances relevance and diversity, where we also see that the parameter"

**Limited Novelty in Uncertainty-Based Data Selection**: The use of uncertainty as a criterion for data selection is not a novel concept and is a common technique in active learning. Could the author explain the difference with existing works such as [1,2]. Furthermore, could you discuss the novelty of utilizing uncertainty in your LLM background?


[1] Symmetric Uncertainty-Aware Feature Transmission for Depth Super-Resolution,
[2] LogitNorm: Mitigating Neural Network Overconfidence with Logit Normalization

**Questions:**

Please refer to Weaknesses.

**Details Of Ethics Concerns:**

This paper does not involve the ethical concerns.

---

> ### Author Response · Authors · 2024-11-14
>
> Thank you for your review and detailed comments! You can find below our detailed response to your questions and concerns. Please let us know if you have any further concerns or suggestions.
>
> > *Effective Data Selection with SIFT:* SIFT combines uncertainty and diversity to select non-redundant data, enhancing test-time fine-tuning efficiency compared to traditional methods. *Comprehensive Experiments:* Wide experiments on the Pile dataset demonstrate SIFT’s effectiveness, consistently outperforming traditional Nearest Neighbor and other baseline methods in fine-tuning efficiency and model performance.
>
> Thank you!
>
> > *Complexity and Clarity of Method Presentation:* The paper’s explanation of the SIFT algorithm, especially on pages 4 and 5, could benefit from clearer descriptions and simplification of symbols. The complex notation and detailed mathematical formulation may obscure understanding for readers, especially those less familiar with active learning or information-theoretic approaches. Providing a more accessible walkthrough or visual aids could improve clarity.
>
> Thank you for raising this important point! We added a discussion of an example to Appendix C.1, which shows the close connection of SIFT to Nearest Neighbor retrieval. We hope that this supports the understanding for readers more familiar with the literature of retrieval-augmented generation and LLM vector search methods.
>
> > *Sensitivity to Hyperparameters:* The method relies on certain hyperparameters, such as the regularization parameter ($\lambda'$), which can significantly impact SIFT’s performance. Techniques for automatic tuning or guidelines for parameter selection would make the approach more user-friendly and robust.
>
> Thank you for suggesting the tuning of $\lambda'$. We agree that this could be an effective approach to further improve performance.
>
> We would like to emphasize that $\lambda'$ is the only additional hyperparameter of SIFT compared to other data selection methods such as Nearest Neighbor (NN) retrieval.
> Our experiments show that — even without any adaptive tuning — SIFT outperforms NN for *any* tested $\lambda'$ between $10^{-8}$ and $10^1$ (cf. Figure 5, right).
> Moreover, as we show in our ablations (cf. Table 5 in Appendix J), SIFT outperforms or performs on-par with NN for all such $\lambda'$ on *any* of the diverse individual datasets of the Pile.
> This suggests that SIFT is *robust* to the choice of $\lambda'$, and that any "reasonable" choice leads to strong performance.
> We added a summary of the above to "Insight 2" in Section 5.
>
> > *Relevance and Diversity.* It is strange to mention that "we provide an example of how SIFT balances relevance and diversity, where we also see that the parameter"
>
> Thank you for pointing this out.
> What we meant to say is that there exists a tradeoff between selecting the most relevant data (i.e., repeatedly selecting the *nearest neighbor*) and selecting diverse data (i.e., where each data point is as dissimilar as possible to all other data points).
> The parameter $\lambda'$ allows to smoothly interpolate between both extremes.
> We added Figure 9 (left) in Appendix C to illustrate this tradeoff and how it is governed by $\lambda'$.
> The performance implications of this tradeoff are visualized in Figure 5 (right), where SIFT converges to NN-F as $\lambda' \to \infty$ and to "purely maximizing diversity" of the selected data as $\lambda' \to 0$.
> These experiments show that not choosing either one of these two extremes (i.e., not choosing $\lambda' \to 0$ or $\lambda' \to \infty$) can vastly improve performance, robustly across a large range of $\lambda'$-values.

---

> > ### Author Response · Authors · 2024-11-14
> >
> > > *Limited Novelty in Uncertainty-Based Data Selection:* The use of uncertainty as a criterion for data selection is not a novel concept and is a common technique in active learning. Could the author explain the difference with existing works such as [1,2]. Furthermore, could you discuss the novelty of utilizing uncertainty in your LLM background?
> >
> > Our work is building primarily upon recent work on Transductive Active Learning (TAL) [1].
> > TAL is a generalization of classical Active Learning (AL) that aims to *reduce uncertainty* at specific prediction targets, as opposed to classical AL that aims to reduce uncertainty everywhere. One of the most classical approaches to AL is "Uncertainty Sampling" which we compare against as a baseline (cf. Section 5).
> > The key limitation of AL is that methods do not take "relevance" with respect to a prediction target into account (i.e., answering the concrete prompt). Instead, these methods can be thought of as the "opposite extreme of NN" where the aim is only to get as diverse samples as possible.
> > This is why AL is commonly applied to pre-training where there is no notion of a concrete prompt that needs to be responded to.
> > SIFT unifies these two extremes and shows that combining retrieval and diversity can vastly outperform either of the two extremes.
> >
> > "Uncertainty" is a ubiquitous notion in machine learning. We would like to caution that uses of this term across areas in ML and even within the literature of AL are not interchangeable.
> > In this work, we do not at all claim to have proposed the "first method that uses some notion of uncertainty".
> > Instead, we claim to be the first to develop this specific notion of uncertainty for LLMs, which is tractable and leads to large gains in performance with test-time fine-tuning.
> > While there are many works using some notion of uncertainty, such as [1,2] suggested by the reviewer, we compare against the most popular and most closely related baselines.
> > Appendices B.2 and B.3 include a comprehensive overview of these methods and their development.
> >
> > To the best of our knowledge, this work is the first to use this particular notion of uncertainty for data selection with LLMs, showing that it is tractable and showing promising performance.
> >
> > ---
> >
> > We believe that our work is highly relevant since (1) SIFT is a novel data selection method for LLMs which (2) substantially improves the performance and robustness of test-time fine-tuning.
> > Further, our work is the first to show that test-time fine-tuning can lead to large performance gains on state-of-the-art LLMs.
> > We added additional results with strong base models to our revised paper, which achieve a new state-of-the-art on the Pile language modeling benchmark (please see our main official comment).
> >
> > We hope to have addressed your questions and concerns.
> > We would greatly appreciate it if you could reconsider the score based on our response and results.
> >
> > ---
> >
> > References:
> > 1. Hübotter et al. "Transductive Active Learning: Theory and Applications."

---

### Author Response · Authors · 2024-11-14
**Incorporated feedback and new results**

We thank all reviewers for their feedback!

Based on the reviewers' feedback, we extended our empirical evaluation of SIFT and test-time fine-tuning by evaluating a larger selection of state-of-the-art models. Additionally to GPT-2, GPT-2-large, Phi-3 (3.8B), we evaluated test-time fine-tuning with Llama-3.2 (both 1B \& 3B).
Further, we evaluated the base model performance of a large selection of state-of-the-art models, such as Phi-3 (14B) and Gemma-2 (27B).

We find that Llama-3.2 (3B) with test-time fine-tuning and SIFT achieves a new state-of-the-art on the Pile language modeling benchmark, substantially outperforming the previous leading 130B parameter model, as well as other state-of-the-art models.
In our estimation, these results indicate that test-time fine-tuning with SIFT can be an effective way to improve model performance via test-time compute (even on "in-distribution tasks" such as the Pile), which is orthogonal to increasing pre-training compute.

We uploaded a revised version of our paper which includes these new results. We made the following changes:
- We added Figure 7 which shows the absolute performance of the base model, test-time fine-tuning, and adding retrieved data to the context.
- We added "Insight 4" to Section 5, which summarizes Figure 7.
- We added Appendix A, which compares our new results on the Pile language modeling benchmark to results of existing models.
- We added Appendix F, which includes extended per-dataset results for all models, expanding on the results in Table 1.
- In Appendix L.2, we added additional qualitative examples as suggested by Reviewer p5NR.

**Summary of our main contributions:**
- We show that test-time fine-tuning is a promising paradigm for language modeling that consistently improves state-of-the-art models.
- We show that SIFT is a robust and significant improvement over Nearest Neighbor retrieval for data selection.
- We give a principled theoretical motivation for SIFT, which highlights when and how it improves over Nearest Neighbor retrieval.

---

### Meta-Review · Area_Chair_yBMP · 2024-12-20

**Metareview:**

This paper received four positive ratings, with all reviewers generally inclined to accept it. The paper introduces SIFT, a data selection algorithm designed to enhance the fine-tuning of language models by addressing the limitations of Nearest Neighbors retrieval, which often selects redundant data. SIFT reduces uncertainty in model responses by optimizing overall information gain and mitigating information duplication. The paper is well-motivated, providing a comprehensive analysis to demonstrate its effectiveness. It offers both theoretical proofs and empirical evidence to show that SIFT reduces model uncertainty and improves fine-tuning outcomes, thereby adding robustness and credibility to the proposed approach. The SIFT strategy is supported by solid theoretical foundations and experimental evidence. The experimental study is convincing and covers a wide range of datasets. The authors have addressed the concerns raised, resolving most of the doubts. Therefore, the Area Chair (AC) recommends accepting the paper.

**Additional Comments On Reviewer Discussion:**

The authors have addressed the raised concerns, all the reviewers recommend acceptance after the rebuttal.

---

### Decision · Program_Chairs · 2025-01-22

Accept (Poster)